# Beyond Softmax and Entropy: Convergence Rates of Policy Gradients with $f$-SoftArgmax Parameterization & Coupled Regularization

**Safwan Labbi[1], Daniil Tiapkin[1,2], Paul Mangold[1], Eric Moulines[3,4]**

[1] CMAP, CNRS, École Polytechnique, Institut Polytechnique de Paris, 91120 Palaiseau, France

[2] Université Paris-Saclay, CNRS, LMO, 91405, Orsay, France

[3] Mohamed bin Zayed University of Artificial Intelligence, UAE

[4] LRE EPITA , 94270 Le Kremlin-Bicêtre, France

{safwan.labbi, daniil.tiapkin, paul.mangold}@polytechnique.edu
eric.moulines@mbzuai.ac.ae

## Abstract

Policy gradient methods are known to be highly sensitive to the choice of policy parameterization. In particular, the widely used softmax parameterization can induce ill-conditioned optimization landscapes and lead to exponentially slow convergence. Although this can be mitigated by preconditioning, this solution is often computationally expensive. Instead, we propose replacing the softmax with an alternative family of policy parameterizations based on the generalized $f$-softargmax. We further advocate coupling this parameterization with a regularizer induced by the same $f$-divergence, which improves the optimization landscape and ensures that the resulting regularized objective satisfies a Polyak–Łojasiewicz inequality. Leveraging this structure, we establish the *first explicit non-asymptotic last-iterate convergence guarantees* for stochastic policy gradient methods for finite MDPs *without any form of preconditioning*. We also derive sample-complexity bounds for the unregularized problem and show that $f-\texttt{PG}$ with Tsallis divergences achieves *polynomial sample complexity* in contrast to the exponential complexity incurred by the standard softmax parameterization.

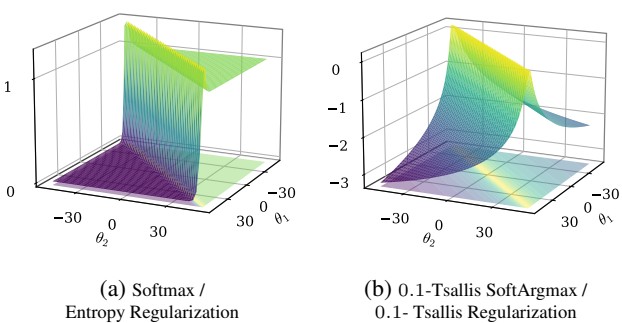

Figure 1: Regularized value landscapes (with temperature $\lambda = 1$) for a one-state, two-action MDP: softmax with entropy (left) versus $\alpha$-Tsallis SoftArgmax with $\alpha$-Tsallis regularization (right, $\alpha = 0.1$). The value of the classical coupling Entropy–Softmax is much flatter than for our proposed coupling Tsallis–Tsallis. Using the latter removes flat areas that are far from the solution, allowing *policy gradient methods to escape the gravitational pull*.

(a) Softmax / Entropy Regularization

(b) 0.1-Tsallis SoftArgmax / 0.1- Tsallis Regularization

# 1 Introduction

Policy gradient methods are a cornerstone of modern reinforcement learning (RL) and underpin many of its most notable successes. Algorithms such as Trust-Region Policy Optimization (TRPO; Schulman et al., 2015) and Proximal Policy Optimization (PPO; Schulman et al., 2017) have demonstrated strong empirical performance across a wide range of domains (Berner et al., 2019; Akkaya et al., 2019). Despite these successes, it has become increasingly clear that the performance and convergence behavior of policy gradient methods are highly sensitive to seemingly low-level design choices, among which the choice of policy parameterization plays a central role (Hsu et al., 2020).

Table 1: Comparison of the performance of *scalable*[1] policy gradient methods with global convergence guarantees on the unregularized objective.

| | Configuration | | Stochastic | Last iterate | Explicit Rates[3] |
|---|---|---|---|---|---|
| | Parameterization. | Regularization. | | | |
| Mei et al. (2020b) | softmax | ✗ | ✗ | ✓ | ✗ |
| Mei et al. (2020a) | EscortTransform | ✗ | ✗ | ✓ | ✗ |
| Zhang et al. (2021) | softmax | Log-Barrier | ✓ | ✗ | ✓$_{\text{poly}}^{(4)}$ |
| Liu et al. (2025) | Hadamard | ✗ | ✗ | ✓ | ✗ |
| **Ours** (Corollary 4.6) | softmax | Entropy | ✓ | ✓ | ✓$_{\text{exp}}^{(5)}$ |
| **Ours** (Corollary 4.7) | $f_\alpha$-softargmax[2] | $\alpha$-Tsallis | ✓ | ✓ | ✓$_{\text{poly}}^{(4)}$ |

(1) We refer by *scalability* to policy gradient methods that do not use any form of preconditionning; (2) refers to the parameterization induced by using the $\alpha$-Tsallis divergence generator (see Table 2); (3) Explicit rates means an explicit dependency on all problem parameters and not on intractable quantities; (4) ✓$_{\text{poly}}$ indicates an explicit convergence rate with explicit polynomial dependency on all problem parameters; (5) ✓$_{\text{exp}}$ indicates an explicit convergence rate with exponential dependence on at least one parameter.

In discrete control scenarios, the default choice is the softmax parameterization, typically coupled with entropy regularization. While ubiquitous, several recent results have revealed fundamental limitations of softmax-based policy gradient methods (Mei et al., 2020a; Li et al., 2023). In particular, in the absence of regularization, the softmax parameterization can induce extremely flat regions in the optimization landscape, leading to an unavoidable exponential lower bound on the rate of convergence (Li et al., 2023). Although entropy regularization is sometimes introduced in an attempt to mitigate this issue, no polynomial convergence guarantees are currently known in this setting. Even with entropy regularization, the landscape remains flat (see Figure 1a for an illustration). These observations motivate treating the policy parameterization itself as a *design choice*: rather than varying the regularizer within softmax, we ask whether moving beyond softmax can fundamentally improve the conditioning of policy gradient methods.

In this paper, we follow the line of work of (Mei et al., 2020a; Liu et al., 2025) and propose a new flexible family of alternative parameterizations induced by divergence generators (denoted $f$ in the following), which we refer to as *$f$-softargmax parameterizations*. We regularize the objective with the corresponding $f$-divergence, and we refer to this as a *coupled* parameterization–regularization pair (i.e., the same generator $f$ induces both the parameterization and the regularizer). This viewpoint generalizes the classical softmax–entropy pairing, in which the policy is both induced and regularized by Shannon entropy. Similar constructions have recently shown theoretical and practical benefits for supervised learning (Blondel et al., 2020; Roulet et al., 2025), but remain largely unexplored in reinforcement learning. This takes policy gradient methods *beyond the softmax and its coupled entropy regularization*, yielding a better conditioned optimization landscape (see Figure 1b). In particular, we show that when these parameterizations are *coupled* with the regularizer induced by the corresponding $f$-divergence, policy gradient methods enjoy improved convergence rates. Remarkably, for the Tsallis divergence, this leads to convergence rates exponentially faster compared to the softmax–entropy pairing.

Formally, we study the $f$-regularized value function under the $f$-softargmax parameterization. We show that it satisfies a *non-uniform Łojasiewicz inequality* and a monotonicity property. This monotonicity allows us to restrict the optimization to regions that are easy to project onto and in which the Łojasiewicz coefficient is uniformly lower bounded, resulting in the *uniform Polyak-Łojasiewicz* inequality over the region of interest. Building on these observations, we establish *global last-iterate convergence guarantees* for stochastic policy gradient methods in the tabular setting, with *fully explicit constants*. To the best of our knowledge, these are the first guarantees of this type for policy gradient methods, even with entropy regularization and softmax parameterization, that do not rely on preconditioning or exponentially large batch sizes. For the KL-induced parameterization–regularization pair (softmax–entropy), the resulting uniform Polyak–Łojasiewicz constant is exponentially small in the problem parameters, recovering known exponential convergence rates (Ding et al., 2025; Labbi et al., 2026). In contrast, for Tsallis divergence generators, this constant scales only polynomially, reflecting a substantially better-conditioned optimization landscape. Addi-

tionally, our analysis shows that moving beyond the entropy-softmax pairing yields a better trade-off between regularization bias and sample complexity. In particular, Tsallis-type couplings yield *polynomial* last-iterate convergence guarantees even for the *unregularized* objective, improving upon the worst-case guarantees known for the standard softmax (see Table 1).

Overall, our contributions are threefold:

- We introduce $f$-softargmax policy parameterizations and study the regularity of the associated $f$-regularized value function as a function of the policy parameters. We show that it is smooth, satisfies a non-uniform Łojasiewicz inequality, and, by exploiting a monotonicity property, admits a uniform bound on a Polyak-Łojasiewicz constant on a region of interest that is easy to project onto.
- We prove global last-iterate convergence guarantees for stochastic tabular policy gradient, with fully explicit sample complexity bounds that apply to both regularized and unregularized objectives.
- We demonstrate that alternative couplings beyond entropy–softmax lead to improved sample complexity for an unregularized problem both theoretically and empirically. In particular, the Tsallis coupling yields polynomial dependencies on problem parameters, resulting in an exponential improvement over softmax, and provides additional flexibility for practical adaptation.

The paper is organized as follows. Section 2 introduces the necessary background. Section 3 presents the $f$-softargmax parameterization and the properties of the $f$-regularized value under this parameterization. Convergence rates for policy gradient are established in Section 4, and numerical experiments are reported in Section 5.

**Related work.** *(Policy gradient methods.)* Global convergence guarantees are known to hold for unregularized policy gradient methods with deterministic gradients, achieving sublinear rates with constant step-sizes (Mei et al., 2020b; Liu et al., 2024). However, the convergence rates of softmax-based policy gradient methods is exponential in the problem parameters (Mei et al., 2020a; Li et al., 2023). Two strategies were proposed to mitigate this issue: first, preconditioning, most notably through natural policy gradient methods (Kakade, 2001), which can alleviate ill-conditioning, but scales poorly to larger problems due to the nature of the updates. Second, log-barrier regularization (Zhang et al., 2021), which yields polynomial rates but has no last iterate convergence guarantees and is unstable in practice. In contrast, our approach avoids preconditioning altogether and therefore retains the scalability of standard policy gradient methods, while providing explicit polynomial convergence guarantees for the *last iterate*.

*(Alternative Parameterizations.)* Alternatives to softmax have been proposed and studied in optimization (Martins & Astudillo, 2016; Peters et al., 2019; Roulet et al., 2025). In RL, the study of alternative parameterizations is still in its early stages. The escort transform of Mei et al. (2020a) avoids exponential slowdowns in deterministic settings, but its guarantees rely on increasing step-sizes and do not extend to stochastic gradients. The Hadamard parameterization (Liu et al., 2025) yields local linear convergence in deterministic regimes, but without explicit constants. In this work, we propose a more flexible family of parameterizations that can adapt to various problems and provide explicit guarantees in the stochastic setting.

## 2 BACKGROUND

**Reinforcement learning.** Consider a discounted Markov decision process $\mathcal{M} = (\mathcal{S}, \mathcal{A}, \gamma, \mathsf{P}, \mathsf{r}, \rho)$ with finite state and action spaces $\mathcal{S}$ and $\mathcal{A}$, discount factor $\gamma \in (0, 1)$, transition kernel $\mathsf{P} \colon \mathcal{S} \times \mathcal{A} \to \mathcal{P}(\mathcal{S})$, bounded reward function $\mathsf{r} \colon \mathcal{S} \times \mathcal{A} \to [0, 1]$, and initial distribution $\rho$. The value of a policy $\pi \colon \mathcal{S} \to \mathcal{P}(\mathcal{A})$ is defined by

$$v_\pi(s) := \mathbb{E}_s^\pi \left[ \sum\nolimits_{t=0}^\infty \gamma^t \mathsf{r}(S_t, A_t) \right], \tag{1}$$

where $S_0 = s$, $A_t \sim \pi(\cdot|S_t)$, and $S_{t+1} \sim \mathsf{P}(\cdot|S_t, A_t)$. For $\rho \in \mathcal{P}(\mathcal{S})$, we define $v_\pi(\rho) := \sum_s \rho(s) v_\pi(s)$. For any $\pi$, we define a corresponding discounted occupancy measure

$$d_\rho^\pi(s) := \frac{1}{1 - \gamma} \mathbb{E}_s^\pi \left[ \sum_{t=0}^\infty \gamma^t \mathbb{1}_s(S_t) \right] .$$

**Parameterizations on the simplex.** Following Roulet et al. (2025), we study a family of parameterizations of the simplex based on divergence generators. For a given *generator* $f \colon (0, \infty) \to \mathbb{R}$

Table 2: Example of classical divergence generators $f$ included in our framework and their associated f-softargmax operators.

| Name | $\mathbf{f(u)}$ | f-softargmax$(x, \nu_{\text{ref}})[a]^{(1)}$ |
|---|---|---|
| KL | $u \log u - (u-1)$ | $\nu_{\text{ref}}(a) \exp(x(a))/(\sum_{b \in \mathcal{A}} \nu_{\text{ref}}(b) \exp(x(b)))$ |
| Tsallis $(0 < \alpha < 1)$ | $(u^{\alpha} - \alpha u + \alpha - 1)/\alpha(\alpha-1)$ | $\nu_{\text{ref}}(a) (1 + (\alpha-1)(x(a) - \mu_x^{\alpha}))^{1/(\alpha-1)}$ |
| Jensen-Shannon | $u \log(u) - (u+1)\log(\frac{u+1}{2})$ | $\nu_{\text{ref}}(a) \exp(2(x(a) - \mu_x))/(2 - \exp(2(x(a) - \mu_x)))$ |

(1) Here $\mu_x^{\alpha}$ and $\mu_x$ are normalization factors that ensures that the weights sum to 1.

strictly convex with $f(1) = 0$, and reference distribution $q$ with *full support*, we define

$$\text{f-softmax}(x, q) := \max_{\nu \in \mathcal{P}(\mathcal{A})} \left\{ \langle \nu, x \rangle - D^f(\nu \| q) \right\} ,$$

$$\text{f-softargmax}(x, q) := \arg\max_{\nu \in \mathcal{P}(\mathcal{A})} \left\{ \langle \nu, x \rangle - D^f(\nu \| q) \right\} ,$$

where $D^f(p \| q)$ is the $f$-divergence between $p$ and $q$ (see Csiszár, 1967, or Appendix A). Since $D^f(p \| q)$ is strictly convex in its first argument on the simplex, the output of the f-softargmax operator is well defined and unique, as it corresponds to the $\arg\max$ of a strictly concave function over a compact set. This construction recovers the classical softmax as a special case and yields a rich family of alternative parameterizations (see Table 2). Computing f-softargmax reduces to solving a one-dimensional root-finding problem, which can be done efficiently by dichotomy; see Roulet et al. (2025) and Lemma B.1 for details.

**$f$-regularized value functions.** Given a reference policy $\pi_{\text{ref}}$, temperature $\lambda \geq 0$, and a divergence generator $f$, the $f$-regularized value function of $\pi$ is defined by

$$v_{\pi}^f(s) := \mathbb{E}_s^{\pi}\left[ \sum_{t=0}^{\infty} \gamma^t (\mathsf{r}(S_t, A_t) - \lambda D^f(\pi(\cdot|S_t) \| \pi_{\text{ref}}(\cdot|S_t))) \right] ,$$

A key result (Geist et al., 2019) is that the optimal regularized value $v_{\star}^f(s) := \max_{\pi} v_{\pi}^f(s)$, together with optimal policy $\pi_{\star}^f$ admits a closed-form Bellman characterization:

$$v_{\star}^f(s) = \max_{\nu \in \mathcal{P}(\mathcal{A})} \{ \langle \nu, q_{\star}^f(s, \cdot) \rangle - \lambda D^f(\nu \| \pi_{\text{ref}}(\cdot|s)) \}, \tag{2}$$

$$\pi_{\star}^f(\cdot|s) = \arg\max_{\nu \in \mathcal{P}(\mathcal{A})} \{ \langle \nu, q_{\star}^f(s, \cdot) \rangle - \lambda D^f(\nu \| \pi_{\text{ref}}(\cdot|s)) \}, \tag{3}$$

where $q_{\star}^f(s, a) := \mathsf{r}(s, a) + \gamma \mathsf{P} v_{\star}^f(s, a)$.

## 3 Coupling Parameterization & Regularization

We introduce a new class of policy parameterizations for reinforcement learning, which we refer to as f-softargmax policies. Let $\pi_{\text{ref}}$ denote a full-support reference policy. For $\theta \in \mathbb{R}^{|\mathcal{S}||\mathcal{A}|}$, we define the f-softargmax policy by

$$\pi_{\theta}^f(\cdot|s) := \text{f-softargmax}(\theta(s, \cdot), \pi_{\text{ref}}(\cdot|s)), \quad \forall s \in \mathcal{S}. \tag{4}$$

This parameterization can be directly used within unregularized policy gradient methods (see Appendix G). However, in practice, unregularized methods tend to over-exploit and converge prematurely to suboptimal policies. This suggests that the choice of parameterization should be guided by the geometry of a suitably regularized objective, rather than considered in isolation.

To understand which regularization is naturally associated with the f-softargmax family, we examine the structure of the $f$-regularized problem (3). The optimal policy of this problem admits the following representation:

$$\pi_{\star}^f(\cdot|s) = \text{f-softargmax}\big(q_{\star}^f(s, \cdot)/\lambda, \ \pi_{\text{ref}}(\cdot|s)\big) .$$

In particular, if we choose logits $\theta_{\star}^f(s, \cdot) = q_{\star}^f(s, \cdot)/\lambda + b(s)$, where $b: \mathcal{S} \to \mathbb{R}$ is an arbitrary state-dependent baseline, then the f-softargmax mapping exactly recovers the optimal $f$-regularized policy, i.e., $\pi_{\theta_{\star}}^f = \pi_{\star}^f$.

This shows that the f-softargmax parameterization is not arbitrary: it is precisely matched to the geometry of the $f$-regularized problem. Under this parameterization, learning the policy is equivalent to learning the regularized optimal $Q$-function, and the associated $f$-divergence regularizer arises naturally from the variational characterization of the optimal policy.

To further formalize the benefits of such coupling, we now establish the smoothness, as well as a Polyak-Łojasiewicz inequality, of the $f$-regularized value with coupled parameterization $v_\theta^f := v_{\pi_\theta^f}^f$. We derive these properties under the following two assumptions on $f$ and $\pi_{\text{ref}}$.

**Assumption P($\underline{\pi_{\text{ref}}}$).** *There exists a number $\underline{\pi_{\text{ref}}} > 0$ such that $\min_{(s,a)} \pi_{\text{ref}}(a|s) > \underline{\pi_{\text{ref}}}$.*

**Assumption A$_f$($\underline{\pi_{\text{ref}}}$).** *The generator function $f$ satisfies:*

*(i) $f$ is bounded, strictly convex on $[0; 1/\underline{\pi_{\text{ref}}}]$, $f(1) = 0$, and $f$ is thrice differentiable on $(0, 1/\underline{\pi_{\text{ref}}})$;*
*(ii) $\lim_{u \downarrow 0+} f'(u) = -\infty$, and $\lim_{u \to 0} |f'(u)/f''(u)| < \infty$;*
*(iii) there exists $\omega_f \in [1, \infty)$, and $\kappa_f \in (0, \infty)$, such that for any $u \in [0; 1/\underline{\pi_{\text{ref}}}]$, we have $1/(uf''(u)) \leq \omega_f$, and $|f'''(u)/f''(u)^2| \leq \kappa_f$;*
*(iv) there exists $\iota_f \in (0, 1]$ such that $f''$ decreases on $[0; \iota_f]$ and for $u \in [\iota_f; 1/\underline{\pi_{\text{ref}}}]$, $f''(\iota_f) \geq f''(u)$.*

These conditions are met by a broad class of commonly used divergence generators, like the KL, Tsallis with $\alpha \leq 1$, and Jensen-Shannon (see Appendix F).

***Remark* 3.1** *Tsallis divergences with $\alpha > 1$ violate condition (ii): since $f'(0)$ is finite, the induced policies are sparse, leading to non-smooth parameterizations.*

Under **P($\underline{\pi_{\text{ref}}}$)** and **A$_f$($\underline{\pi_{\text{ref}}}$)**, we can define the weights $w_\theta^f(a|s)$ and the sum $W_\theta^f(s)$, defined as

$$w_\theta^f(a|s) := \frac{1}{W_\theta^f(s)} \frac{\pi_{\text{ref}}(a|s)}{f''(\pi_\theta^f(a|s)/\pi_{\text{ref}}(a|s))} \quad , \quad W_\theta^f(s) := \sum_{a \in \mathcal{A}} \frac{\pi_{\text{ref}}(a|s)}{f''(\pi_\theta^f(a|s)/\pi_{\text{ref}}(a|s))} \quad , \quad (5)$$

which will play a central role in our analysis. In the KL divergence case, we recover simple expressions $w_\theta^f(a|s) \equiv \pi_\theta^f(a|s)$ and $W_\theta^f(s) = 1$. Using the notations in (5), we can express the gradient of the regularized value.

**Lemma 3.2.** *Assume that, for some $\underline{\pi_{\text{ref}}} > 0$, $f$ and $\pi_{\text{ref}}$ satisfy $A_f(\underline{\pi_{\text{ref}}})$ and $P(\underline{\pi_{\text{ref}}})$. For any $s \in \mathcal{S}$, we have*

$$\frac{\partial v_\theta^f(\rho)}{\partial \theta(s, \cdot)} = (1 - \gamma)^{-1} W_\theta^f(s) d_\rho^\theta(s) H(w_\theta^f(\cdot|s)) \left[ q_\theta^f(s, \cdot) - \lambda \theta(s, \cdot) \right],$$

*where for any vector $u \in \mathbb{R}^{|\mathcal{A}|}$, $H(u) := \text{diag}(u) - uu^\top$, $q^f := r + \gamma P v_\theta^f$, and $d_\rho^\theta(s) := d_\rho^{\pi_\theta^f}(s)$.*

Next, we introduce three quantities that that arise naturally in the expression of the Hessian of $v_\theta^f(\rho)$.

$$y_f := \max_{(s,\nu) \in \mathcal{S} \times \mathcal{P}(\mathcal{A})} \sum_{a \in \mathcal{A}} \pi_{\text{ref}}(a|s) \frac{|f'(\nu(a)/\pi_{\text{ref}}(a|s))|}{f''(\nu(a)/\pi_{\text{ref}}(a|s))}, \quad d_f := \max_{(s,\nu) \in \mathcal{S} \times \mathcal{P}(\mathcal{A})} D^f(\nu \| \pi_{\text{ref}}(\cdot|s)) \quad , \quad (6)$$

$$\zeta_f := \min_{(s,\nu) \in \mathcal{S} \times \mathcal{P}(\mathcal{A})} \sum_{a \in \mathcal{A}} \frac{\pi_{\text{ref}}(a|s)}{f''(\nu(a)/\pi_{\text{ref}}(a|s))} > 0 \quad . \quad (7)$$

Since they are bounded under **A$_f$($\underline{\pi_{\text{ref}}}$)** and **P($\underline{\pi_{\text{ref}}}$)**, we can establish the smoothness of $v_\theta^f(\rho)$.

**Theorem 3.3.** *Assume $A_f(\underline{\pi_{\text{ref}}})$ and $P(\underline{\pi_{\text{ref}}})$ for some $\underline{\pi_{\text{ref}}} > 0$. For any $\theta \in \mathbb{R}^{|\mathcal{S}||\mathcal{A}|}$, $\|\nabla^2 v_\theta^f(\rho)\|_2 \leq L_f$ with*

$$L_f := \mathcal{O}\left( \frac{\omega_f^2 + \omega_f \kappa_f + \lambda \cdot (\omega_f^2 d_f + \omega_f(\kappa_f d_f + y_f) + \omega_f + 2\kappa_f y_f)}{(1 - \gamma)^3} \right) \quad .$$

We refer to Appendix B.4 for a proof and a complete expression of $L_f$. We now introduce the classical exploration assumption (Mei et al., 2020a;b; Agarwal et al., 2021).

**Assumption A$_\rho$.** *The coefficient $\rho_{\min} := \min_{s \in \mathcal{S}} \rho(s)$ of the initial distribution $\rho$ satisfies $\rho_{\min} > 0$.*

Next, we derive a Non-Uniform Łojasiewicz inequality.

**Theorem 3.4.** *Assume that, for some $\pi_{\text{ref}} > 0$, $\mathbf{A}_f(\pi_{\text{ref}})$ and $\mathbf{P}(\pi_{\text{ref}})$ hold. Assume in addition that the initial distribution $\rho$ satisfies $\mathbf{A}_\rho$. Then, it holds that*

$$\|\nabla_\theta v_\theta^f(\rho)\|_2^2 \geq \mu_f(\theta)(v_\star^f(\rho) - v_\theta^f(\rho)) \ ,$$

*with* $\mu_f(\theta) := \lambda(1-\gamma)\rho_{\min}^2(\zeta_f/\omega_f)^2 \min_{s,a} \mathrm{w}_\theta^f(a|s)^2$.

We prove this theorem in Appendix C. To highlight the main steps of the proof, we give a sketch of the proof in the bandits setting, where the state space is a singleton.

***Sketch of the proof in the bandits case.*** The proof consists of two steps: (1) we bound the sub-optimality gap by the distance between the logit and the rescaled reward; (2) then link it to the gradient of the function.

*Step 1:* By (2), the optimal regularized value is equal to $v_\star^f = \lambda$ f-softmax$(\mathsf{r}/\lambda, \pi_{\text{ref}})$. Next, since f-softmax$(\theta, \pi_{\text{ref}}) = \langle \pi_\theta^f, \theta\rangle - \mathrm{D}^f(\pi_\theta^f \| \pi_{\text{ref}})$, and $v_\theta^f = \langle \pi_\theta^f, \mathsf{r}\rangle - \lambda\, \mathrm{D}^f(\pi_\theta^f \| \pi_{\text{ref}})$, the suboptimality gap rewrite as

$$v_\star^f - v_\theta^f = \lambda \cdot \text{f-softmax}(\frac{\mathsf{r}}{\lambda}, \pi_{\text{ref}}) - \langle \pi_\theta^f, \mathsf{r}\rangle + \lambda\, \mathrm{D}^f(\pi_\theta^f \| \pi_{\text{ref}})$$

$$= \lambda\big[\,\text{f-softmax}(\tfrac{\mathsf{r}}{\lambda}, \pi_{\text{ref}}) - \text{f-softmax}(\theta, \pi_{\text{ref}}) - \langle \pi_\theta^f, \tfrac{\mathsf{r}}{\lambda} - \theta\rangle\big].$$

Combining $\pi_\theta^f = \nabla$ f-softmax$(\theta, \pi_{\text{ref}})$, and that for $a \in \mathbb{R}$, f-softmax$(\theta + \alpha 1_{|\mathcal{A}|}, \pi_{\text{ref}}) =$ f-softmax$(\theta, \pi_{\text{ref}}) + \alpha$ yields

$$v_\star^f - v_\theta^f = \lambda\big[\text{f-softmax}(\tfrac{\mathsf{r}}{\lambda}, \pi_{\text{ref}}) - \text{f-softmax}(\theta + K_\theta^f 1_{|\mathcal{A}|}, \pi_{\text{ref}})$$
$$- \langle \nabla\, \text{f-softmax}(\theta + K_\theta^f 1_{|\mathcal{A}|}, \pi_{\text{ref}}), \mathsf{r}/\lambda - \theta - K_\theta^f 1_{|\mathcal{A}|}\rangle\big],$$

where we have defined $K_\theta^f = \langle \mathsf{r}/\lambda - \theta, 1_{|\mathcal{A}|}\rangle/|\mathcal{A}|$. Defining $\zeta_\theta^f = \mathsf{r}/\lambda - \theta - K_\theta^f 1_{|\mathcal{A}|}$ and using a second-order Taylor expansion of the function f-softmax$(\cdot)$ between, we obtain

$$v_\star^f - v_\theta^f = \tfrac{\lambda}{2}(\zeta_\theta^f)^\top \nabla^2\, \text{f-softmax}_f(\xi)\zeta_\theta^f, \tag{8}$$

for some $\xi$ on the segment joining $\theta + K_\theta^f 1_{|\mathcal{A}|}$ and $\mathsf{r}/\lambda$. Next, by Lemma B.5, it holds that $\big\|\nabla^2\, \text{f-softmax}(\xi)\big\|_2 \leq 2\omega_f$, which implies $v_\star^f - v_\theta^f \leq \lambda\omega_f\|\zeta_\theta^f\|_2^2$.

*Step 2:* Using Lemma 3.2, we have $\nabla v_\theta^f = \mathrm{W}_\theta^f H(\mathrm{w}_\theta^f)[\mathsf{r} - \lambda\theta]$. Next, applying Lemma 23 of Mei et al. (2020b) (see Lemma I.4) gives

$$\|\nabla v_\theta^f\|_2^2 \geq \mathrm{W}_\theta^f \min_{a \in \mathcal{A}} \mathrm{w}_\theta^f \|\zeta_\theta^f\|_2^2 \geq \zeta_f \min_{a \in \mathcal{A}} \mathrm{w}_\theta^f \|\zeta_\theta^f\|_2^2 \ .$$

where by Lemma B.6, we have $\mathrm{W}_\theta^f \geq \zeta_f$. Combining the two bounds concludes the proof $\qquad\square$

For softmax parameterization coupled with entropy regularization, we retrieve a property outlined by Mei et al. (2020b). However, their proof is highly specific to the entropy-softmax pairing case and cannot be extended to general $f$-divergences, because it relies in an essential way on the logarithm's special properties. Indeed, their proof require rewriting the soft sub-optimality gap $v_\star^{\text{KL}}(\rho) - v_\theta^{\text{KL}}(\rho)$ as $\frac{\lambda}{1-\gamma}\sum_{s'\in\mathcal{S}} d_\rho^\theta(s')\, \mathrm{D}^{\text{KL}}(\pi_\theta^{\text{KL}}(\cdot|s')\|\pi_\star^{\text{KL}}(\cdot|s))$ which is not possible for a general $f$. Our proof is more natural, as it simply relies on Taylor expansion to obtain the inequality (8).

**From Non-Uniform Łojasiewicz to Polyak-Łojasiewicz.** To obtain a Polyak-Łojasiewicz bound, we need to bound the coefficient $\mu_f(\theta)$ uniformly. To this end, we restrict the optimization to a smaller subspace, eliminating policies for which the regularization is too large. Given a policy $\pi$ and $0 < \tau < \underline{\pi_{\text{ref}}}/2$, we define the projection-like operator $\mathcal{U}_\tau$, for every $(s, a) \in \mathcal{S} \times \mathcal{A}$, as

$$\mathcal{U}_\tau(\pi)(a|s) = \begin{cases} \pi_{\text{ref}}(a|s)\tau, & \text{if } \pi(a|s) \leq \pi_{\text{ref}}(a|s)\tau/2, \\ \pi(a|s) - b_\pi(s), & \text{if } a = a_\pi^{\max}(s), \\ \pi(a|s), & \text{otherwise,} \end{cases}$$

where we define $b_\pi(s) = \sum_{b\in\mathcal{A}_\tau^\pi(s)} \pi_{\text{ref}}(b|s)\tau - \pi(b|s)$, $a_\pi^{\max}(s) = \arg\max_{a\in\mathcal{A}}\{\pi(a|s)/\pi_{\text{ref}}(a|s)\}$, where ties in the $\arg\max$ are resolved arbitrary, and

$$\mathcal{A}_\tau^\pi(s) := \{a \in \mathcal{A}, \pi(a|s)/\pi_{\text{ref}}(a|s) \leq \tau/2\} \ .$$

---

**Algorithm 1** $f$-SoftArgmax Policy Gradient

---

1: **Initialization:** Learning rate $\eta > 0$, initial parameter $\theta_0$, divergence generator $f$, batch size $B$.
2: **for** $t = 0$ to $T - 1$ **do**
3:   Collect $Z_t := (S^b_{t,0:H-1}, A^b_{t,0:H-1})^{B-1}_{b=0}$ using $\pi^f_{\theta_t}$
4:   Compute the gradient $\mathrm{g}^f_{Z_t}(\theta_t)$ using (11)
5:   Update $\theta_{t+1} = \mathcal{T}_{\tau_\lambda}(\theta_t + \eta \mathrm{g}^f_{Z_t}(\theta_t))$
6: Return $\theta_T$

---

This operator prevents policies from becoming "too deterministic": if the probability of any action gets too close to zero, it is increased above a threshold that depends on $\tau$ and $\pi_{\mathrm{ref}}$. For a proper choice of $\tau$, applying this operator on a policy returns a policy with a higher regularized value.

**Theorem 3.5.** *Assume that, for some $\underline{\pi_{\mathrm{ref}}} > 0$, $f$ and $\pi_{\mathrm{ref}}$ satisfy $\boldsymbol{A}_f(\underline{\pi_{\mathrm{ref}}})$ and $\boldsymbol{P}(\underline{\pi_{\mathrm{ref}}})$, and that $\rho$ satisfies $\boldsymbol{A}_\rho$. Let*

$$\tau_\lambda = \min([f']^{-1}(-\tfrac{16+8\gamma\lambda\mathrm{d}_f}{\lambda(1-\gamma)^2\rho_{\min}}), [f']^{-1}(-4|f'(\tfrac{1}{2})|), \tfrac{\underline{\pi_{\mathrm{ref}}}}{2}) \ .$$

*Then, for any policy $\pi$ and for $\widetilde{\pi} = \mathcal{U}_{\tau_\lambda}(\pi)$, it holds that $v^f_{\widetilde{\pi}}(\rho) \geq v^f_\pi(\rho)$ and that $\widetilde{\pi}(a|s) \geq \underline{\pi_{\mathrm{ref}}}\tau_\lambda$.*

Since $\mathcal{U}_\tau$ operates in the space of policies, we lift it to the parameter space by defining an operator $\mathcal{T}_\tau$ such that for $\theta \in \mathbb{R}^{|\mathcal{S}||\mathcal{A}|}$, we have $\pi^f_{\mathcal{T}_\tau\theta} = \mathcal{U}_\tau\pi^f_\theta$ (see Appendix D for an explicit construction). Finally, we show that with the choice of threshold from Theorem 3.5, we can give a uniform lower bound of $\mu_f$ on the set of restricted logits. This shows how the non-uniform Łojasiewicz inequality is upgraded to a Polyak–Łojasiewicz condition: *it suffices to restrict the search to parameters that encode non-degenerate policies, since such policies are provably suboptimal.*

**Corollary 3.6.** *Assume that, for some $\underline{\pi_{\mathrm{ref}}} > 0$, $f$ and $\pi_{\mathrm{ref}}$ satisfy $\boldsymbol{A}_f(\underline{\pi_{\mathrm{ref}}})$ and $\boldsymbol{P}(\underline{\pi_{\mathrm{ref}}})$ and that $\rho$ satisfies $\boldsymbol{A}_\rho$. If $\lambda \lesssim \frac{1}{(1-\gamma)^2\rho_{\min}}\min(\frac{1}{|f'(\iota_f)|}, \frac{1}{|f'(\frac{1}{2})|}, \frac{1}{|f'(\frac{1}{2}\underline{\pi_{\mathrm{ref}}})|})$ where $\tau_\lambda$ is defined in Theorem 3.5. Under this condition, it holds that $\inf_{\theta \in \mathbb{R}^d} \mu_f(\mathcal{T}_{\tau_\lambda}\theta) \geq \underline{\mu}_f$, where*

$$\underline{\mu}_f := \lambda(1-\gamma)\rho^2_{\min}\zeta^2_f\underline{\pi_{\mathrm{ref}}}^2(f^\star)''\big(\tfrac{-16-8\gamma\lambda\mathrm{d}_f}{\lambda(1-\gamma)^2\rho_{\min}}\big)^2/\omega^2_f \ ,$$

*where $f^\star$ is a convex conjugate of $f$.*

## 4 CONVERGENCE ANALYSIS OF $\boldsymbol{f}$−PG

In this section, we aim to optimize the $f$-regularized value under f-softargmax parameterization.

$$\max_{\theta \in \Theta}\big\{J^f(\theta) := v^f_{\pi^f_\theta}(\rho)\big\} \ . \tag{9}$$

**The $\boldsymbol{f}$−PG algorithm.** We introduce $f$-PG (Algorithm 1), an $f$-SoftArgmax policy gradient method with coupled regularization. At each iteration of $f$-PG, the agent samples a batch of independent truncated trajectories of length $H$ from $\nu(\pi;\cdot)$ defined for a single truncated trajectory $z = (s_h, a_h)^{H-1}_{h=0} \in (\mathcal{S} \times \mathcal{A})^H$ by $\nu(\pi;z) := \rho(s_0)\pi(a_0|s_0)\prod^{H-1}_{h=0} \mathrm{P}(s_h|s_{h-1}, a_{h-1})\pi(a_h|s_h)$. Then, the agent performs the update

$$\theta_{t+1} = \mathcal{T}_{\tau_\lambda}\big(\theta_t + \eta \cdot \mathrm{g}^f_{Z_t}(\theta_t)\big) \ , \quad \text{for } t \geq 0 \ , \tag{10}$$

where $\eta > 0$ is a learning rate, $\mathcal{T}_{\tau_\lambda} : \mathbb{R}^{|\mathcal{S}||\mathcal{A}|} \to \mathbb{R}^{|\mathcal{S}||\mathcal{A}|}$ is the projection-like operator defined in Section 3, and $\mathrm{g}^f_{Z_t}(\theta_t)$ is a REINFORCE-like estimator (Williams, 1992) of $\nabla v^f_{\theta_t}(\rho)$ that uses a batch of $B$ independent trajectories $Z_t \sim [\nu(\theta_t)]^{\otimes B}$. For a batch of trajectories $z = (s^b_{0:H-1}, a^b_{0:H-1})^{B-1}_{b=0}$, this estimator is defined

$$\mathrm{g}^f_z(\theta) := \frac{1}{B}\sum^{B-1}_{b=0}\sum^{H-1}_{h=0}\Big\{\sum^h_{\ell=0}\frac{\partial \log \pi^f_\theta(a^b_\ell|s^b_\ell)}{\partial\theta}\gamma^h\mathrm{r}^f_\theta(s^b_h, a^b_h) - \lambda\gamma^h\mathrm{F}^f_\theta(s^b_h)\Big\}, \tag{11}$$

where $\mathrm{r}^f_\theta(s^b_h, a^b_h) = \mathrm{r}(s^b_h, a^b_h) - \lambda\mathrm{D}^f(\pi^f_\theta(\cdot|s^b_h)\|\pi_{\mathrm{ref}}(\cdot|s^b_h))$, and $\mathrm{F}^f_\theta(s) \in \mathbb{R}^{|\mathcal{S}||\mathcal{A}|}$ is defined by

$$[\mathrm{F}^f_\theta(s)]_{(s',b)} := \mathbb{1}_{s'}(s)\,\mathrm{W}^f_\theta(s)\,\mathrm{w}^f_\theta(b|s)\Big(f'(\tfrac{\pi^f_\theta(b|s)}{\pi_{\mathrm{ref}}(b|s)}) - \sum_{a\in\mathcal{A}}\mathrm{w}^f_\theta(a|s)f'(\tfrac{\pi^f_\theta(a|s)}{\pi_{\mathrm{ref}}(a|s)})\Big), \forall(s',b) \in \mathcal{S}\times\mathcal{A}.$$

**Remark 4.1** (Connection with (Lazy) Mirror Descent.) *We stress that $f$-PG is fundamentally different from mirror descent. Let $\Phi(\pi) = \sum_{s \in \mathcal{S}} \mathrm{D}^f(\pi(\cdot|s) \| \pi_{\mathrm{ref}}(\cdot|s))$, mirror descent iterations are*

$$\nabla \Phi(\widetilde{\pi}_{t+1}) = \nabla \Phi(\widetilde{\pi}_t) + \eta \nabla_\pi v_\pi^f(\rho)|_{\pi=\pi_t}, \quad \pi_{t+1} = \arg\min_\pi \{\Phi(\pi) - \langle \nabla \Phi(\widetilde{\pi}_{t+1}), \pi - \widetilde{\pi}_{t+1} \rangle\}. \quad (12)$$

*where $\pi \in \mathcal{P}(\mathcal{A})^{|\mathcal{S}|}$ is a policy. Denoting $\theta_t = \nabla \Phi(\widetilde{\pi}_t)$, one obtains updates that resemble (10) (without $\mathcal{T}_{\tau_\lambda}$), with one key difference: the gradient in (12) is taken with respect to the policy $\pi$ whereas in (10) it is computed w.r.t the "dual" parameter $\theta$ (in the mirror descent terminology). Moreover, the update (12) can be expressed as, by the chain rule, $\theta_{t+1} = \theta_t + \eta [\frac{\partial \pi_\theta}{\partial \theta}|_{\theta=\theta_t}]^{-1} \nabla_\theta J^f(\theta_t)$, which have an additional preconditioning term given by the inverse of the policy Jacobian. See Appendix H for more details.*

Next, we bound the bias and variance of the gradient estimator (a proof is provided in Appendix E.1).

**Lemma 4.2.** *Assume that, for some $\underline{\pi}_{\mathrm{ref}} > 0$, $f$ satisfy $A_f(\underline{\pi}_{\mathrm{ref}})$. For any $\theta \in \mathbb{R}^{|\mathcal{S}||\mathcal{A}|}$, we have*

$$\|\mathrm{g}^f(\theta) - \tfrac{\partial v_\theta^f(\rho)}{\partial \theta}\|_2 \leq \beta_f, \quad \mathbb{E}_Z\big[\|\mathrm{g}^f(\theta) - \mathrm{g}_Z^f(\theta)\|_2^2\big] \leq \tfrac{\sigma_f^2}{B},$$

*where $Z \sim [\nu(\theta)]^{\otimes B}$, and where*

$$\sigma_f^2 := \tfrac{12}{(1-\gamma)^4}\left[\omega_f^3 + \lambda^2 \gamma^2 \omega_f^3 \mathrm{d}_f^2 + \lambda^2 (1-\gamma)^2 \omega_f^2 \mathrm{y}_f^2\right], \quad \beta_f := \tfrac{2\gamma^H(H+1)}{(1-\gamma)^2}\omega_f\left[2 + 2\lambda \mathrm{d}_f + \lambda(1-\gamma)\mathrm{y}_f\right].$$

**Convergence analysis.** We now present our main result for this section, which gives a convergence rate for $f$-PG *with explicit constants* for the regularized problem. This result is based on the regularity properties of the regularized value function, which we developed in Section 3.

**Theorem 4.3.** *Assume that, for some $\underline{\pi}_{\mathrm{ref}} > 0$, $f$ and $\pi_{\mathrm{ref}}$ satisfy $A_f(\underline{\pi}_{\mathrm{ref}})$ and $P(\underline{\pi}_{\mathrm{ref}})$, and that $\rho$ satisfies $A_\rho$. Fix $\eta \leq 1/(2L_f)$, and $\lambda$ and $\tau_\lambda$ as in Corollary 3.6. Then, for any $t \geq 0$, the iterates of $f$-PG satisfy*

$$\mathbb{E}\left[\Delta_t\right] \leq \left(1 - \tfrac{\eta \underline{\mu}_f}{4}\right)^t \Delta_0 + \tfrac{6\eta \sigma_f^2}{B\underline{\mu}_f} + \tfrac{6\beta_f^2}{\underline{\mu}_f},$$

*with $\Delta_t = v_\star^f(\rho) - v_{\theta_t}^f(\rho)$, and $\sigma_f^2$ and $\beta_f^2$ from Corollary 3.6 and Lemma 4.2 respectively.*

We provide a proof of this result in Appendix E.2. A crucial feature of this theorem is that it is *explicit*, as all the terms that appear can be expressed using problem-dependent constants. This allows us to derive the following sample complexity result for optimizing the regularized value.

**Corollary 4.4.** *Let $\epsilon > 0$ and set $H \lesssim (1-\gamma)^2 \cdot \log(1/\underline{\mu}_f)$, and $\eta \lesssim \min(L_f^{-1}, \epsilon B \underline{\mu}_f \cdot \sigma_f^{-2})$. Under assumptions of Theorem 4.3, the final iterate of $f$-PG satisfies $\mathbb{E}[v_\star^f(\rho) - v_{\theta_T}^f(\rho)] \leq \epsilon$ with*

$$T \lesssim \max\left(\tfrac{L_f}{\underline{\mu}_f}, \tfrac{\sigma_f^2}{\epsilon B \underline{\mu}_f^2}\right) \log\left(\tfrac{v_\star^f(\rho) - v_{\theta_0}^f(\rho)}{\epsilon}\right).$$

Importantly, Corollary 4.4 shows that $f$-PG achieves convergence rates comparable to stochastic gradient ascent in the strongly convex regime: $O(\kappa \log(1/\epsilon))$ in the low-variance setting, with condition number $\kappa > 0$, and $O((1/\epsilon) \log(1/\epsilon))$ in general.

**Convergence for unregularized objective.** A natural question is then how to compare different choices of regularizers, since each method optimizes a distinct regularized objective. By appropriately tuning the temperature $\lambda$, we recover the final sample complexity bound for the *unregularized problem*. We give a precise statement and proof of this result in Corollary E.10.

**Corollary 4.5.** *Let $0 < \epsilon < (1-\gamma)^{-3}\rho_{\min}^{-1}\min(|f'(\iota_f)|^{-1}, |f'(\tfrac{1}{2})|^{-1}, |f'(\tfrac{\pi_{\mathrm{ref}}}{2})|^{-1})$. Define $\mathrm{c}_f = \min(1/\mathrm{d}_f, 1/\mathrm{y}_f, 1)$, and set $\lambda = (1-\gamma)\epsilon/4 \cdot \mathrm{c}_f$. Under assumptions of Theorem 4.3, the final iterate of $f$-PG satisfies $\mathbb{E}[v_\star(\rho) - v_{\theta_T}(\rho)] \leq \epsilon$ with*

$$T \lesssim \tfrac{(f^\star)''(\tfrac{-1}{\epsilon \mathrm{c}_f (1-\gamma)^3 \rho_{\min}})^{-4}}{\epsilon^3 B (1-\gamma)^8 \rho_{\min}^4 \underline{\pi}_{\mathrm{ref}}^4} \log\left(\tfrac{6(v_\star^f(\rho) - v_{\theta_0}^f(\rho))}{\epsilon}\right),$$

*$H \lesssim (1-\gamma)^{-2} \cdot \log(1/\underline{\mu}_f)$, and $\eta \lesssim a \wedge b$ with $a = (1-\gamma)^3$ and $b = \epsilon^2 (f^\star)''(\tfrac{-1}{\epsilon \mathrm{c}_f (1-\gamma)^6 \rho_{\min}})^2 (1-\gamma)^3 B \rho_{\min}^2 \underline{\pi}_{\mathrm{ref}}^2$, where $f^\star$ is a convex conjugate of $f$.*

This corollary shows that the convergence rate to the unregularized optimum is primarily controlled by the asymptotic behaviour of $(f^\star)''$. As the target precision $\varepsilon \to 0$, divergences for which $(f^\star)''$ grows faster yield better conditioning, which in turn results in faster convergence.

**Sample complexity for specific choices of $f$.** We now provide a more complete interpretation of these results by stating sample complexity bounds for specific choices of $f$.

**Corollary 4.6** (Complexity for Softmax-Entropy). *Let $f$ be the Kullback-Leibler divergence generator. Let $\epsilon > 0$. Under the choice of $\eta, \lambda, H, \tau_\lambda$, and $T$ of Corollary 4.5, the final iterate of $f$-PG achieves $\mathbb{E}[v_\star(\rho) - v_{\theta_T}(\rho)] \le \epsilon$ in $TBH \lesssim \frac{|\log(\pi_{\mathrm{ref}})|^3}{\epsilon^4(1-\gamma)^{12}\rho_{\min}^5 \underline{\pi_{\mathrm{ref}}}^4} \exp(\frac{|\log(\pi_{\mathrm{ref}})|}{\epsilon(1-\gamma)^3\rho_{\min}})$ samples.*

This corollary shows that the number of samples required by the softmax policy gradient method is exponential in $1/(1-\gamma)$. This is in line with recent work on vanilla softmax policy gradient, which demonstrated that the number of steps is at least exponential in $1/(1-\gamma)$ (Li et al., 2023).

**Corollary 4.7** (Complexity for $\alpha$-Tsallis SoftArgmax with $\alpha$-Tsallis regularization). *Let $f$ be the $\alpha$-Csiszár–Cressie–Read divergence generator for $\alpha \in (0,1)$ (see Table 2 for its expression). Let $\epsilon > 0$. Under the choice of $\eta, \lambda, H, \tau_\lambda$, and $T$ of Corollary 4.5, the last iterate of $f$-PG achieves $\mathbb{E}[v_\star(\rho) - v_{\theta_T}(\rho)] \le \epsilon$ in a number of samples*

$$TBH \lesssim \frac{|\log(\pi_{\mathrm{ref}})|^3}{\epsilon^4\alpha^6(1-\gamma)^{12}\rho_{\min}^5 \underline{\pi_{\mathrm{ref}}}^{4+7(1-\alpha)}} \left(1 + \frac{(1-\alpha)|\log(\pi_{\mathrm{ref}})|}{\epsilon\alpha^2(1-\gamma)^3\rho_{\min}}\right)^{4+\frac{4}{1-\alpha}}.$$

We give detailed versions and prove these corollaries in Appendix F. These corollaries show that Tsallis SoftArgmax parameterization with coupled regularisation allows for faster learning, reducing the dependency on $(1-\gamma)^{-1}$ from exponential in Corollary 4.6 to polynomial in Corollary 4.7. Next, we approximate the choice of $\alpha$ that achieves the fastest convergence (according to our bounds).

**Corollary 4.8.** *Assume the conditions of Corollary 4.7 hold. The value of $\alpha$ that minimizes the sample complexity in Corollary 4.7 is given by $\alpha^\star(\epsilon) = 11/(2\log(1/\epsilon)) + o(1/\log(1/\epsilon))$. Moreover, for $\epsilon$ sufficiently small, choosing $\alpha = \alpha^\star(\epsilon)$ yields a sample complexity $\epsilon^{-12}\mathrm{poly}(1/(1-\gamma), 1/\rho_{\min}, 1/\underline{\pi_{\mathrm{ref}}})$ up to logarithmic factors.*

These results show that the best choice of $\alpha$ is not $\alpha = 0$ nor $\alpha = 1$, but depends on the desired precision level. This corroborates results from the bandit literature (Zimmert & Seldin, 2021), and gives strong evidence that Tsallis-SoftArgmax with coupled regularization has the potential to accelerate RL algorithms. It also highlights the strength of our framework: one can choose, among multiple parameterizations, the one that is best suited for the problem at hand.

## 5 EXPERIMENTS

In this section, we demonstrate the generalizability of our framework by showing that our class of parameterizations, with its coupled regularization, can be readily integrated into modern on-policy reinforcement learning algorithms. For this purpose, we introduce and evaluate $\alpha$-`Tsallis` `PPO`, a simple yet principled extension of Proximal Policy Optimization (PPO; Schulman et al. 2017)[1]. Our approach is obtained by replacing both the policy parameterization and the entropy regularization in PPO with their Tsallis counterparts (see Appendix J for full experimental details). We compare the performance of $\alpha$-`Tsallis` `PPO` against the standard `PPO` baseline (Schulman et al., 2017) on two families of environments that we describe below. Our code is available on GitHub: `https://github.com/Labbi-Safwan/f-regularised-policy-gradient`.

**Noisy CartPole** (Osband et al., 2020). This environment is a variant of the classic `CartPole` control task in which additive noise is injected into the reward signal. The underlying dynamics is unchanged: at each time step, the agent applies a left or right force to a cart in order to keep an inverted pendulum balanced, receiving a base reward of $+1$ for each step the pole remains upright, and the episode terminates when stability is lost or after a time limit. However, the reward returned by the environment is perturbed as $\tilde{r}_t = r_t + \sigma \xi_t$, $\xi_t \sim \mathcal{N}(0,1)$, where $\sigma > 0$ controls the noise level. This preserves the dynamics and optimal policy, but increases the variance of observed returns.

---

[1]Additional experiments on the exact $f$-PG algorithm are provided in Appendix J.

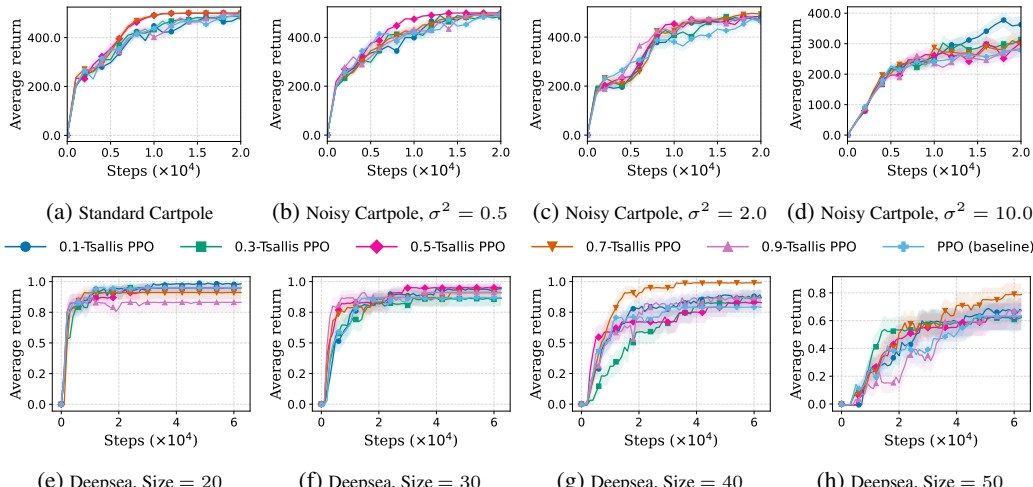

Figure 2: Learning curves for `Noisy CartPole` (top row) and `DeepSea` (bottom row) under different choices of the Tsallis parameter $\alpha$. For `Noisy CartPole`, we report the standard *unnoised* `CartPole` environment (a) and reward–noisy variants with increasing noise levels (b–d). For `DeepSea`, we consider grid sizes $L \in \{20, 30, 40, 50\}$ (e–h). Each curve corresponds to the best temperature and step-size for a given $\alpha$, and shaded regions indicate $\pm$ one standard error over 25 seeds. On `Noisy CartPole`, values $\alpha < 1$ consistently improve performance over the PPO baseline in the standard and low-noise settings, with the gap increasing as the reward noise grows. On `DeepSea`, the improvement over the PPO baseline becomes more pronounced with increasing $L$, where $\alpha = 0.7$ achieves the highest returns and the fastest learning.

**DeepSea ([Osband et al., 2019](#)).** `DeepSea` is an RL environment designed to study deep exploration under sparse rewards. The environment is a directed grid of size $L \times L$. The agent starts in the top-left corner $(0, 0)$ and, at each step, moves downward while choosing between two actions that shift agent's position either left or right. Thus, each episode lasts exactly $L$ steps and corresponds to selecting a binary action sequence of length $L$, which defines a unique path through the grid. Only a single trajectory, the one that selects the hidden *correct* (right) action at every depth reaches the rewarding terminal state at $(L-1, L-1)$. However, selecting the right action is not free: every time the agent moves right, it incurs a small movement cost $0.01/L$.

**Problem-adaptive couplings yield better performance.** Figure 2 illustrates that no single choice of $\alpha$ is uniformly optimal, and that different tasks favor different couplings of parameterization and regularization. On `Noisy CartPole`, the standard setting (Figure 2a) and the mildly noisy variant (Figure 2b) show a small but systematic advantage for $\alpha < 1$ over the PPO baseline, which becomes more pronounced as the reward noise increases (Figures 2c and 2d). By contrast, on `DeepSea`, where performance depends on discovering a single sparse-reward trajectory, the hardest instances (Figures 2g and 2h) favor an intermediate value $\alpha = 0.7$. These observations suggest that highly noisy environments and deep exploration problems may benefit from different regions of the Tsallis family, supporting the need for a tunable parameterization–regularization pair.

## 6 CONCLUSION

We proposed a new class of policy parameterizations based on operators induced by $f$-divergences. Equipped with a matching $f$-divergence regularizer, this framework generalizes the classical softmax–entropy pairing and allows flexible alternative parameterizations. Using Tsallis divergence instead of Shannon entropy, we showed that the resulting algorithm yields polynomial, rather than exponential, convergence guarantees for the unregularized RL problem. Empirically, this choice leads to improved performance in exploration-heavy and noisy environments. An important direction for future work is to extend these guarantees for adversarial MDPs, where Tsallis regularization has already proven effective in the bandit setting ([Zimmert & Seldin, 2021](#)).

## ACKNOWLEDGEMENTS

We would like to thank Vincent Roulet for a fruitful discussion on Mirror Descent and for providing the implementation of f-softargmax operators in JAX. The work of S. Labbi, and P. Mangold has been supported by Technology Innovation Institute (TII), project Fed2Learn. The work of D.Tiapkin has been supported by the Paris Île-de-France Région in the framework of DIM AI4IDF. The work of E. Moulines has been partly funded by the European Union (ERC-2022-SYG-OCEAN-101071601). Views and opinions expressed are however those of the author(s) only and do not necessarily reflect those of the European Union or the European Research Council Executive Agency. Neither the European Union nor the granting authority can be held responsible for them.

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

# Appendix

## Table of Contents

## A   NOTATIONS

$f$-**Divergence.**   Let $f : (0, \infty) \to \mathbb{R}$ be a strictly convex *generator* with $f(1) = 0$. Its *adjoint* (or reverse generator) is $f_\dagger(u) := u \, f(1/u)$, $u > 0$, which is convex, strictly convex if $f$ is, and satisfies $f_\dagger(1) = 0$. Boundary conventions are $f(0) := \lim_{u\downarrow 0} f(u) \in (-\infty, \infty]$ and $f_\dagger(0) := \lim_{u\downarrow 0} f_\dagger(u) = \lim_{t\uparrow\infty} f(t)/t \in (-\infty, \infty]$. For $p, q \in \mathcal{P}(\mathcal{A})$ over finite $\mathcal{A}$, the $f$–divergence is

$$\mathrm{D}^f(p\|q) := \sum_{a\in\mathcal{A}:q(a)>0} q(a)f(p(a)/q(a)) + f_\dagger(0) \cdot \sum_{a\in\mathcal{A}:q(a)=0} p(a), \tag{13}$$

with conventions: $q(a)f(0)$ if $q(a) > 0, p(a) = 0$, and 0 if $p(a) = q(a) = 0$ (Rényi, 1961; Csiszár, 1967; Liese & Vajda, 2006). $f$–divergences satisfy $\mathrm{D}^f(p\|q) \in [0, \infty]$, are jointly convex (Csiszár, 1967), vanish iff $p = q$, and are not symmetric ($\mathrm{D}^{f_\dagger}(p\|q) = \mathrm{D}^f(p\|q)$).

| Symbols | Meaning | Definition |
|---|---|---|
| $\mathcal{S}$ | State space | Section 2 |
| $\mathcal{A}$ | Action space | Section 2 |
| $\gamma$ | Discount factor | Section 2 |
| P | Transition kernel | Section 2 |
| r | Reward function | Section 2 |
| $\pi$ | Policy | Section 2 |
| $\pi_{\mathrm{ref}}$ | Reference policy used in the regularization problem | Section 2 |
| $f$ | Divergence generator | Section 2 |
| $\lambda$ | Temperature of the regularization | Section 2 |
| $\rho$ | Initial state distribution | Section 2 |
| $\kappa_f$ | Upper bound on $|f'''(x)/f''(x)^2|$ | $\mathbf{A}_f(\pi_{\mathrm{ref}})$ |
| $\omega_f$ | Upper bound on $1/(xf''(x)^2)$ | $\mathbf{A}_f(\pi_{\mathrm{ref}})$ |
| $\mathrm{d}_f$ | Upper bound on a set of divergences | (6) |
| $\mathrm{y}_f$ | Upper bound on a quantity that depends on $f''$ and $f'$ | (6) |
| $\zeta_f$ | Lower bound on a quantity that depends on $f''$ | (7) |
| $v_\pi$ | Value function of a policy $\pi$ | (1) |
| $v_\pi^f$ | Regularized Value function of a policy $\pi$ | (42) |
| $\mathrm{P}_\pi$ | Transition kernel induced by policy $\pi$ | Section 2 |
| $q_\pi^f$ | Regularized Q-function of a policy $\pi$ | (47) |
| $d_\rho^\pi$ | discounted state visitation of a policy $\pi$ | (48) |
| $\theta$ | Parameter of the policy (element of $\mathbb{R}^{|\mathcal{S}||\mathcal{A}|}$) | Section 3 |
| $\pi_\theta^f$ | The soft-$f$-argmax policy associated with $\theta$ | (4) |
| $\mathrm{w}_\theta^f$ | A matrix of size $\mathbb{R}^{\mathcal{S}\times\mathcal{A}}$ such that for any $s \in \mathcal{S}$, $\mathrm{w}_\theta^f(\cdot|s) \in \mathcal{P}(\mathcal{A})$ | (5) |
| $f_\theta'(a|s)$ | shorthand notation for $f'(\pi_\theta^f(a|s)/\pi_{\mathrm{ref}}(a|s))$ | (35) |
| $f_\theta''(a|s)$ | shorthand notation for $f''(\pi_\theta^f(a|s)/\pi_{\mathrm{ref}}(a|s))$ | (36) |
| $f_\theta'''(a|s)$ | short hand notation for $f'''(\pi_\theta^f(a|s)/\pi_{\mathrm{ref}}(a|s))$ | (37) |
| $\mathrm{W}_\theta^f(s)$ | A function of $f_\theta''(\cdot|s)$ | (38) |
| $\mathrm{Y}_\theta^f(s)$ | A function of $f_\theta'(\cdot|s)$ and $f_\theta''(\cdot|s)$ | (38) |
| $T$ | Number of iterations performed by $f$-PG | Algorithm 1 |
| $\mathrm{g}_Z^f(\theta)$ | Stochastic estimator of the gradient at $\theta$ | (11) |
| $H$ | Truncation horizon in $f$-PG | Algorithm 1 |
| $\mathrm{g}_Z^f(\theta)$ | Stochastic estimator of the gradient at $\theta$ | (11) |
| $\beta_f$ | Bias of the stochastic estimator at $\theta$ | Lemma 4.2 |
| $\sigma_f$ | Variance of the stochastic estimator at $\theta$ | Lemma 4.2 |
| $L_f$ | Local smoothness of the objective at $\theta$ | Theorem 3.3 |
| $\mathcal{P}(\mathcal{A})$ | Set of probability measures over $\mathcal{A}$ | Section 2 |
| $\mathrm{D}^f(p\|q)$ | $f$-divergence between two probability measures $p$ and $q$ | (13) |

**Distribution of the state-action sequence.** The state–action sequence $(S_t, A_t)_{t\geq 0}$ defines a stochastic process on the canonical space $(\mathcal{S} \times \mathcal{A})^{\mathbb{N}}$. For any initial state $s_0 \in \mathcal{S}$, we denote by $\mathbb{P}_{s_0}^\pi$ the law of this process. That is, for any $n \in \mathbb{N}$ and any subset $B \subset (\mathcal{S} \times \mathcal{A})^n$,

$$\mathbb{P}_{s_0}^\pi(B) = \sum_{(a_0,\ldots,a_{n-1})\in\mathcal{A}^n} \sum_{(s_1,\ldots,s_{n-1})\in\mathcal{S}^{n-1}} \mathbb{1}_B\big((s_0,a_0),\ldots,(s_{n-1},a_{n-1})\big) \prod_{i=0}^{n-1} \pi(a_i|s_i)\, \mathrm{P}(s_{i+1}|s_i,a_i),$$

with the convention $s_0$ is the given initial state. We denote by $\mathbb{E}_{s_0}^\pi$ the corresponding expectation operator. In particular, the state sequence $(s_t)_{t\geq 0}$ defines a Markov reward process (Section 2.1.6 in Puterman (1994)) with transition kernel

$$\mathrm{P}_\pi(s' \mid s) = \sum_{a\in\mathcal{A}} \mathrm{P}(s' \mid s, a)\, \pi(a \mid s) \ .$$

**Norms.** For $x \in \mathbb{R}^d$, we define the norms

$$\|x\|_\infty = \max_{i\in\{1,\ldots,d\}} |x_i| \ , \quad \|x\|_1 = \sum_{i=1}^d |x_i| \ , \quad \|x\|_2 = \left(\sum_{i=1}^d |x_i|^2\right)^{1/2} \ .$$

For a $d \times d$ matrix $M$, we denote by $\|M\|_\infty$, and $\|M\|_2$ respectively the max *row* sum, and the spectral norm:

$$\|M\|_\infty = \sup_{x \neq 0}\{\|Mx\|_\infty / \|x\|_\infty\} = \sup_{i \in \{1,\dots,d\}} \sum_{j=1}^{d} |M_{i,j}| \ , \quad \|M\|_2 = \sup_{x \neq 0}\{\|Mx\|_2 / \|x\|_2\} \ . \tag{14}$$

Recall that, for any $x \in \mathbb{R}^d$, $\|Mx\|_\infty \leq \|M\|_\infty \|x\|_\infty$ and $\|Mx\|_2 \leq \|M\|_2 \|x\|_2$.

**Functional and matrix forms.** For notational convenience, we also view $\mathsf{P}$ as a $(|\mathcal{S}| \cdot |\mathcal{A}|) \times |\mathcal{S}|$ matrix with entries $\mathsf{P}_{(s,a),s'} = \mathsf{P}(s' \mid s, a)$. Similarly, $v_\pi^f$ is a vector of size $|\mathcal{S}|$ and $q_\pi^f$ a vector of size $|\mathcal{S}| \times |\mathcal{A}|$. Finally, we identify the parameter $\theta \in \mathbb{R}^{\mathcal{S} \times \mathcal{A}}$ with its matrix representation $\theta \in \mathbb{R}^{|\mathcal{S}||\mathcal{A}|}$, indexed by $(s, a) \in \mathcal{S} \times \mathcal{A}$. This slight abuse of notation allows us to conveniently switch between functional and matrix views.

# B   SMOOTHNESS OF THE OBJECTIVE

In this section, we establish the smoothness of the regularized value function $v_\theta^f := v_{\pi_\theta}^f(\rho)$ with respect to the parameter $\theta$. As a first step, we show that the policy $\pi_\theta^f$ is smooth under suitable assumptions on the divergence generator $f$ and compute its first and second derivatives. To do so, we start by studying the properties of the soft-$f$-argmax operator and then apply the obtained results to derive properties of the policy $\pi_\theta^f$.

## B.1   PROPERTIES OF THE SOFT-$f$-ARGMAX

In this Section, we compute the derivative of $\nu_x^f(\cdot) := \text{f-softargmax}(x, \nu_{\text{ref}})$ and x-softmax$(\cdot) := \text{f-softmax}(x, \nu_{\text{ref}})$, where

$$\text{f-softmax}(x, \nu_{\text{ref}}) := \max_{\nu \in \mathcal{P}(\mathcal{A})} \left\{ \langle \nu, x \rangle - \mathrm{D}^f(\nu \| \nu_{\text{ref}}) \right\} \ . \tag{15}$$

The function x-softmax$(\cdot)$ is the Fenchel–Legendre transform of $\mathrm{D}^f(\cdot \| \nu_{\text{ref}})$, and the results in this section are therefore standard results from convex analysis, statements of which can be found in various forms in Hiriart-Urruty & Lemaréchal (2004); Mensch & Blondel (2018); Geist et al. (2019); Roulet et al. (2025).

In this section, we fix a reference probability distribution $\nu_{\text{ref}} \in \mathcal{P}(\mathcal{A})$ such that for any $a \in \mathcal{A}$, we have $\nu_{\text{ref}}(a) > \underline{\nu_{\text{ref}}}$ for some $\min_{a \in \mathcal{A}} \nu_{\text{ref}}(a) > \underline{\nu_{\text{ref}}} > 0$. For a given $x \in \mathbb{R}^{|\mathcal{A}|}$, we define

$$\nu_x^f(\cdot) := \text{f-softargmax}(x, \nu_{\text{ref}}) = \arg\max_{\nu \in \mathcal{P}(\mathcal{A})} \left\{ \langle \nu, x \rangle - \mathrm{D}^f(\nu \| \nu_{\text{ref}}) \right\} \ . \tag{16}$$

The following result is a simplified version of (Roulet et al., 2025, Proposition 1). For the sake of completeness, we provide a full proof.

**Lemma B.1.** *Assume that $f$ is strictly convex on $[0, 1/\underline{\nu_{ref}}]$, differentiable on $(0, 1/\underline{\nu_{ref}})$, with $\lim_{x \downarrow 0^+} f'(u) = -\infty$, for some $\underline{\nu_{ref}} > 0$. Let $\nu_{\text{ref}}$ be a policy such that $\min_{a \in \mathcal{A}} \nu_{\text{ref}}(a) > \underline{\nu_{ref}}$. For any $x \in \mathbb{R}^{|\mathcal{A}|}$ and $a \in \mathcal{A}$, we have $0 < \nu_x^f(a)$. Moreover, for all $x \in \mathbb{R}^{|\mathcal{A}|}$, there exists a unique $\mu_x \in I_{\underline{\nu_{ref}}}(x)$, where*

$$I_{\underline{\nu_{ref}}}(x) := (\max_{a \in \mathcal{A}} x(a) - f'(1/\underline{\nu_{ref}}), \max_{a \in \mathcal{A}} x(a) - f'(1)), \tag{17}$$

*such that*

$$\nu_x^f(a) = \nu_{\text{ref}}(a)[f']^{-1}(x(a) - \mu_x) \,. \tag{18}$$

*Moreover, $\mu_x \in \mathbb{R}$ is the unique root of the equation*

$$F(x, \mu) := \sum_{a \in \mathcal{A}} \nu_{\text{ref}}(a)[f']^{-1}(x(a) - \mu) - 1 = 0 \quad \text{for } \mu \in I_{\underline{\nu_{ref}}}(x). \tag{19}$$

*Proof.* Under the stated assumption, the map $f'$ is strictly increasing on $(0, 1/\nu_{\text{ref}}]$, hence injective; therefore it is invertible *onto its image*. Moreover,

$$\text{Dom}\left([f']^{-1}\right) = f'\left((0, 1/\nu_{\text{ref}}]\right) = \left(-\infty, \ f'(1/\nu_{\text{ref}})\right], \tag{20}$$

which is an interval and the function $[f']^{-1}$ is strictly increasing.

Fix $x \in \mathcal{A}$, and $(a, b) \in \mathcal{A} \times \mathcal{A}$. Recall from (16) the definition of the soft-$f$-argmax ,

$$\nu_x^f = \arg\max_{\nu \in \mathcal{P}(\mathcal{A})} \left\{ \langle \nu, x \rangle - \mathrm{D}^f(\nu \| \nu_{\text{ref}}) \right\}$$

This is a strictly concave optimization problem over the probability simplex $\mathcal{P}(\mathcal{A})$ so it admits a unique maximizer. We now characterize the maximizer via the KKT conditions. Introduce multipliers $\mu \in \mathbb{R}$ for the equality constraint $\sum_{c \in \mathcal{A}} \nu(c) = 1$, and for every $c \in \mathcal{A}$, $\lambda(c) \in \mathbb{R}^+$ for the non-negativity constraints $\nu(c) \geq 0$. The Lagrangian reads

$$L(\nu, \mu, \{\lambda(c)\}_{c \in \mathcal{A}}) = \sum_{c \in \mathcal{A}} \nu(c)\, x(c) - \mathrm{D}^f(\nu \| \nu_{\text{ref}}) + \mu\left(1 - \sum_{c \in \mathcal{A}} \nu(c)\right) - \sum_{c \in \mathcal{A}} \lambda(c)\, \nu(c) \ .$$

By the differentiability of $f$, differentiating the Lagrangian with respect to $\nu(a)$ gives

$$\frac{\partial L}{\partial \nu(a)} = x(a) - f'\left(\frac{\nu(a)}{\nu_{\text{ref}}(a)}\right) - \mu - \lambda(a) \ . \tag{21}$$

At the optimum $(\nu_x^f, \mu_x, \{\lambda_x(c)\}_{c \in \mathcal{A}})$, the KKT conditions yield:

$$\nu_x^f(c)\, \lambda_x(c) = 0, \qquad \forall c \in \mathcal{A}, \tag{22}$$

$$x(c) - f'\left(\frac{\nu_x^f(c)}{\nu_{\text{ref}}(c)}\right) - \mu_x - \lambda_x(c) = 0, \qquad \forall c \in \mathcal{A}. \tag{23}$$

Under the stated assumptions, $\lim_{x \to 0^+} f'(x) = -\infty$. Hence, if $\nu_x^f(c) = 0$ for some $c$, the stationarity condition (23) cannot hold with finite multipliers. Therefore $\nu_x^f(c) > 0$ for all $c \in \mathcal{A}$, which by (22) implies $\lambda_x(c) = 0$. Thus, for each $c \in \mathcal{A}$ the stationarity condition reduces to

$$x(c) - f'\left(\frac{\nu_x^f(c)}{\nu_{\text{ref}}(c)}\right) = \mu_x \ . \tag{24}$$

Note also that (24) also implies that for all $a \in \mathcal{A}$, $x(a) - \mu_x \in \text{Dom}(f')$, which implies, using (20) that $\max_{a \in \mathcal{A}} x(a) - f'(1/\nu_{\text{ref}}) \leq \mu_x$. Together with (24), this shows (18). Note that $\mu_x$ is a root of (19), since using (18),

$$\sum_{a \in \mathcal{A}} \nu_{\text{ref}}(a)[f']^{-1}(x(a) - \mu_x) = \sum_{a \in \mathcal{A}} \nu_x^f(a) = 1.$$

Because $[f']^{-1}$ is strictly increasing on $(-\infty, f'(1/\nu_{\text{ref}})]$, for each $x \in \mathbb{R}^{|\mathcal{A}|}$, the function $\mu \mapsto F(x, \mu)$ (see (19)) is strictly decreasing on $(-\infty, f'(1/\nu_{\text{ref}})]$. Strict monotonicity gives the uniqueness of $\mu_x$. Note finally that if $\mu > \max_{a \in \mathcal{A}} x(a) - f'(1)$, then $x(a) - \mu < f'(1)$, and since $[f']^{-1}$ is strictly increasing, $[f']^{-1}(x(a) - \mu) < 1$, showing that $F(x, \mu) < 0$, which concludes the proof. $\qquad\square$

***Remark* B.2** *Let $f : (0, \tau) \to \mathbb{R}$ be a strictly convex and differentiable on $(0, \tau)$, $\text{Dom}(f') = (0, \tau)$, and $f'$ is strictly increasing and continuous on $(0, \tau)$. Let $\alpha := \lim_{x \to 0} f'(x) \in [-\infty, +\infty)$ and $\beta := \lim_{x \to \tau} f'(x) \in (-\infty, +\infty]$. Then $f'\left((0, \tau)\right) = (\alpha, \beta)$, i.e., $f' : (0, \tau) \to (\alpha, \beta)$ is a strictly increasing bijection (hence admits a continuous inverse $[f']^{-1} : (\alpha, \beta) \to (0, \tau)$). Define the convex conjugate of $f$,*

$$f^*(y) := \sup_{x \in (0, \tau)} \{xy - f(x)\} \ . \tag{25}$$

*The two following properties hold,*

*(i) For every $y \in (\alpha, \beta)$, the supremum is attained at a unique point $x = (f')^{-1}(y)$.*

*(ii)* $\mathrm{Dom}(f^*) \subseteq [\alpha, \beta]$ *(with the convention that an infinite endpoint is excluded). Specifically,* $\mathrm{Dom}(f^*) \cap (\alpha, \beta) = (\alpha, \beta)$, *and* $f^*(y) = +\infty$ *for* $y \notin [\alpha, \beta]$. *At an endpoint* $y = \alpha$ *(resp.* $y = \beta$*), if finite,* $f^*(y) = \lim_{x \downarrow 0}(yx - f(x))$ *(resp.* $\lim_{x \uparrow \tau}(yx - f(x))$*), so* $f^*(\alpha)$ *(resp.* $f^*(\beta)$*) is finite iff the corresponding one–sided limit is finite.*

*On the open interval* $(\alpha, \beta)$, *the function* $f^*$ *is differentiable and*

$$(f^*)'(y) = [f']^{-1}(y), \qquad y \in (\alpha, \beta),$$

*We retrieve the statement in Proposition 1 of* Roulet et al. (2025) *by replacing* $[f']^{-1}$ *by* $(f^*)'$. *In most examples, there is no need to resort to the convex conjugate to compute the inverse.*

**Lemma B.3.** *Assume, in addition to Lemma B.1, that* $f$ *is two-times continuously differentiable on* $(0, \underline{\nu_{ref}}$. *Then, the function* $x \mapsto \mu_x$ *is continuously differentiable on* $\mathbb{R}^{|\mathcal{A}|}$, *and for all* $a \in \mathcal{A}$,

$$\frac{\partial}{\partial x(a)} \mu_x = -\frac{\partial}{\partial x(a)} F(x, \mu_x) \Big/ \frac{\partial}{\partial \mu} F(x, \mu_x) \tag{26}$$

*Proof.* The function $(x, \mu) \mapsto F(x, \mu)$ is two-times continuously differentiable on the open set $U := \{(x, \mu) : x \in \mathbb{R}^{|\mathcal{A}|}, \mu \in I_{\nu_{\mathrm{ref}}}(x)\} \subset \mathbb{R}^{|\mathcal{A}|} \times \mathbb{R}$. Let $x_0 \in \mathbb{R}^{|\mathcal{A}|}$. Since $[f']^{-1}$ is strictly increasing,

$$\frac{\partial}{\partial \mu} F(x, \mu_x) = \sum_{a \in \mathcal{A}} \nu_{\mathrm{ref}}(a) \frac{1}{f''([f']^{-1}(x_0(a) - \mu_{x_0}))} > 0.$$

Hence, we may apply the implicit function theorem, which shows that there exists an open neighborhood $V_{x_0}$ and a unique function $x \mapsto \mu_x$ on $V_{x_0}$, such that for all $x \in V_{x_0}$, $F(x, \mu_x) = 0$ and (26) holds. $\qquad \square$

We now introduce some compact notations that will be used throughout the sequel. For any $a \in \mathcal{A}$, define the first three derivatives of $f$ evaluated at the probability ratio $\nu_x^f(a)/\nu_{\mathrm{ref}}(a)$:

$$f_x'(a) := f'\left(\frac{\nu_x^f(a)}{\nu_{\mathrm{ref}}(a)}\right) \ , \quad f_x''(a) := f''\left(\frac{\nu_x^f(a)}{\nu_{\mathrm{ref}}(a)}\right) \ , \quad f_x'''(a) := f'''\left(\frac{\nu_x^f(a)}{\nu_{\mathrm{ref}}(a)}\right) \ . \tag{27}$$

In addition, we introduce the quantities

$$\mathrm{W}_x^f := \sum_{a \in \mathcal{A}} \frac{\nu_{\mathrm{ref}}(a)}{f_x''(a)} \ . \tag{28}$$

Importantly, as $f$ is strictly convex, its second derivative is strictly positive, making the preceding quantity well-defined. For any $a \in \mathcal{A}$, we also define the following normalized weights

$$\mathrm{w}_x^f(a) = \frac{1}{\mathrm{W}_x^f} \frac{\nu_{\mathrm{ref}}(a)}{f_x''(a)} \ . \tag{29}$$

**Lemma B.4.** *Assume* $A_f(\nu_{ref})$. *Then the soft-$f$-argmax* $\nu_x^f$ *is twice continuously differentiable with respect to* $x$. *Moreover, for any* $x \in \mathbb{R}^{|\mathcal{A}|}$, *and* $(a, b) \in \mathcal{A}^2$, *we have*

$$\frac{1}{\mathrm{W}_x^f} \cdot \frac{\partial \nu_x^f(a)}{\partial x(b)} = \mathbb{1}_b(a) \, \mathrm{w}_x^f(a) - \mathrm{w}_x^f(a) \, \mathrm{w}_x^f(b) \ ,$$

*In addition, for any* $(a, b, c) \in \mathcal{A}^3$, *the second derivative satisfies*

$$\begin{aligned}
\frac{1}{\mathrm{W}_x^f} \cdot \frac{\partial \nu_x^f(a)}{\partial x(b)\partial x(c)} &= -\mathbb{1}_b(a)\mathbb{1}_c(a)\frac{f_x'''(a)}{f_x''(a)^2} \cdot \mathrm{w}_x^f(a) + \mathbb{1}_c(b)\,\mathrm{w}_x^f(a)\,\mathrm{w}_x^f(b)\frac{f_x'''(b)}{f_x''(b)^2} \\
&\quad + \mathbb{1}_b(a)\,\mathrm{w}_x^f(a)\,\mathrm{w}_x^f(c)\frac{f_x'''(a)}{f_x''(a)^2} + \mathbb{1}_c(a)\,\mathrm{w}_x^f(a)\,\mathrm{w}_x^f(b)\frac{f_x'''(a)}{f_x''(a)^2} \\
&\quad - \mathrm{w}_x^f(a) \cdot \mathrm{w}_x^f(b) \cdot \mathrm{w}_x^f(c) \cdot \left[\frac{f_x'''(a)}{f_x''(a)^2} + \frac{f_x'''(b)}{f_x''(b)^2} + \frac{f_x'''(c)}{f_x''(c)^2}\right] \\
&\quad + \mathrm{w}_x^f(a) \cdot \mathrm{w}_x^f(b) \cdot \mathrm{w}_x^f(c) \cdot \sum_{d \in \mathcal{A}} \mathrm{w}_x^f(d) \cdot \frac{f_x'''(d)}{f_x''(d)^2} \ .
\end{aligned}$$

*Proof.* Fix $x \in \mathbb{R}^{|\mathcal{A}|}$ and $(a, b, c) \in \mathcal{A}^3$. Define

$$F(x; \mu) = \sum_{a \in \mathcal{A}} \nu_{\text{ref}}(a)[f']^{-1}(x(a) - \mu_x) - 1 \ .$$

**First derivative.** Importantly, using Lemma B.1 we have that

$$F(x; \mu_x) = 0 \ .$$

Differentiating the previous identity with respect to $x(b)$, yields

$$\frac{\partial F(x; \mu_x)}{\partial x(b)} = \sum_{a \in \mathcal{A}} \nu_{\text{ref}}(a) \frac{1}{f''([f']^{-1}(x(a) - \mu_x))} \left( 1_b(a) - \frac{\partial \mu_x}{\partial x(b)} \right) \ ,$$

where we used that the derivative of $[f']^{-1}$ is $1/f''([f']^{-1})$. Next, using from Lemma B.1 that $\nu_x^f(a) = \nu_{\text{ref}}(a)[f']^{-1}(x(a) - \mu_x)$ yields

$$\frac{\partial F(x; \mu_x)}{\partial x(b)} = \sum_{a \in \mathcal{A}} \nu_{\text{ref}}(a) \frac{1}{f_x''(a)} \left( 1_b(a) - \frac{\partial \mu_x}{\partial x(b)} \right) = 0 \ ,$$

where $f_x''(a)$ is defined in (27). This implies

$$\frac{\partial \mu_x}{\partial x(b)} = \text{w}_x^f(b) \ , \tag{30}$$

where $\text{w}_x^f(b)$ is defined in (29). Now that we have computed the derivative of the normalization factor $\mu_x$, we can compute the derivative of the policy. Starting from Lemma B.1, we have that

$$\nu_x^f(a) = \nu_{\text{ref}}(a)[f']^{-1}(x(a) - \mu_x) \ .$$

Differentiating the previous identity with respect to $x(b)$, yields

$$\frac{\partial \nu_x^f(a)}{\partial x(b)} = \frac{\nu_{\text{ref}}(a)}{f_x''(a)} \left[ 1_b(a) - \frac{\partial \mu_x}{\partial x(b)} \right] = \frac{\nu_{\text{ref}}(a)}{f_x''(a)} \left[ 1_b(a) - \text{w}_x^f(b) \right] \ ,$$

where in the last identity, we used the expression of the derivative of $\mu_x$ given in (30). Finally, using the definition of $\text{W}_x^f$ given in (28) gives

$$\boxed{\frac{\partial \nu_x^f(a)}{\partial x(b)} = 1_b(a) \frac{\nu_{\text{ref}}(a)}{f_x''(a)} - \frac{\nu_{\text{ref}}(a)}{f_x''(a)} \frac{\nu_{\text{ref}}(b)}{f_x''(b)} \cdot \frac{1}{\text{W}_x^f}} \ , \tag{31}$$

**Second derivative** From (31), we aim to differentiate once more with respect to $x(c)$. First, note that it holds that

$$\frac{\partial f_x''(a)}{\partial x(c)} = \frac{f_x'''(a)}{\nu_{\text{ref}}(a)} \cdot \frac{\partial \nu_x^f(a)}{\partial x(c)} = 1_c(a) \frac{f_x'''(a)}{f_x''(a)} - \frac{\nu_{\text{ref}}(c) f_x'''(a)}{f_x''(a) f_x''(c)} \cdot \frac{1}{\text{W}_x^f} \ , \tag{32}$$

$$\frac{\partial \text{W}_x^f}{\partial x(c)} = -\sum_{d \in \mathcal{A}} \frac{\nu_{\text{ref}}(d)}{f_x''(d)^2} \frac{\partial f_x''(d)}{\partial x(c)} = -\frac{\nu_{\text{ref}}(c) f_x'''(c)}{f_x''(c)^3} + \frac{1}{\text{W}_x^f} \sum_{d \in \mathcal{A}} \frac{\nu_{\text{ref}}(c) \nu_{\text{ref}}(d) f_x'''(d)}{f_x''(d)^3 f_x''(c)} \ , \tag{33}$$

Now computing the second derivative of $\nu_x^f$ gives

$$\frac{\partial \nu_x^f(a)}{\partial x(b) \partial x(c)} = -1_b(a) \frac{\nu_{\text{ref}}(a)}{f_x''(a)^2} \frac{\partial f_x''(a)}{\partial x(c)} + \frac{\nu_{\text{ref}}(a) \nu_{\text{ref}}(b)}{f_x''(a)^2 f_x''(b)} \cdot \frac{1}{\text{W}_x^f} \frac{\partial f_x''(a)}{\partial x(c)}$$

$$+ \frac{\nu_{\text{ref}}(a) \nu_{\text{ref}}(b)}{f_x''(a) f_x''(b)^2} \cdot \frac{1}{\text{W}_x^f} \frac{\partial f_x''(b)}{\partial x(c)} + \frac{\nu_{\text{ref}}(a) \nu_{\text{ref}}(b)}{f_x''(a) f_x''(b)} \cdot \frac{1}{(\text{W}_x^f)^2} \frac{\partial \text{W}_x^f}{\partial x(c)}$$

Plugging in (32), and (33) in the preceding inequality yields

$$\frac{\partial \nu_{(}^{\theta} a)}{\partial x(b) \partial x(c)} = -1_b(a) \frac{\nu_{\text{ref}}(a)}{f_x(a)^2} \left[ 1_c(a) \frac{f_x'''(a)}{f_x''(a)} - \frac{\nu_{\text{ref}}(c) f_x'''(a)}{f_x''(a) f_x''(c)} \cdot \frac{1}{\text{W}_x^f} \right]$$

$$+ \frac{\nu_{\text{ref}}(a)\pi_{\text{ref}}(b)}{f_x''(a)^2 f_x''(b)} \cdot \frac{1}{W_x^f} \left[ 1_c(a) \frac{f_x'''(a)}{f_x''(a)} - \frac{\nu_{\text{ref}}(c) f_x'''(a)}{f_x''(a) f_x''(c)} \cdot \frac{1}{W_x^f} \right]$$

$$+ \frac{\nu_{\text{ref}}(a)\pi_{\text{ref}}(b)}{f_x''(a) f_x''(b)^2} \cdot \frac{1}{W_x^f} \left[ 1_c(b) \frac{f_x'''(b)}{f_x''(b)} - \frac{\nu_{\text{ref}}(c) f_x'''(b)}{f_x''(b) f_x''(c)} \cdot \frac{1}{W_x^f} \right]$$

$$+ \frac{\nu_{\text{ref}}(a)\nu_{\text{ref}}(b)}{f_x''(a) f_x''(b)} \cdot \frac{1}{(W_x^f)^2} \left[ -\frac{\nu_{\text{ref}}(c) f_x'''(c)}{f_x''(c)^3} + \frac{1}{W_x^f} \sum_{d \in \mathcal{A}} \frac{\nu_{\text{ref}}(c)\nu_{\text{ref}}(d) f_x'''(d)}{f_x''(d)^3 f_x''(c)} \right] \;,$$

which concludes the proof. $\qquad \square$

The following lemma links the gradients f-softmax and f-softargmax operators.

**Lemma B.5.** *Assume $\mathbf{A}_f(\nu_{\text{ref}})$. For any $x \in \mathbb{R}^{|\mathcal{A}|}$, it holds that*

$$\frac{\partial \, \text{f-softmax}(x, \nu_{\text{ref}})}{\partial x} = \text{f-softargmax}(x, \nu_{\text{ref}}) \quad \text{and} \quad \left\| \frac{\partial^2 \, \text{f-softmax}(x, \nu_{\text{ref}})}{\partial x^2} \right\|_2 \le 2\, W_x^f \;.$$

*Proof.* For any $x \in \mathcal{A}$ and $\nu \in \mathcal{P}(\mathcal{A})$, define

$$h^f(x, \nu) = \langle \nu, x \rangle - D^f(\nu \| \nu_{\text{ref}}) \;. \tag{34}$$

Fix $b \in \mathcal{A}$ and note that

$$\frac{\partial h^f(x, \nu)}{\partial x(b)} = \nu(b) \;.$$

It holds that $\text{f-softmax}(x) = \max_{\nu \in \mathcal{P}(\mathcal{A})} h^f(x, \nu)$. As $h^f$ is continuous in its two variables, $\mathcal{P}(\mathcal{A})$ is a compact set, and for every $x \in \mathbb{R}^{|\mathcal{A}|}$, the function $h^f(x, \cdot)$ admits a unique optimizer in $\mathcal{P}(\mathcal{A})$, then by Danskin's theorem (Lemma I.5)

$$\frac{\partial \, \text{f-softmax}(x, \nu_{\text{ref}})}{\partial x(b)} = \frac{\partial h^f(x, \nu^\star(x))}{\partial x} = \nu^\star(x) \;, \quad \text{where } \nu^\star(x) = \arg\max_{\nu \in \mathcal{P}(\mathcal{A})} h^f(x, \nu) \;.$$

Finally, using that

$$\nu^\star(x) = \text{f-softargmax}(x, \nu_{\text{ref}}) \;,$$

establishes the first identity of the lemma. Using the fact that for any matrix $A \in \mathbb{R}^{d \times d}$, we have $\|A\|_2 \le \sum_{i=1}^d \sum_{j=1}^d |a_{i,j}|$, implies

$$\left\| \frac{\partial^2 \, \text{f-softmax}(x, \nu_{\text{ref}})}{\partial x^2} \right\|_2 \le \sum_{a \in \mathcal{A}} \sum_{b \in \mathcal{A}} \left| \frac{\partial \nu_x^f(a)}{\partial x(b)} \right| \le \sum_{a \in \mathcal{A}} \sum_{b \in \mathcal{A}} W_x^f \left| 1_b(a) \, w_x^f(a) - w_x^f(a) \, w_x^f(b) \right|$$

where in the last equality, we used Lemma B.4. Finally, applying the triangle inequality establishes the second claim of the lemma. $\qquad \square$

## B.2 Specification to the $f$-divergence generators of Table 2

We now specify the particular forms that these quantities take under three choices of the function $f$: the KL divergence generator($f(u) = u \log u$), the $\alpha$-Csiszár–Cressie–Read divergence generator for $0 < \alpha < 1$, and the Hellinger divergence generator

**KL case** ($f(u) = u \log u$). Since $f'(u) = \log u + 1$, $f''(u) = 1/u$, and $[f']^{-1}(y) = \exp(y - 1)$, the soft-$f$ operators specialize as follows (with base measure $\nu_{\text{ref}} \in \mathcal{P}(\mathcal{A})$):

$$\nu_x^{\text{KL}}(a) = \nu_{\text{ref}}(a)[f']^{-1}\big(x(a) - \mu_x\big) = \frac{\nu_{\text{ref}}(a) \, \exp(x(a))}{\sum_{b \in \mathcal{A}} \nu_{\text{ref}}(b) \, \exp(x(b))} \,, \qquad a \in \mathcal{A},$$

where the normalizer is

$$\mu_x \;=\; -1 + \log\Big( \sum_{b \in \mathcal{A}} \nu_{\text{ref}}(b) \, \exp(x(b)) \Big).$$

The associated softmax function is the log-partition:

$$\text{KL-softmax}(x, \nu_{\text{ref}}) = \log\Big(\sum_{a \in \mathcal{A}} \nu_{\text{ref}}(a) \exp(x(a))\Big).$$

For the curvature quantities in (28)–(29), since

$$f''\Big(\frac{\nu_x^{\text{KL}}(a)}{\nu_{\text{ref}}(a)}\Big) = \Big(\frac{\nu_x^{\text{KL}}(a)}{\nu_{\text{ref}}(a)}\Big)^{-1} = \frac{\nu_{\text{ref}}(a)}{\nu_x^{\text{KL}}(a)},$$

we obtain

$$\text{W}_x^{\text{KL}} = \sum_{a \in \mathcal{A}} \frac{\nu_{\text{ref}}(a)}{f_x''(a)} = \sum_{a \in \mathcal{A}} \nu_x^{\text{KL}}(a) = 1$$

and the corresponding normalized weights are

$$\text{w}_x^{\text{KL}}(a) = \frac{1}{\text{W}_x^{\text{KL}}} \frac{\nu_{\text{ref}}(a)}{f_x''(a)} = \nu_x^{\text{KL}}(a), \qquad a \in \mathcal{A}.$$

**Tsallis-$\alpha$ case** ($0 < \alpha < 1$). Let $f(u) = \dfrac{u^\alpha - \alpha u + \alpha - 1}{\alpha(\alpha - 1)}$, so that

$$f'(u) = \frac{u^{\alpha-1} - 1}{\alpha - 1}, \qquad f''(u) = u^{\alpha-2}, \qquad [f']^{-1}(y) = \big[1 + (\alpha - 1)y\big]^{\frac{1}{\alpha-1}}.$$

The soft-$f$ operators specialize (with base measure $\nu_{\text{ref}} \in \mathcal{P}(\mathcal{A})$) to

$$\nu_x^{\text{TS}}(a) = \nu_{\text{ref}}(a)[f']^{-1}(x(a) - \mu_x) = \nu_{\text{ref}}(a)\big[1 + (\alpha - 1)\big(x(a) - \mu_x\big)\big]^{1/(\alpha-1)}, \qquad a \in \mathcal{A},$$

where $\mu_x$ is the unique normalizer satisfying the constraint

$$\sum_{a \in \mathcal{A}} \nu_{\text{ref}}(a) \big[1 + (\alpha - 1)\big(x(a) - \mu_x\big)\big]^{\frac{1}{\alpha-1}} = 1,$$

with the domain condition $1 + (\alpha - 1)\big(x(a) - \mu_x\big) > 0$ for all $a \in \mathcal{A}$.

The associated softmax is

$$\text{TS-softmax}(x, \nu_{\text{ref}}) = \mu_x - \frac{1}{\alpha} + \frac{1}{\alpha}\sum_{a \in \mathcal{A}} \nu_{\text{ref}}(a) \big[1 + (\alpha - 1)\big(x(a) - \mu_x\big)\big]^{\frac{\alpha}{\alpha-1}}.$$

For the curvature quantities in (28)–(29), set

$$u_x(a) := \frac{\nu_x^{\text{TS}}(a)}{\nu_{\text{ref}}(a)} = \big[1 + (\alpha - 1)\big(x(a) - \mu_x\big)\big]^{\frac{1}{\alpha-1}}.$$

Since $f''\big(u_x(a)\big) = u_x(a)^{\alpha-2}$, we have

$$\text{W}_x^{\text{TS}} = \sum_{a \in \mathcal{A}} \frac{\nu_{\text{ref}}(a)}{f_x''(a)} = \sum_{a \in \mathcal{A}} \nu_{\text{ref}}(a) \big[1 + (\alpha - 1)\big(x(a) - \mu_x\big)\big]^{(2-\alpha)/(\alpha-1)},$$

and the corresponding normalized weights are, for $a \in \mathcal{A}$,

$$\text{w}_x^{\text{TS}}(a) = \frac{1}{\text{W}_x^f} \frac{\nu_{\text{ref}}(a)}{f_x''(a)} = \frac{\nu_{\text{ref}}(a) \big[1 + (\alpha - 1)\big(x(a) - \mu_x\big)\big]^{(2-\alpha)/(\alpha-1)}}{\sum_{c \in \mathcal{A}} \nu_{\text{ref}}(c) \big[1 + (\alpha - 1)\big(x(c) - \mu_x\big)\big]^{(2-\alpha)/(\alpha-1)}}.$$

**Jensen-Shannon.** Let $f(u) = \frac{1}{2}\big(u \log(u) - (u+1)\log(\frac{u+1}{2})\big)$. In this case, we have

$$f'(u) = \frac{1}{2}\log\left(\frac{2u}{u+1}\right), \qquad f''(u) = \frac{1}{2u(u+1)}, \qquad [f']^{-1}(u) = \frac{\exp(2u)}{2 - \exp(2u)}.$$

The JS-softargmax operator specialize (with base measure $\nu_{\text{ref}} \in \mathcal{P}(\mathcal{A})$) to

$$\nu_x^{JS}(a) = \nu_{\text{ref}}(a)[f']^{-1}(x(a) - \mu_x) = \nu_{\text{ref}}(a)\frac{\exp(2(x(a) - \mu_x))}{2 - \exp(2(x(a) - \mu_x))}, \qquad a \in \mathcal{A},$$

where $\mu_x$ is the unique normalizer satisfying the constraint

$$\sum_{a \in \mathcal{A}} \nu_{\text{ref}}(a)\frac{\exp(2(x(a) - \mu_x))}{2 - \exp(2(x(a) - \mu_x))} = 1,$$

### B.3 DERIVATIVES OF THE POLICY

Next, we exploit the expression of the derivatives of the soft-$f$-argmax derived in Lemma B.4 to compute the first and second derivatives of the policy. We begin by extending the notations defined in (27), (28), and (29) to encompass a dependence on the state. For any pair $(s, a) \in \mathcal{S} \times \mathcal{A}$, define the first three derivatives of $f$ evaluated at the likelihood ratio $\pi_\theta^f(a|s)/\pi_{\text{ref}}(a|s)$:

$$f_\theta'(a|s) \; := \; f_{\theta(s,\cdot)}'(a) = f'\left(\frac{\pi_\theta^f(a|s)}{\pi_{\text{ref}}(a|s)}\right) \; , \tag{35}$$

$$f_\theta''(a|s) \; := \; f_{\theta(s,\cdot)}''(a) = f''\left(\frac{\pi_\theta^f(a|s)}{\pi_{\text{ref}}(a|s)}\right) \; , \tag{36}$$

$$f_\theta'''(a|s) \; := \; f_{\theta(s,\cdot)}'''(a) = f'''\left(\frac{\pi_\theta^f(a|s)}{\pi_{\text{ref}}(a|s)}\right) \; . \tag{37}$$

In addition, for every $s \in \mathcal{S}$ we introduce the quantities

$$\mathrm{W}_\theta^f(s) \; := \; \mathrm{W}_{\theta(s,\cdot)}^f = \sum_{a\in\mathcal{A}} \frac{\pi_{\text{ref}}(a|s)}{f_\theta''(a|s)}, \quad \text{and} \quad \mathrm{Y}_\theta^f(s) \; := \; \sum_{a\in\mathcal{A}} \frac{\pi_{\text{ref}}(a|s)}{f_\theta''(a|s)} \, |f_\theta'(a|s)| \; . \tag{38}$$

For every $(s, a) \in \mathcal{S} \times \mathcal{A}$, we define the normalized weights

$$\mathrm{w}_\theta^f(a|s) = \frac{1}{\mathrm{W}_\theta^f(s)} \frac{\pi_{\text{ref}}(a|s)}{f_\theta''(a|s)} \; . \tag{39}$$

The following lemma provides bounds on several key quantities that will appear in the appendix.

**Lemma B.6.** *Assume that, for some $\underline{\pi}_{\text{ref}} > 0$, $f$ and $\pi_{\text{ref}}$ satisfy $\mathbf{A}_f(\underline{\pi}_{\text{ref}})$ and $\mathbf{P}(\underline{\pi}_{\text{ref}})$, respectively. For any parameter $\theta \in \mathbb{R}^{|\mathcal{S}||\mathcal{A}|}$, it holds that*

$$\left\|\mathrm{W}_\theta^f\right\|_\infty \le \omega_f, \;\; \left\|\mathrm{Y}_\theta^f\right\|_\infty \le \mathrm{y}_f, \;\; \max_{s\in\mathcal{S}} \mathrm{D}^f(\pi_\theta^f(\cdot|s)\|\pi_{\text{ref}}(\cdot|s)) \le \mathrm{d}_f, \;\; \max_{(s,a)\in\mathcal{S}\times\mathcal{A}} \frac{\mathrm{w}_\theta^f(a|s)}{\pi_\theta^f(a|s)} \le \omega_f \; .$$

*Proof.* The proof follows from the definition of the different quantities and $\mathbf{A}_f(\underline{\pi}_{\text{ref}})$ $\qquad\square$

Using Lemma B.4, we get the following expression for the derivatives of the policy.

**Corollary B.7.** *Assume that, for some $\underline{\pi}_{\text{ref}} > 0$, $f$ and $\pi_{\text{ref}}$ satisfy $\mathbf{A}_f(\underline{\pi}_{\text{ref}})$ and $\mathbf{P}(\underline{\pi}_{\text{ref}})$, respectively. Then the policy $\pi_\theta^f$ is twice continuously differentiable with respect to $\theta$. Additionally, for all $\theta \in \mathbb{R}^{|\mathcal{S}||\mathcal{A}|}$ and $s \in \mathcal{S}$, there exists a unique $\mu_\theta(s) \in \mathbb{R}$ such that for any $a \in \mathcal{A}$, we have*

$$\pi_\theta^f(a|s) = \pi_{\text{ref}}(a|s)[f']^{-1}\left(\theta(s, a) - \mu_\theta(s)\right). \tag{40}$$

*For any $s \in \mathcal{S}$, the $\theta \mapsto \mu_\theta(s)$ is continuously differentiable. For any $s' \neq s$, $\partial/\partial\theta(s',\cdot)\mu_\theta(s) = 0$ and*

$$\frac{\partial\mu_\theta(s)}{\partial\theta(s,\cdot)} = \mathrm{w}_\theta^f(\cdot|s) \; .$$

*Moreover, for any $\theta \in \mathbb{R}^{|\mathcal{S}||\mathcal{A}|}$, $s \in \mathcal{S}$, and $(a, b) \in \mathcal{A} \times \mathcal{A}$, we have*

$$\frac{1}{\mathrm{W}_\theta^f(s)} \cdot \frac{\partial\pi_\theta^f(a|s)}{\partial\theta(s, b)} = 1_b(a)\,\mathrm{w}_\theta^f(a|s) - \mathrm{w}_\theta^f(a|s)\,\mathrm{w}_\theta^f(b|s) \; ,$$

*In addition, for any $(a, b, c) \in \mathcal{A}^3$, the second derivative satisfies*

$$\frac{1}{\mathrm{W}_\theta^f(s)} \cdot \frac{\partial\pi_\theta^f(a|s)}{\partial\theta(s,b)\partial\theta(s,c)} = -1_b(a)1_c(a)\frac{f_\theta'''(a|s)}{f_\theta''(a|s)^2} \cdot \mathrm{w}_\theta^f(a|s) + 1_c(b)\,\mathrm{w}_\theta^f(a|s)\,\mathrm{w}_\theta^f(b|s)\frac{f_\theta'''(b|s)}{f_\theta''(b|s)^2}$$

$$+ 1_b(a)\,\mathrm{w}_\theta^f(a|s)\,\mathrm{w}_\theta^f(c|s)\frac{f_\theta'''(a|s)}{f_\theta''(a|s)^2} + 1_c(a)\,\mathrm{w}_\theta^f(a|s)\,\mathrm{w}_\theta^f(b|s)\frac{f_\theta'''(a|s)}{f_\theta''(a|s)^2}$$

$$- w_\theta^f(a|s) \cdot w_\theta^f(b|s) \cdot w_\theta^f(c|s) \cdot \left[ \frac{f_\theta'''(a|s)}{f_\theta''(a|s)^2} + \frac{f_\theta'''(b|s)}{f_\theta''(b|s)^2} + \frac{f_\theta'''(c|s)}{f_\theta''(c|s)^2} \right]$$

$$+ w_\theta^f(a|s) \cdot w_\theta^f(b|s) \cdot w_\theta^f(c|s) \cdot \sum_{d \in \mathcal{A}} w_\theta^f(d|s) \cdot \frac{f_\theta'''(d|s)}{f_\theta''(d|s)^2} .$$

**Lemma B.8.** *Assume that, for some $\underline{\pi_{\mathrm{ref}}} > 0$, $f$ and $\pi_{\mathrm{ref}}$ satisfy $\mathbf{A}_f(\underline{\pi_{\mathrm{ref}}})$ and $\mathbf{P}(\underline{\pi_{\mathrm{ref}}})$, respectively. Then, it holds that*

$$\sum_{(a,b) \in \mathcal{A}^2} \left| \frac{\partial \pi_\theta^f(a|s)}{\partial \theta(s,b)} \right| \le 2\, W_\theta^f(s) \ , \qquad \sum_{(a,b,c) \in \mathcal{A}^3} \left| \frac{\partial^2 \pi_\theta^f(a|s)}{\partial \theta(s,b) \partial \theta(s,c)} \right| \le 8\kappa_f\, W_\theta^f(s) \ .$$

*Proof.* Using the expression of the derivative of the policy provided in Corollary B.7, we have by the triangle inequality

$$\sum_{b \in \mathcal{A}} \left| \frac{\partial \pi_\theta^f(a|s)}{\partial \theta(s,b)} \right| = \sum_{b \in \mathcal{A}} W_\theta^f(s)\, w_\theta^f(a|s) \left| \mathbf{1}_b(a) - w_\theta^f(b|s) \right| = 2\, W_\theta^f(s)\, w_\theta^f(a|s)(1 - w_\theta^f(a|s)) \ .$$

where we used that

$$\sum_{b \in \mathcal{A}} \left| \mathbf{1}_b(a) - w_\theta^f(b|s) \right| = 2(1 - w_\theta^f(a|s)) \ . \tag{41}$$

Hence, we have

$$\sum_{(a,b) \in \mathcal{A}^2} \left| \frac{\partial \pi_\theta^f(a|s)}{\partial \theta(s,b)} \right| \le 2\, W_\theta^f(s) \ .$$

Fix $a \in \mathcal{A}$. By using the expression of the second derivative of the policy provided in Corollary B.7 combined with the triangle inequality and (41), we get

$$\sum_{(b,c) \in \mathcal{A}^2} \left| \frac{\partial^2 \pi_\theta^f(a|s)}{\partial \theta(s,b) \partial \theta(s,c)} \right| \le 4\, W_\theta^f(s) \frac{f_\theta'''(a|s)}{f_\theta''(a|s)^2}\, w_\theta^f(a|s) + 4\, W_\theta^f(s)\, w_\theta^f(a|s) \sum_{b \in \mathcal{A}} \frac{f_\theta'''(b|s)}{f_\theta''(b|s)^2}\, w_\theta^f(b|s).$$

Next combining $\mathbf{A}_f(\underline{\pi_{\mathrm{ref}}})$ and (41), we obtain

$$\sum_{(b,c) \in \mathcal{A}^2} \left| \frac{\partial^2 \pi_\theta^f(a|s)}{\partial \theta(s,b) \partial \theta(s,c)} \right| \le 8\, W_\theta^f(s)\, w_\theta^f(a|s)\kappa_f \ .$$

Finally, summing over the actions concludes the proof. $\qquad\square$

### B.4 SMOOTHNESS OF OBJECTIVE

Firstly, we recall that the regularized value satisfies the following fixed-point equation Geist et al. (2019):

$$v_\pi^f(s) = \sum_{a \in \mathcal{A}} \pi(a|s) r(s,a) - \lambda\, D^f(\pi(\cdot|s) \| \pi_{\mathrm{ref}}(\cdot|s)) + \gamma \sum_{(a,s') \in \mathcal{A} \times \mathcal{S}} \pi(a|s) P(s'|s,a)\, v_\pi^f(s') \ . \tag{42}$$

In order to prove the smoothness of the objective, we will prove that the all the second-order directional derivatives are bounded. Denote $\theta_\alpha = \theta + \alpha u$ where $\alpha \in \mathbb{R}$ and $u \in \mathbb{R}^{|\mathcal{S}||\mathcal{A}|}$. By Equation (42), for any $s \in \mathcal{S}$ it holds that

$$v_{\theta_\alpha}^f(s) = \mathbf{e}_s^\top M(\alpha) r_{\theta_\alpha}^f \ , \tag{43}$$

where $M(\alpha)$ is a matrix of $\mathbb{R}^{\mathcal{S} \times \mathcal{S}}$ defined by

$$M(\alpha) = (\mathrm{Id} - \gamma \mathsf{P}_{\theta_\alpha})^{-1} \ ,$$

and where $r^f_{\theta_\alpha}$, and $\mathsf{P}_{\theta_\alpha}$ are defined in Section 2. Taking the derivative of (43) with respect to $\alpha$ yields

$$\frac{\partial v^f_{\theta_\alpha}(s)}{\partial \alpha} = \gamma \mathbf{e}_s^\top M(\alpha) \frac{\partial \mathsf{P}_{\theta_\alpha}}{\partial \alpha} M(\alpha) r^f_{\theta_\alpha} + \mathbf{e}_s^\top M(\alpha) \frac{\partial r^f_{\theta_\alpha}}{\partial \alpha} \ .$$

Taking the derivative of the preceding equation with respect to $\alpha$ gives

$$\frac{\partial^2 v^f_{\theta_\alpha}(s)}{\partial \alpha^2} = 2\gamma^2 \mathbf{e}_s^\top M(\alpha) \frac{\partial \mathsf{P}_{\theta_\alpha}}{\partial \alpha} M(\alpha) \frac{\partial \mathsf{P}_{\theta_\alpha}}{\partial \alpha} M(\alpha) r^f_{\theta_\alpha} + \gamma \mathbf{e}_s^\top M(\alpha) \frac{\partial^2 \mathsf{P}_{\theta_\alpha}}{\partial^2 \alpha} M(\alpha) r^f_{\theta_\alpha}$$
$$+ 2\gamma \mathbf{e}_s^\top M(\alpha) \frac{\partial \mathsf{P}_{\theta_\alpha}}{\partial \alpha} M(\alpha) \frac{\partial r^f_{\theta_\alpha}}{\partial \alpha} + \mathbf{e}_s^\top M(\alpha) \frac{\partial^2 r^f_{\theta_\alpha}}{\partial^2 \alpha} \ . \tag{44}$$

In order to control the second-order directional derivative of the regularised value function, we establish first several properties of the quantities that appear in the preceding equality.

**Lemma B.9.** *Assume that, for some $\underline{\pi_{\mathrm{ref}}} > 0$, $f$ and $\pi_{\mathrm{ref}}$ satisfy $\mathbf{A}_f(\pi_{\mathrm{ref}})$ and $\mathbf{P}(\pi_{\mathrm{ref}})$, respectively. We have*

$$\left\| \left. \frac{\partial \mathsf{P}_{\theta_\alpha}}{\partial \alpha} \right|_{\alpha=0} \right\|_\infty \leq 2 \max_s \{ \mathrm{W}^f_\theta(s) \} \, \|u\|_2 \ ,$$

*Similarly, we have*

$$\left\| \left. \frac{\partial^2 \mathsf{P}_{\theta_\alpha}}{\partial^2 \alpha} \right|_{\alpha=0} \right\|_\infty \leq 8 \kappa_f \max_s \{ \mathrm{W}^f_\theta(s) \} \, \|u\|_2^2 \ .$$

*Proof.* **Bounding the first-order directional derivative.** The derivative with respect to $\alpha$ is

$$\left[ \left. \frac{\partial \mathsf{P}_{\theta_\alpha}}{\partial \alpha} \right|_{\alpha=0} \right]_{s,s'} = \sum_{a \in \mathcal{A}} \left[ \left. \frac{\partial \pi^f_{\theta_\alpha}(a|s)}{\partial \alpha} \right|_{\alpha=0} \right] \mathsf{P}(s'|s,a) \ .$$

Fix $s \in \mathcal{S}$. Because $\pi^f_{\theta_\alpha}(a|s)$ depends only on $\theta(s,\cdot)$, by the chain rule

$$\sum_{a \in \mathcal{A}} \left| \left. \frac{\partial \pi^f_{\theta_\alpha}(a|s)}{\partial \alpha} \right|_{\alpha=0} \right| = \sum_{a \in \mathcal{A}} \left| \langle \frac{\partial \pi^f_\theta(a|s)}{\partial \theta(s,\cdot)}, u(s,\cdot) \rangle \right| \leq \sum_{a \in \mathcal{A}} \left\| \frac{\partial \pi^f_\theta(a|s)}{\partial \theta(s,\cdot)} \right\|_1 \|u(s,\cdot)\|_2 \ ,$$

where in the last inequality we used Cauchy-Schwarz inequality and the fact that the $L_1$ norm dominates the $L_2$ norm. Now using Lemma B.8, we get that

$$\sum_{a \in \mathcal{A}} \left\| \frac{\partial \pi^f_\theta(a|s)}{\partial \theta(s,\cdot)} \right\|_1 \|u(s,\cdot)\|_2 \leq 2 \|u\|_2 \, \mathrm{W}^f_\theta(s) \ .$$

**Bounding the second-order directional derivative.** Similarly, taking the second derivative with respect to $\alpha$ yields

$$\left[ \left. \frac{\partial^2 \mathsf{P}_{\theta_\alpha}}{\partial^2 \alpha} \right|_{\alpha=0} \right]_{s,s'} = \sum_{a \in \mathcal{A}} \left[ \left. \frac{\partial^2 \pi^f_{\theta_\alpha}(a|s)}{\partial^2 \alpha} \right|_{\alpha=0} \right] \mathsf{P}(s'|s,a) \ .$$

Fix $s \in \mathcal{S}$. It holds that

$$\sum_{a \in \mathcal{A}} \left| \left. \frac{\partial^2 \pi^f_{\theta_\alpha}(a|s)}{\partial^2 \alpha} \right|_{\alpha=0} \right| = \sum_{a \in \mathcal{A}} \left| \langle \frac{\partial^2 \pi^f_\theta(a|s)}{\partial^2 \theta(s,\cdot)} u(s,\cdot), u(s,\cdot) \rangle \right| \leq \sum_{a \in \mathcal{A}} \left\| \frac{\partial^2 \pi^f_\theta(a|s)}{\partial^2 \theta(s,\cdot)} \right\|_2 \|u(s,\cdot)\|_2^2 \ ,$$

where in the last inequality, we used the Cauchy-Schwarz inequality and the definition of the matrix operator norm. Additionally, using that for any matrix $A \in \mathbb{R}^{d \times d}$, we have

$$\|A\|_2 \leq \sum_{i=1}^d \sum_{j=1}^d |a_{i,j}|$$

combined with Lemma B.8, we get that

$$\sum_{a \in \mathcal{A}} \left| \left. \frac{\partial^2 \pi^f_{\theta_\alpha}(a|s)}{\partial^2 \alpha} \right|_{\alpha=0} \right| \leq 8 \|u(s,\cdot)\|_2^2 \, \kappa_f \max_{s \in \mathcal{S}} \mathrm{W}^f_\theta(s) \ ,$$

which concludes the proof. $\qquad\square$

**Lemma B.10.** *Assume that, for some $\pi_{\text{ref}} > 0$, $f$ and $\pi_{\text{ref}}$ satisfy $\boldsymbol{A}_f(\pi_{\text{ref}})$ and $\boldsymbol{P}(\pi_{\text{ref}})$, respectively. Then, the regularized reward satisfies*

$$\|\mathsf{r}_\theta^f\|_\infty \le 1 + \lambda \max_{s \in \mathcal{S}} \mathrm{D}^f(\pi_\theta^f(\cdot|s)\|\pi_{\text{ref}}(\cdot|s)) \ .$$

*Additionally, we have that*

$$\left\|\left.\frac{\partial \mathsf{r}_{\theta_\alpha}^f}{\partial \alpha}\right|_{\alpha=0}\right\|_\infty \le 2 \max_{s \in \mathcal{S}} \left\{ \mathrm{W}_\theta^f(s) + \lambda \, \mathrm{Y}_\theta^f(s) \right\} \|u\|_2 \ ,$$

*and that*

$$\left\|\left.\frac{\partial^2 \mathsf{r}_{\theta_\alpha}^f}{\partial^2 \alpha}\right|_{\alpha=0}\right\|_\infty \le \max_{s \in \mathcal{S}} \left\{ 4(2\kappa_f + \lambda) \, \mathrm{W}_\theta^f(s) + 8\lambda\kappa_f \, \mathrm{Y}_\theta^f(s) \right\} \|u\|_2^2 \ .$$

*Proof.* The bound on $\|\mathsf{r}_\theta^f\|_\infty$ is immediate.

**Bounding the first derivative.** It holds that

$$\left\|\left.\frac{\partial \mathsf{r}_{\theta_\alpha}^f}{\partial \alpha}\right|_{\alpha=0}\right\|_\infty = \max_{s \in \mathcal{S}} \left|\left\langle \frac{\partial \mathsf{r}_\theta^f(s)}{\partial \theta}, u \right\rangle\right| = \max_{s \in \mathcal{S}} \left|\left\langle \frac{\partial \mathsf{r}_\theta^f(s)}{\partial \theta(s,\cdot)}, u(s,\cdot) \right\rangle\right| \le \max_{s \in \mathcal{S}} \left\|\frac{\partial \mathsf{r}_\theta^f(s)}{\partial \theta(s,\cdot)}\right\|_1 \|u\|_\infty \ .$$

Computing the derivative of $\mathsf{r}_\theta^f(s)$ with respect to $\theta(s,b)$ yields

$$\frac{\partial \mathsf{r}_\theta^f(s)}{\partial \theta(s,b)} = \sum_{a \in \mathcal{A}} \frac{\partial \pi_\theta^f(a|s)}{\partial \theta(s,b)} \mathsf{r}(s,a) - \lambda \frac{\partial \pi_\theta^f(a|s)}{\partial \theta(s,b)} f'\left(\frac{\pi_\theta^f(a|s)}{\pi_{\text{ref}}(a|s)}\right) \ .$$

Plugging in the expression of the derivative of the policy of Corollary B.7 in the preceding identity yields

$$\frac{\partial \tilde{\mathsf{r}}_\theta(s)}{\partial \theta(s,b)} = \frac{\pi_{\text{ref}}(b|s)}{f_\theta''(b|s)} \mathsf{r}(s,b) - \lambda \frac{\pi_{\text{ref}}(b|s)}{f_\theta''(b|s)} f_\theta'(b|s) - \sum_{a \in \mathcal{A}} \frac{\pi_{\text{ref}}(a|s)\pi_{\text{ref}}(b|s)}{f_\theta''(a|s) f_\theta''(b|s)} \cdot \frac{\mathsf{r}(s,a)}{\mathrm{W}_\theta^f(s)}$$

$$+ \lambda \frac{1}{\mathrm{W}_\theta^f(s)} \sum_{a \in \mathcal{A}} \frac{\pi_{\text{ref}}(a|s)\pi_{\text{ref}}(b|s)}{f_\theta''(a|s) f_\theta''(b|s)} f_\theta'(a|s) \ . \tag{45}$$

Taking the absolute value, applying the triangle inequality, and using that the rewards are bounded by 1 gives

$$\sum_{b \in \mathcal{A}} \left|\frac{\partial \mathsf{r}_\theta^f(s)}{\partial \theta(s,b)}\right| \le 2 \, \mathrm{W}_\theta^f(s) + 2\lambda \, \mathrm{Y}_\theta^f(s) \ .$$

**Bounding the second derivative.** It holds that

$$\left\|\left.\frac{\partial^2 \mathsf{r}_{\theta_\alpha}^f}{\partial \alpha^2}\right|_{\alpha=0}\right\|_\infty = \max_{s \in \mathcal{S}} \left|\left.\frac{\partial^2 \mathsf{r}_{\theta_\alpha}^f(s)}{\partial \alpha^2}\right|_{\alpha=0}\right| \le \max_{s \in \mathcal{S}} \left\|\frac{\partial^2 \mathsf{r}_\theta^f(s)}{\partial \theta(s,\cdot)^2}\right\|_2 \|u\|_2^2 \ ,$$

where in the last inequality, we used the Cauchy-Schwarz inequality and the definition of the matrix operator norm.. We now compute the second derivative of $\mathsf{r}_\theta^f(s)$. Starting from (45), we get

$$\frac{\partial^2 \mathsf{r}_\theta^f(s)}{\partial \theta(s,b)\partial \theta(s,c)} = -\frac{\pi_{\text{ref}}(b|s)}{f_\theta''(b|s)^2} \left[ \mathbb{1}_c(b) \frac{f_\theta'''(b|s)}{f_\theta''(b|s)} - \frac{\pi_{\text{ref}}(c|s) f_\theta'''(b|s)}{f_\theta''(b|s) f_\theta''(c|s)} \cdot \frac{1}{\mathrm{W}_\theta^f(s)} \right] \mathsf{r}(s,b)$$

$$+ \lambda \frac{\pi_{\text{ref}}(b|s) f_\theta'(b|s)}{f_\theta''(b|s)^2} \left[ \mathbb{1}_c(b) \frac{f_\theta''(b|s)}{f_\theta''(b|s)} - \frac{\pi_{\text{ref}}(c|s) f_\theta'''(b|s)}{f_\theta''(b|s) f_\theta''(c|s)} \cdot \frac{1}{\mathrm{W}_\theta^f(s)} \right]$$

$$- \lambda \left[ \mathbb{1}_b(c) \frac{\pi_{\text{ref}}(b|s)}{f_\theta''(b|s)} - \frac{\pi_{\text{ref}}(b|s)\pi_{\text{ref}}(c|s)}{f_\theta''(b|s) f_\theta''(c|s)} \cdot \frac{1}{\mathrm{W}_\theta^f(s)} \right]$$

$$+ \sum_{a \in \mathcal{A}} \frac{\pi_{\mathrm{ref}}(a|s)\pi_{\mathrm{ref}}(b|s)}{f''_\theta(a|s)^2 f''_\theta(b|s)} \cdot \frac{\mathsf{r}(s,a)}{\mathrm{W}^f_\theta(s)} \left[ 1_c(a) \frac{f'''_\theta(a|s)}{f''_\theta(a|s)} - \frac{\pi_{\mathrm{ref}}(c|s) f'''_\theta(a|s)}{f''_\theta(a|s) f''_\theta(c|s)} \cdot \frac{1}{\mathrm{W}^f_\theta(s)} \right]$$

$$+ \sum_{a \in \mathcal{A}} \frac{\pi_{\mathrm{ref}}(a|s)\pi_{\mathrm{ref}}(b|s)}{f''_\theta(a|s) f''_\theta(b|s)^2} \cdot \frac{\mathsf{r}(s,a)}{\mathrm{W}^f_\theta(s)} \left[ 1_c(b) \frac{f'''_\theta(b|s)}{f''_\theta(b|s)} - \frac{\pi_{\mathrm{ref}}(c|s) f'''_\theta(b|s)}{f''_\theta(b|s) f''_\theta(c|s)} \cdot \frac{1}{\mathrm{W}^f_\theta(s)} \right]$$

$$+ \sum_{a \in \mathcal{A}} \frac{\pi_{\mathrm{ref}}(a|s)\pi_{\mathrm{ref}}(b|s)}{f''_\theta(a|s) f''_\theta(b|s)} \frac{\mathsf{r}(s,a)}{\mathrm{W}^f_\theta(s)^2} \left[ \frac{-\pi_{\mathrm{ref}}(c|s) f'''_\theta(c|s)}{f''_\theta(c|s)^3} + \sum_{d \in \mathcal{A}} \frac{\pi_{\mathrm{ref}}(c|s)\pi_{\mathrm{ref}}(d|s) f'''_\theta(d|s)}{\mathrm{W}^f_\theta(s) f''_\theta(d|s)^3 f''_\theta(c|s)} \right]$$

$$- \lambda \sum_{a \in \mathcal{A}} \frac{\pi_{\mathrm{ref}}(a|s)\pi_{\mathrm{ref}}(b|s)}{f''_\theta(a|s) f''_\theta(b|s)} \frac{f'_\theta(b|s)}{\mathrm{W}^f_\theta(s)^2} \left[ \frac{-\pi_{\mathrm{ref}}(c|s) f'''_\theta(c|s)}{f''_\theta(c|s)^3} + \sum_{d \in \mathcal{A}} \frac{\pi_{\mathrm{ref}}(c|s)\pi_{\mathrm{ref}}(d|s) f'''_\theta(d|s)}{\mathrm{W}^f_\theta(s) f''_\theta(d|s)^3 f''_\theta(c|s)} \right]$$

$$- \lambda \sum_{a \in \mathcal{A}} \frac{\pi_{\mathrm{ref}}(a|s)\pi_{\mathrm{ref}}(b|s)}{f''_\theta(a|s)^2 f''_\theta(b|s)} \frac{f'_\theta(b|s)}{\mathrm{W}^f_\theta(s)} \left[ 1_c(a) \frac{f'''_\theta(a|s)}{f''_\theta(a|s)} - \frac{\pi_{\mathrm{ref}}(c|s) f'''_\theta(a|s)}{f''_\theta(a|s) f''_\theta(c|s)} \cdot \frac{1}{\mathrm{W}^f_\theta(s)} \right] \cdot$$

$$- \lambda \sum_{a \in \mathcal{A}} \frac{\pi_{\mathrm{ref}}(a|s)\pi_{\mathrm{ref}}(b|s)}{f''_\theta(a|s) f''_\theta(b|s)^2} \frac{f'_\theta(b|s)}{\mathrm{W}^f_\theta(s)} \left[ 1_c(b) \frac{f'''_\theta(b|s)}{f''_\theta(b|s)} - \frac{\pi_{\mathrm{ref}}(c|s) f'''_\theta(b|s)}{f''_\theta(b|s) f''_\theta(c|s)} \cdot \frac{1}{\mathrm{W}^f_\theta(s)} \right] \cdot$$

$$+ \lambda \sum_{a \in \mathcal{A}} \frac{\pi_{\mathrm{ref}}(a|s)}{f''_\theta(a|s)} \frac{1}{\mathrm{W}^f_\theta(s)} \left[ 1_b(c) \frac{\pi_{\mathrm{ref}}(b|s)}{f''_\theta(b|s)} - \frac{\pi_{\mathrm{ref}}(b|s)\pi_{\mathrm{ref}}(c|s)}{f''_\theta(b|s) f''_\theta(c|s)} \cdot \frac{1}{\mathrm{W}^f_\theta(s)} \right] \cdot$$

Taking the absolute value, applying the triangle inequality, using that under $\mathbf{A}_f(\pi_{\mathrm{ref}})$, for all $x \in \mathbb{R}+$, we have $|f'''(x)/f''(x)^2| \le \kappa_f$, and using that the rewards are bounded by 1 gives

$$\sum_{b \in \mathcal{A}} \sum_{c \in \mathcal{A}} \left| \frac{\partial^2 \mathsf{r}^f_{\theta_0}(s)}{\partial\theta(s,b)\partial\theta(s,c)} \right| \le 8\kappa_f \, \mathrm{W}^f_\theta(s) + 8\lambda\kappa_f \, \mathrm{Y}^f_\theta(s) + 4\lambda \, \mathrm{W}^f_\theta(s) \;,$$

which concludes the proof. $\qquad\qquad\square$

**Lemma B.11.** *Assume that, for some $\pi_{\mathrm{ref}} > 0$, $f$ and $\pi_{\mathrm{ref}}$ satisfy $\mathbf{A}_f(\pi_{\mathrm{ref}})$ and $\mathbf{P}(\pi_{\mathrm{ref}})$ respectively. Then, for any $\theta \in \mathbb{R}^{|\mathcal{S}||\mathcal{A}|}$ and $u \in \mathbb{R}^{|\mathcal{S}||\mathcal{A}|}$,*

$$\left| u^\top \frac{\partial^2 v^f_\theta(s)}{\partial\theta^2} u \right| \le \left( \sum_{i=1}^{3} \frac{L^{(i)}_{\lambda,f}(\theta)}{(1-\gamma)^i} \right) \|u\|^2_2 \;,$$

*where for $i \in \{1,2,3\}$, $L^{(i)}_{\lambda,f}(\theta)$, are defined as*

$$L^{(1)}_{\lambda,f}(\theta) := 4(2\kappa_f + \lambda) \left\| \mathrm{W}^f_\theta \right\|_\infty + 8\lambda\kappa_f \left\| \mathrm{Y}^f_\theta \right\|_\infty \;,$$

$$L^{(2)}_{\lambda,f}(\theta) := 8\gamma \left\| \mathrm{W}^f_\theta \right\|_\infty \left( \kappa_f \{1 + \lambda \max_{s \in \mathcal{S}} \mathrm{D}^f(\pi^f_\theta(\cdot|s)\|\pi_{\mathrm{ref}}(\cdot|s))\} + \left\| \mathrm{W}^f_\theta \right\|_\infty + \lambda \left\| \mathrm{Y}^f_\theta \right\|_\infty \right) \;,$$

$$L^{(3)}_{\lambda,f}(\theta) := 8\gamma^2 \left\| \mathrm{W}^f_\theta \right\|^2_\infty (1 + \lambda \max_{s \in \mathcal{S}} \mathrm{D}^f(\pi^f_\theta(\cdot|s)\|\pi_{\mathrm{ref}}(\cdot|s)) \;.$$

*Proof.* By construction, we get

$$\left| u^\top \frac{\partial^2 v^f_\theta(s)}{\partial\theta^2} u \right| = \left| \frac{\partial^2 v^f_{\theta_\alpha}(s)}{\partial\alpha^2} \right|_{\alpha=0} \Bigg| \;.$$

Using (44), we get that

$$\left| \frac{\partial^2 v^f_{\theta_\alpha}(s)}{\partial\alpha^2} \right|_{\alpha=0} \le \underbrace{\left| 2\gamma^2 \mathbf{e}^\top_s M(0) \frac{\partial \mathsf{P}_{\theta_\alpha}}{\partial\alpha} \right|_{\alpha=0} M(0) \frac{\partial \mathsf{P}_{\theta_\alpha}}{\partial\alpha} \Bigg|_{\alpha=0} M(0) \mathsf{r}^f_\theta \right|}_{(\mathbf{A})}$$

$$+ \underbrace{\left| \gamma \mathbf{e}_s^\top M(0) \frac{\partial^2 \mathsf{P}_{\theta_\alpha}}{\partial^2 \alpha}\bigg|_{\alpha=0} M(0) \mathsf{r}_\theta^f \right|}_{(\mathbf{B})} + \underbrace{\left| 2\gamma \mathbf{e}_s^\top M(0) \frac{\partial \mathsf{P}_{\theta_\alpha}}{\partial \alpha}\bigg|_{\alpha=0} M(0) \frac{\partial \mathsf{r}_{\theta_\alpha}^f}{\partial \alpha}\bigg|_{\alpha=0} \right|}_{(\mathbf{C})}$$

$$+ \underbrace{\left| \mathbf{e}_s^\top M(0) \frac{\partial^2 \mathsf{r}_{\theta_\alpha}^f}{\partial^2 \alpha}\bigg|_{\alpha=0} \right|}_{(\mathbf{D})} \quad .$$

We now bound each of these terms separately

**Bounding (A).** First note that, for any vector $x \in \mathbb{R}^{\mathcal{S}}$ and $\alpha \in \mathbb{R}$, we have

$$\|M(\alpha)x\|_\infty \le \frac{1}{1-\gamma} \|x\|_\infty \quad , \tag{46}$$

This yields

$$(\mathbf{A}) \le 2\gamma^2 \left\| M(0) \frac{\partial \mathsf{P}_{\theta_\alpha}}{\partial \alpha}\bigg|_{\alpha=0} M(0) \frac{\partial \mathsf{P}_{\theta_\alpha}}{\partial \alpha}\bigg|_{\alpha=0} M(0) \mathsf{r}_\theta^f \right\|_\infty \le \frac{2\gamma^2}{(1-\gamma)^3} \left\| \frac{\partial \mathsf{P}_{\theta_\alpha}}{\partial \alpha}\bigg|_{\alpha=0} \right\|_\infty^2 \|\mathsf{r}_\theta^f\|_\infty$$

By using again (46), Lemma B.9, and Lemma B.10 we get

$$(\mathbf{A}) \le \frac{8\gamma^2 \max_s \{\mathsf{W}_\theta^f(s)\}^2 \|u\|_2^2}{(1-\gamma)^3} (1 + \lambda \max_{s \in \mathcal{S}} \mathsf{D}^f(\pi_\theta^f(\cdot|s)\|\pi_{\mathrm{ref}}(\cdot|s))) \quad .$$

**Bounding (B).** Using (46), Lemma B.9, and Lemma B.10 we get

$$(\mathbf{B}) \le \frac{\gamma}{(1-\gamma)^2} \left\| \frac{\partial^2 \mathsf{P}_{\theta_\alpha}}{\partial^2 \alpha}\bigg|_{\alpha=0} \right\|_\infty \|\mathsf{r}_\theta^f\|_\infty$$

$$\le \frac{8\gamma}{(1-\gamma)^2} \kappa_f \max_s \{\mathsf{W}_\theta^f(s)\} \|u\|_2^2 \{1 + \lambda \max_{s \in \mathcal{S}} \mathsf{D}^f(\pi_\theta^f(\cdot|s)\|\pi_{\mathrm{ref}}(\cdot|s))\} \quad .$$

**Bounding (C).** Similarly, using (46), Lemma B.9, and Lemma B.10 we get

$$(\mathbf{C}) \le \frac{8\gamma}{(1-\gamma)^2} \max_{s \in \mathcal{S}} \{\mathsf{W}_\theta^f(s)\} \max_{s \in \mathcal{S}} \left\{ \mathsf{W}_\theta^f(s) + \lambda \, \mathsf{Y}_\theta^f(s) \right\} \|u\|_2^2 \quad .$$

**Bounding (D).** Using (46), Lemma B.9, and Lemma B.10 we get

$$(\mathbf{D}) \le \frac{1}{1-\gamma} \left\| \frac{\partial^2 \mathsf{r}_{\theta_\alpha}^f}{\partial^2 \alpha}\bigg|_{\alpha=0} \right\|_\infty \le \frac{1}{1-\gamma} \max_{s \in \mathcal{S}} \left\{ 4(2\kappa_f + \lambda) \, \mathsf{W}_\theta^f(s) + 8\lambda \kappa_f \, \mathsf{Y}_\theta^f(s) \right\} \|u\|_2^2 \quad .$$

The proof is completed by collecting these upper bounds. $\quad\square$

**Theorem B.12.** *Assume that, for some $\underline{\pi_{\mathrm{ref}}} > 0$, $f$ and $\pi_{\mathrm{ref}}$ satisfy $\mathbf{A}_f(\underline{\pi_{\mathrm{ref}}})$ and $\mathbf{P}(\underline{\pi_{\mathrm{ref}}})$ respectively. Then for any $\theta, \theta' \in \mathbb{R}^{|\mathcal{S}||\mathcal{A}|}$, it holds that*

$$\left| v_{\theta'}^f(\rho) - v_\theta^f(\rho) - \langle \frac{\partial v_\theta^f(\rho)}{\partial \theta}, \theta' - \theta \rangle \right| \le \frac{L_f}{2} \|\theta' - \theta\|_2^2 \quad .$$

*where*

$$L_f := \frac{8\omega_f \left(\gamma\omega_f + (1-\gamma)\kappa_f\right)}{(1-\gamma)^3} + 4\lambda \frac{2\gamma^2\omega_f^2 \mathrm{d}_f + 2\gamma(1-\gamma)\omega_f \left[\kappa_f \mathrm{d}_f + \mathrm{y}_f\right] + (1-\gamma)^2 \left[\omega_f + 2\kappa_f \mathrm{y}_f\right]}{(1-\gamma)^3} .$$

*Proof.* Fix any vector $u \in \mathbb{R}^{|\mathcal{S}||\mathcal{A}|}$ and $\theta \in \mathbb{R}^{|\mathcal{S}||\mathcal{A}|}$. Using Lemma B.11, it holds that

$$\left| u^\top \frac{\partial^2 v_\theta^f(s)}{\partial \theta^2} u \right| \le \left( \sum_{i=1}^3 \frac{L_{\lambda,f}^{(i)}(\theta)}{(1-\gamma)^i} \right) \|u\|_2^2 \quad ,$$

where for $i \in \{1, 2, 3\}$, $L_{\lambda,f}^{(i)}(\theta)$, are defined as

$$L_{\lambda,f}^{(1)}(\theta) := 4(2\kappa_f + \lambda) \left\| W_\theta^f \right\|_\infty + 8\lambda\kappa_f \left\| Y_\theta^f \right\|_\infty ,$$

$$L_{\lambda,f}^{(2)}(\theta) := 8\gamma \left\| W_\theta^f \right\|_\infty \left( \kappa_f \{1 + \lambda \max_{s \in \mathcal{S}} D^f(\pi_\theta^f(\cdot|s) \| \pi_{\mathrm{ref}}(\cdot|s))\} + \left\| W_\theta^f \right\|_\infty + \lambda \left\| Y_\theta^f \right\|_\infty \right) ,$$

$$L_{\lambda,f}^{(3)}(\theta) := 8\gamma^2 \left\| W_\theta^f \right\|_\infty^2 (1 + \lambda \max_{s \in \mathcal{S}} D^f(\pi_\theta^f(\cdot|s) \| \pi_{\mathrm{ref}}(\cdot|s)) .$$

Using Lemma B.6 combined with Lemma I.1 concludes the proof. $\qquad \square$

## C  NON-UNIFORM ŁOJASIEWICZ INEQUALITY

Firstly, define respectively $q_\theta^f$ and $d_\rho^\theta$ as the regularized Q-function and discounted state visitation associated with the policy $\pi_\theta^f$, i.e.

$$q_\theta^f(s, a) = r(s, a) + \gamma \sum_{s' \in \mathcal{S}} P(s'|s, a) v_\theta^f(s') , \tag{47}$$

$$d_\rho^\theta(s) = (1 - \gamma) \sum_{t=0}^{\infty} \gamma^t \rho P_{\pi_\theta^f}^t(s) . \tag{48}$$

The goal of this section is to prove that the global objective satisfies a non-uniform Łojasiewicz inequality, i.e we aim to show the following theorem

**Theorem C.1.** *Assume that, for some $\pi_{\mathrm{ref}} > 0$, $f$ and $\pi_{\mathrm{ref}}$ satisfy $A_f(\pi_{\mathrm{ref}})$ and $P(\pi_{\mathrm{ref}})$ respectively. Assume in addition that the initial distribution $\rho$ satisfy $A_\rho$. Then, it holds that*

$$\left\| \frac{\partial v_\theta^f(\rho)}{\partial \theta} \right\|_2^2 \geq \mu_f(\theta) \left( v_\star^f(\rho) - v_\theta^f(\rho) \right) ,$$

*where*

$$\mu_f(\theta) := \frac{\lambda(1 - \gamma)\rho_{\min}^2 \zeta_f^2}{\omega_f^2} \min_{(s,a) \in \mathcal{S} \times \mathcal{A}} w_\theta^f(a|s)^2 .$$

One of the main challenges in establishing such an inequality lies in connecting global information (the suboptimality gap) to local information (the gradient norm). Recall from Section 2 that if

$$\theta(s, a) = q_\star^f(s, a)/\lambda, \quad \forall a \in \mathcal{A},$$

then $\pi_\theta^f = \pi_\star^f$. This observation highlights that, under this parameterization and regularization, the key quantity is the closeness between $\theta$ and $q_\theta^f/\lambda$. Formally, we will show that both the suboptimality gap and the gradient norm can be upper and lower bounded, respectively, by a quantity proportional to $\|\zeta_\theta(s)\|_2$, where we define

$$\zeta_\theta^f(s) := q_\theta^f(s, \cdot)/\lambda - \theta(s, \cdot) - K_\theta^f(s) 1_{|\mathcal{A}|}, \tag{49}$$

$$K_\theta^f(s) := \frac{\langle q_\theta^f(s, \cdot)/\lambda - \theta(s, \cdot), 1_{|\mathcal{A}|} \rangle}{|\mathcal{A}|} . \tag{50}$$

Note that $\zeta_\theta^f$ is the projection of $q_\theta^f(s, \cdot)/\lambda - \theta(s, \cdot)$ onto the subspace orthogonal to $1_{|\mathcal{A}|}$.

The proof proceeds in three steps:

1. Derive an explicit expression for the gradient of the objective and establish a lower bound in terms of $\|\zeta_\theta^f\|$.

2. Upper bound the suboptimality gap by a quantity directly related to $\|\zeta_\theta^f\|$.

3. Combine these two bounds to identify the corresponding non-uniform PL coefficient.

We now detail each step in turn.

### C.1 LOWER BOUNDING THE NORM OF THE GRADIENT

Before deriving a lower bound on the norm of the gradient, we start by deriving an expression for the latter.

**Lemma C.2.** *Assume that, for some $\underline{\pi_{\mathrm{ref}}} > 0$, $f$ and $\pi_{\mathrm{ref}}$ satisfy $\boldsymbol{A}_f(\underline{\pi_{\mathrm{ref}}})$ and $\boldsymbol{P}(\underline{\pi_{\mathrm{ref}}})$ respectively. For any $s \in \mathcal{S}$ and $b \in \mathcal{A}$, we have*

$$
\frac{1}{\mathrm{W}_\theta^f(s)} \frac{\partial v_\theta^f(\rho)}{\partial \theta(s,b)} = \frac{d_\rho^\theta(s)}{1-\gamma} \mathrm{w}_\theta^f(b|s) \left[ q_\theta^f(s,b) - \lambda\theta(s,b) - \sum_{a \in \mathcal{A}} \mathrm{w}_\theta^f(a|s) \left[ q_\theta^f(s,a) - \lambda\theta(s,a) \right] \right] .
$$

*Proof.* Fix $s \in \mathcal{S}$ and $b \in \mathcal{A}$. Additionally fix any $\tilde{s} \in \mathcal{S}$. Using (42), we have

$$
v_\theta^f(\tilde{s}) = \sum_{a \in \mathcal{A}} \pi_\theta^f(a|\tilde{s}) \mathrm{r}(\tilde{s},a) - \lambda \mathrm{D}^f(\pi_\theta^f(\cdot|\tilde{s}) \| \pi_{\mathrm{ref}}(\cdot|\tilde{s})) + \gamma \sum_{a \in \mathcal{A}} \sum_{\tilde{s}' \in \mathcal{S}} \pi_\theta^f(a|\tilde{s}) \mathrm{P}(\tilde{s}'|\tilde{s},a) v_\theta^f(\tilde{s}')
$$

Deriving the preceding recursion with respect to $\theta(s,b)$ yields

$$
\frac{\partial v_\theta^f(\tilde{s})}{\partial \theta(s,b)} = \underbrace{\sum_{a \in \mathcal{A}} \frac{\partial \pi_\theta^f(a|\tilde{s})}{\partial \theta(s,b)} \left[ \mathrm{r}(\tilde{s},a) - \lambda f'\left( \frac{\pi_\theta^f(a|\tilde{s})}{\pi_{\mathrm{ref}}(a|\tilde{s})} \right) + \gamma \sum_{\tilde{s}' \in \mathcal{S}} \mathrm{P}(\tilde{s}'|\tilde{s},a) v_\theta^f(\tilde{s}') \right]}_{Z(\tilde{s})}
$$
$$
+ \gamma \sum_{a \in \mathcal{A}} \sum_{\tilde{s}' \in \mathcal{S}} \pi_\theta^f(a|\tilde{s}) \mathrm{P}(\tilde{s}'|\tilde{s},a) \frac{\partial v_\theta^f(\tilde{s}')}{\partial \theta(s,b)} .
$$

Using the definition of the regularized Q-function and writing the preceding recursion in a vector form yields

$$
\frac{\partial v_\theta^f(\cdot)}{\partial \theta(s,b)} = Z(\cdot) + \gamma \mathrm{P}_\theta \frac{\partial v_\theta^f(\cdot)}{\partial \theta(s,b)} .
$$

which implies

$$
\rho^\top \frac{\partial v_\theta^f(\cdot)}{\partial \theta(s,b)} = \frac{\partial v_\theta^f(\rho)}{\partial \theta(s,b)} = \rho^\top (\mathrm{Id} - \gamma \mathrm{P}_\theta)^{-1} Z(\cdot) .
$$

Next, using the definition of the discounted state visitation (48) and the regularized Q-function (47) implies

$$
\frac{\partial v_\theta^f(\rho)}{\partial \theta(s,b)} = \frac{1}{1-\gamma} \sum_{s' \in \mathcal{S}} d_\rho^\theta(s') \sum_{a \in \mathcal{A}} \frac{\partial \pi_\theta^f(a|s')}{\partial \theta(s,b)} \left[ q_\theta^f(s',a) - \lambda f'\left( \frac{\pi_\theta^f(a|s')}{\pi_{\mathrm{ref}}(a|s)} \right) \right] . \tag{51}
$$

Using that $\sum_{a \in \mathcal{A}} \frac{\partial \pi_\theta^f(a|s)}{\partial \theta(s,b)} = 0$ and that for $s \neq s'$, we have $\frac{\partial \pi_\theta^f(a|s')}{\partial \theta(s,b)} = 0$ yields

$$
\frac{\partial v_\theta^f(\rho)}{\partial \theta(s,b)} = \frac{1}{1-\gamma} d_\rho^\theta(s) \sum_{a \in \mathcal{A}} \frac{\partial \pi_\theta^f(a|s)}{\partial \theta(s,b)} \left[ q_\theta^f(s,a) - \lambda f'\left( \frac{\pi_\theta^f(a|s)}{\pi_{\mathrm{ref}}(a|s)} \right) - \lambda\mu_\theta(s) \right] .
$$

where $\mu_\theta(s)$ is defined in Corollary B.7 and satisfies for any $a \in \mathcal{A}$

$$
\theta(s,a) - f'\left( \frac{\pi_\theta^f(a|s)}{\pi_{\mathrm{ref}}(a|s)} \right) = \mu_\theta(s) .
$$

Thus, we obtain

$$
\frac{\partial v_\theta^f(\rho)}{\partial \theta(s,b)} = \frac{1}{1-\gamma} d_\rho^\theta(s) \sum_{a \in \mathcal{A}} \frac{\partial \pi_\theta^f(a|s)}{\partial \theta(s,b)} \left[ q_\theta^f(s,a) - \lambda\theta(s,a) \right] ,
$$

Finally, plugging in the expression of the derivative of the policy derived in Corollary B.7 in the previous equality concludes the proof. $\qquad\square$

Using the previous lemma, we prove the following lower bound for the norm of the gradient.

**Lemma C.3.** *Assume that, for some $\pi_{\text{ref}} > 0$, $f$ and $\pi_{\text{ref}}$ satisfy $\mathbf{A}_f(\pi_{\text{ref}})$ and $\mathbf{P}(\pi_{\text{ref}})$ respectively. Assume in addition that the initial distribution $\rho$ satisfy $\mathbf{A}_\rho$. We have*

$$\left\|\frac{\partial v_\theta^f(\rho)}{\partial \theta}\right\|_2^2 \geq \lambda^2 \rho_{\min}^2 \min_{(s,a) \in \mathcal{S} \times \mathcal{A}} \{\mathrm{w}_\theta^f(a|s)^2\} \min_{s \in \mathcal{S}} \{\mathrm{W}_\theta^f(s)^2\} \sum_{s \in \mathcal{S}} \|\zeta_\theta(s)\|_2^2 \ .$$

*Proof.* It holds that

$$\left\|\frac{\partial v_\theta^f(\rho)}{\partial \theta}\right\|_2^2 = \sum_{s \in \mathcal{S}} \left\|\frac{\partial v_\theta^f(\rho)}{\partial \theta(s, \cdot)}\right\|_2^2 \ .$$

Fix $s \in \mathcal{S}$. Using Lemma C.2, we observe that

$$\frac{1}{\mathrm{W}_\theta^f(s)} \frac{\partial v_\theta^f(\rho)}{\partial \theta(s, \cdot)} = \frac{d_\rho^\theta(s)}{1 - \gamma} H(\mathrm{w}_\theta^f(\cdot|s)) \left[q_\theta^f(s, \cdot) - \lambda \theta(s, \cdot)\right] \ .$$

where for any vector $u \in \mathbb{R}^{|\mathcal{A}|}$, we define $H(u) := \mathrm{diag}(u) - uu^\top$. Thus, we get that

$$\frac{1 - \gamma}{\mathrm{W}_\theta^f(s)} \left\|\frac{\partial v_\theta^f(\rho)}{\partial \theta(s, \cdot)}\right\|_2 = d_\rho^\theta(s) \left\|H(\mathrm{w}_\theta^f(\cdot|s)) \left[q_\theta^f(s, \cdot) - \lambda \theta(s, \cdot)\right]\right\|_2$$

$$= \lambda d_\rho^\theta(s) \left\|H(\mathrm{w}_\theta^f(\cdot|s)) \left[q_\theta^f(s, \cdot)/\lambda - \theta(s, \cdot) - K_\theta^f(s)\mathbf{1}_{|\mathcal{A}|}\right]\right\|_2 \quad \text{(using } H(u)\mathbf{1}_{|\mathcal{A}|} = 0 \text{ and (50))}$$

$$\geq \lambda d_\rho^\theta(s) \min_{a \in \mathcal{A}} \mathrm{w}_\theta^f(a|s) \|\zeta_\theta(s)\|_2 \quad \text{(where } \zeta_\theta(s) \text{ is defined in (49) and using Lemma I.4)} \ .$$

Finally, using $d_\rho^\theta(s) \geq (1 - \gamma)\rho(s)$ and $\mathbf{A}_\rho$ concludes the proof. $\qquad \square$

## C.2 BOUNDING THE SUBOPTIMALITY GAP

The first step is to connect the suboptimality gap to information localized at $\theta$. This is achieved via the performance difference lemma for the regularized value function yields (see Lemma I.3)

$$v_\star^f(\rho) - v_\theta^f(\rho) = \sum_{s \in \mathcal{S}} \frac{\lambda d_\rho^{\pi_\star^f}(s)}{1 - \gamma} \underbrace{\left[\sum_{a \in \mathcal{A}} \pi_\star^f(a|s)\frac{q_\theta^f(s, a)}{\lambda} - \mathrm{D}^f(\pi_\star^f(\cdot|s)\|\pi_{\text{ref}}(\cdot|s)) - \frac{v_\theta^f(s)}{\lambda}\right]}_{(\mathbf{A}(\mathbf{s}))} . \quad (52)$$

Fix $s \in \mathcal{S}$. Using the definition of the regularized value functions and Q-functions combined with Equation (42), we have

$$v_\theta^f(s) = \langle \pi_\theta^f(\cdot|s), q_\theta^f(s, \cdot)\rangle - \lambda \mathrm{D}^f(\pi_\theta^f(\cdot|s)\|\pi_{\text{ref}}(\cdot|s)) \ .$$

This implies $\mathbf{A}(s) = \mathbf{A}_1(s) - \mathbf{A}_2(s) - \mathbf{A}_3(s)$ where

$$\begin{aligned} \mathbf{A}_1(s) &= \langle \pi_\star^f(\cdot|s), q_\theta^f(s, \cdot)/\lambda\rangle - \mathrm{D}^f(\pi_\star^f(\cdot|s)\|\pi_{\text{ref}}(\cdot|s)) \\ \mathbf{A}_2(s) &= \langle \pi_\theta^f(\cdot|s), \theta(s, \cdot)\rangle - \mathrm{D}^f(\pi_\theta^f(\cdot|s)\|\pi_{\text{ref}}(\cdot|s)) \\ \mathbf{A}_3(s) &= \langle \pi_\theta^f(\cdot|s), q_\theta^f(s, \cdot)/\lambda - \theta(s, \cdot)\rangle \ . \end{aligned} \quad (53)$$

Using (4) and (15), combined with $\mathbf{A}_1(s) \leq \text{f-softmax}(q_\theta^f(s, \cdot)/\lambda, \pi_{\text{ref}}(\cdot|s))$ and the fact that $\mathbf{A}_2(s) = \text{f-softmax}(\theta(s, \cdot), \pi_{\text{ref}}(\cdot|s))$, gives

$$\begin{aligned} \mathbf{A}(s) &\leq \text{f-softmax}(q_\theta^f(s, \cdot)/\lambda, \pi_{\text{ref}}(\cdot|s)) - \text{f-softmax}(\theta(s, \cdot), \pi_{\text{ref}}(\cdot|s)) - \mathbf{A}_3(s) \\ &= \text{f-softmax}(q_\theta^f(s, \cdot)/\lambda, \pi_{\text{ref}}(\cdot|s)) - \text{f-softmax}(\theta(s, \cdot) + K_\theta^f(s)\mathbf{1}_{|\mathcal{A}|}) \\ &\quad - \langle \pi_\theta^f(\cdot|s), q_\theta^f(s, \cdot)/\lambda - \theta(s, \cdot) - K_\theta^f(s)\mathbf{1}_{|\mathcal{A}|}\rangle \ , \end{aligned} \quad (54)$$

where in the last equality we used that, for any $x \in \mathbb{R}^{|\mathcal{A}|}$ and $\alpha \in \mathbb{R}$,

$$\text{f-softmax}(x + \alpha\mathbf{1}_{|\mathcal{A}|}, \pi_{\text{ref}}(\cdot|s)) = \text{f-softmax}(x, \pi_{\text{ref}}(\cdot|s)) + \alpha \ .$$

The structure of $(\mathbf{A}(\mathbf{s}))$ closely resembles that of a first-order Taylor expansion as by Lemma B.5, we have that

$$\frac{\partial\, \text{f-softmax}(\theta(s,\cdot), \pi_{\text{ref}}(\cdot|s))}{\partial\theta(s,\cdot)} = \text{f-softargmax}(\theta(s,\cdot), \pi_{\text{ref}}(\cdot|s)) = \pi_\theta^f(\cdot|s) \ . \tag{55}$$

**Lemma C.4.** *Assume* $\mathbf{A}_f(\pi_{\text{ref}})$*. It holds that*

$$v_\star^f(\rho) - v_\theta^f(\rho) \le \frac{\lambda}{1-\gamma} \sup_{\theta \in \mathbb{R}^{|\mathcal{S}||\mathcal{A}|}} \{\|W_\theta\|_\infty\} \sum_{s \in \mathcal{S}} \left\| \zeta_\theta^f(s) \right\|_2^2$$

*Proof.* In this proof, we denote by $g_s(x) = \text{f-softmax}(x, \pi_{\text{ref}}(\cdot|s))$. Combining (52) and (54) yields

$$v_\star^f(\rho) - v_\theta^f(\rho) \le \frac{\lambda}{1-\gamma} \sum_{s \in \mathcal{S}} d_\rho^{\pi_\star^f}(s) B(s) \ , \tag{56}$$

where we have defined

$$B(s) = g_s\big(q_\theta^f(s,\cdot)/\lambda\big) - g_s\big(\theta(s,\cdot) + K_\theta^f(s)\mathbf{1}_{|\mathcal{A}|}\big) - \langle \pi_\theta^f(\cdot|s), q_\theta^f(s,\cdot)/\lambda - \theta(s,\cdot) - K_\theta^f(s)\mathbf{1}_{|\mathcal{A}|}\rangle \ .$$

Next, by Lemma B.5, it holds that

$$\left.\frac{\partial g_s(x)}{\partial x}\right|_{x=\theta(s,\cdot)+K_\theta(s)\mathbf{1}_{|\mathcal{A}|}} = \pi_\theta^f(\cdot|s) \ .$$

Standard one-dimensional Taylor theorem with Lagrange remainder shows that there exists $y \in \mathbb{R}^{|\mathcal{A}|}$ which belongs to the segment between $\theta(s,\cdot) + K_\theta(s)\mathbf{1}_{\mathcal{A}}$ and $q_\theta^f(s,\cdot)/\lambda$ such that

$$B(s) = \frac{1}{2}\langle \left.\frac{\partial^2 g_s(x)}{\partial x^2}\right|_{x=y} \zeta_\theta^f(s), \zeta_\theta^f(s)\rangle$$

Using the bound on the spectral norm of the Hessian of $g_s$ derived in Lemma B.5, we obtain

$$B(s) = \frac{1}{2}\langle \left.\frac{\partial^2 g_s(x)}{\partial x^2}\right|_{x=y}, \zeta_\theta^f, \zeta_\theta^f\rangle \le \frac{1}{2}\left\| \left.\frac{\partial^2 g_s(x)}{\partial x^2}\right|_{x=y}\right\|_2 \left\|\zeta_\theta^f(s)\right\|_2^2 \le \mathrm{W}_y^f(s) \left\|\zeta_\theta^f(s)\right\|_2^2 \ .$$

Finally, bounding the discounted state visitation measure in (56) by 1 and plugging in the preceding bound on $B(s)$ concludes the proof. □

The proof of Theorem C.1 follows immediately from Lemma C.3, Lemma C.4, Lemma B.6, and (7).

## D    MONOTONE IMPROVEMENT OPERATORS

A key challenge in analyzing stochastic policy gradient methods is that the Łojasiewicz inequality depends on $\theta$ and degenerates whenever the probability of an action becomes small. The goal of this section is therefore to show the existence of an operator $\text{IMP}^f$ with two crucial properties: (i) for any policy, applying this operator produces a new policy with higher objective value, and (ii) every policy generated by this operator assigns at least a fixed minimum probability to every action. The main idea is to build the improvement operator such that it slightly augments the smallest probability weights, such that for any state action pair $(s,a) \in \mathcal{S} \times \mathcal{A}$ the probability ratio $\pi(a|s)/\pi_{\text{ref}}(a|s)$ stays above a certain threshold. We will show below that this procedure improves the global objective while keeping the probabilities uniformly bounded away from 0 when the threshold is properly chosen. Let $\underline{\pi_{\text{ref}}} > 0$ be such that $\mathbf{A}_f(\pi_{\text{ref}})$ and $\mathbf{P}(\pi_{\text{ref}})$ hold. For any policy $\pi$, state $s \in \mathcal{S}$, $\tau < \underline{\pi_{\text{ref}}}/2$, we respectively define $\mathcal{A}_\tau^\pi(s)$, and $a_{\max}^\pi(s)$ as

$$\mathcal{A}_\tau^\pi(s) := \{a \in \mathcal{A}, \pi(a|s)/\pi_{\text{ref}}(a|s) \le \tau/2\} \ , \quad a_{\max}^\pi(s) = \arg\max_{a \in \mathcal{A}}\{\pi(a|s)/\pi_{\text{ref}}(a|s)\} \ ,$$

where the $\arg\max$ is chosen at random in the case of ties. Note that the definition of $\tau$ ensures that $a_{\max}^\pi(s)$ does not belong to the set $\mathcal{A}_\tau^\pi(s)$ as

$$\max_{a \in \mathcal{A}} \frac{\pi(a|s)}{\pi_{\mathrm{ref}}(a|s)} \geq 1 \ .$$

Finally, we define the improvement operator as follows:

$$\begin{aligned} \mathcal{U}_\tau : \mathcal{P}(\mathcal{A})^\mathcal{S} &\longrightarrow \mathcal{P}(\mathcal{A})^\mathcal{S}, \\ \pi &\longmapsto \mathcal{U}_\tau(\pi), \end{aligned} \tag{57}$$

where for every $(s, a) \in \mathcal{S} \times \mathcal{A}$,

$$\mathcal{U}_\tau(\pi)(a|s) = \begin{cases} \pi_{\mathrm{ref}}(a|s)\tau, & \text{if } \pi(a|s) \leq \pi_{\mathrm{ref}}(a|s)\tau/2, \\ \pi(a|s) - \displaystyle\sum_{b \in \mathcal{A}_\tau^\pi(s)} \big(\pi_{\mathrm{ref}}(b|s)\tau - \pi(b|s)\big), & \text{if } a = a_{\max}^\pi(s), \\ \pi(a|s), & \text{otherwise.} \end{cases}$$

The operator $\mathcal{U}_\tau$ builds $\mathcal{U}_\tau(\pi)(a|s)$ by (statewise) raising each $a \in \mathcal{A}_\tau^\pi(s)$ to $\pi_{\mathrm{ref}}(a|s)\tau$, substracting the total added mass from the single action $a_{\max}^\pi(s)$, and leaving other actions unchanged. If $\mathcal{A}_\tau^\pi(s) = \emptyset$, for all $s \in \mathcal{S}$, then $\mathrm{IMP}^f(\pi) = \pi$. Note that mass conservation is immediate from the definition and the fact that $\tau < \underline{\pi_{\mathrm{ref}}}/2$. Non-negativity of $\mathcal{U}_\tau(\pi)(a_{\max}^\pi(s)|s)$ follows because the removed mass is

$$\sum_{a \in \mathcal{A}_\tau^\pi(s)} \{\pi_{\mathrm{ref}}(a|s)\tau - \pi(a|s)\} \leq \tau \sum_{a \in \mathcal{A}_\tau^\pi(s)} \pi_{\mathrm{ref}}(a|s) \leq \tau$$

Since $\pi(a_{\max}^\pi(s)|s) \geq \pi_{\mathrm{ref}}(a_{\max}^\pi(s)|s) \geq \underline{\pi_{\mathrm{ref}}}$, and $\tau \leq \underline{\pi_{\mathrm{ref}}}/2$, we get that $\mathcal{U}_\tau(\pi)(a_{\max}^\pi(s)|s) \geq \tau/2$. This in particular shows that $\mathcal{U}_\tau(\pi)$ is a policy. As by $\mathbf{A}_f(\underline{\pi_{\mathrm{ref}}})$ we have $\lim_{x \to 0^+} f'(x) = -\infty$, we consider $[f']^{-1} : (-\infty, f'(1/\underline{\pi_{\mathrm{ref}}})] \mapsto \mathbb{R}_+$ the inverse of $f'$ and define

$$\tau_\lambda := \min\left( [f']^{-1}\left(-\frac{16 + 8\gamma\lambda\mathrm{d}_f}{\lambda(1-\gamma)^2\rho_{\min}}\right), [f']^{-1}\left(-4\left|f'\left(\frac{1}{2}\right)\right|\right), \frac{1}{2}\underline{\pi_{\mathrm{ref}}} \right) \ . \tag{58}$$

The following lemma establishes the crucial improvement property when $\tau = \tau_\lambda$.

**Lemma D.1.** *Assume that, for some $\underline{\pi_{\mathrm{ref}}} > 0$, $f$ and $\pi_{\mathrm{ref}}$ satisfy $\mathbf{A}_f(\underline{\pi_{\mathrm{ref}}})$ and $\mathbf{P}(\underline{\pi_{\mathrm{ref}}})$ respectively. Assume in addition that the initial distribution $\rho$ satisfies $\mathbf{A}_\rho$. For any policy $\pi$, it holds that*

$$v_{\mathcal{U}_{\tau_\lambda}(\pi)}^f(\rho) \geq v_\pi^f(\rho) \ .$$

*Additionally, for any policy $\pi$, we have that*

$$\mathcal{U}_{\tau_\lambda}(\pi)(a|s) \geq \underline{\pi_{\mathrm{ref}}}\tau_\lambda \ .$$

*Proof.* Set an arbitrary policy $\pi$. For avoiding heavy notations, we will, through this proof, denote by $A_\tau^\pi = A_{\tau_\lambda}^\pi$. We consider the case where there is $s \in \mathcal{S}$ such that $\mathcal{A}_\tau^\pi(s) \neq \emptyset$ (alternatively $\mathcal{U}_{\tau_\lambda}(\pi) = \pi$, which makes the previous inequality immediately valid). Define $\tilde{\pi} = \mathcal{U}_{\tau_\lambda}(\pi)$. The following applies

$$\begin{aligned} v_{\tilde{\pi}}^f(\rho) - v_\pi^f(\rho) &= \sum_{s \in \mathcal{S}} d_\rho^{\tilde{\pi}}(s) \sum_{a \in \mathcal{A}} \left[ \tilde{\pi}(a|s)\mathsf{r}(s,a) - \lambda\pi_{\mathrm{ref}}(a|s)f\left(\frac{\tilde{\pi}(a|s)}{\pi_{\mathrm{ref}}(a|s)}\right) \right] \\ &\quad - \sum_{s \in \mathcal{S}} d_\rho^\pi(s) \sum_{a \in \mathcal{A}} \left[ \pi(a|s)\mathsf{r}(s,a) - \lambda\pi_{\mathrm{ref}}(a|s)f\left(\frac{\pi(a|s)}{\pi_{\mathrm{ref}}(a|s)}\right) \right] \\ &= \underbrace{\sum_{s \in \mathcal{S}} \big(d_\rho^{\tilde{\pi}}(s) - d_\rho^\pi(s)\big) \sum_{a \in \mathcal{A}} \left[ \tilde{\pi}(a|s)\mathsf{r}(s,a) - \lambda\pi_{\mathrm{ref}}(a|s)f\left(\frac{\tilde{\pi}(a|s)}{\pi_{\mathrm{ref}}(a|s)}\right) \right]}_{\textbf{(I)}} \\ &\quad + \underbrace{\sum_{s \in \mathcal{S}} d_\rho^\pi(s) \sum_{a \in \mathcal{A}} (\tilde{\pi}(a|s) - \pi(a|s))\mathsf{r}(s,a)}_{\textbf{(II)}} \end{aligned}$$

$$+ \lambda \sum_{s \in \mathcal{S}} d_\rho^\pi(s) \sum_{a \in \mathcal{A}} \pi_{\text{ref}}(a|s) \underbrace{\left[ f\left( \frac{\pi(a|s)}{\pi_{\text{ref}}(a|s)} \right) - f\left( \frac{\tilde{\pi}(a|s)}{\pi_{\text{ref}}(a|s)} \right) \right]}_{\textbf{(III)}} \ .$$

We now lower-bound each of the three terms separately.

**Bounding (I).** Using Lemma I.2, we have

$$\textbf{(I)} \geq - \left\| d_\rho^{\tilde{\pi}} - d_\rho^\pi \right\|_1 \max_{s \in \mathcal{S}} \left| \sum_{a \in \mathcal{A}} \left[ \tilde{\pi}(a|s) \mathsf{r}(s,a) - \lambda \pi_{\text{ref}}(a|s) f\left( \frac{\tilde{\pi}(a|s)}{\pi_{\text{ref}}(a|s)} \right) \right] \right|$$

$$\geq -\frac{\gamma}{1-\gamma} \sup_{s \in \mathcal{S}} \|\tilde{\pi}(\cdot|s) - \pi(\cdot|s)\|_1 \sup_{s \in \mathcal{S}} \left[ 1 + \lambda \, \mathrm{D}^f(\tilde{\pi}(\cdot|s) \| \pi_{\text{ref}}(\cdot|s)) \right]$$

$$\geq -\frac{2\gamma}{1-\gamma} \tau_\lambda \max_{s \in \mathcal{S}} \left\{ \sum_{a \in A_\tau^\pi(s)} \pi_{\text{ref}}(a|s) \right\} \sup_{\nu \in \mathcal{P}(\mathcal{A})} \sup_{s \in \mathcal{S}} \left[ 1 + \lambda \, \mathrm{D}^f(\nu \| \pi_{\text{ref}}(\cdot|s)) \right] \ ,$$

where in the last inequality we used that (because we increase the probability of the actions in $A_\tau^\pi(s)$ by $\tau_\lambda$ and remove the total added mass from the probability of $\pi(a_{\max}^\pi(s))$)

$$\sup_{s \in \mathcal{S}} \|\tilde{\pi}(\cdot|s) - \pi(\cdot|s)\|_1 \leq 2 \max_{s \in \mathcal{S}} \left\{ \sum_{a \in A_\tau^\pi(s)} \pi_{\text{ref}}(a|s) \right\} \tau_\lambda \ .$$

**Bounding (II).** Using the triangle inequality yields

$$\textbf{(II)} \geq -\sup_{s \in \mathcal{S}} \|\tilde{\pi}(\cdot|s) - \pi(\cdot|s)\|_1 \geq -2 \max_{s \in \mathcal{S}} \left\{ \sum_{a \in A_\tau^\pi(s)} \pi_{\text{ref}}(a|s) \right\} \tau_\lambda \ .$$

**Bounding (III).** All the state-action pairs on which the original $\pi$ allocates the same probability then the policy $\tilde{\pi}$ are equal to $0$ in **(III)** allowing us to simplify this term

$$\textbf{(III)} = \lambda \sum_{s \in \mathcal{S}} d_\rho^\pi(s) \sum_{a \in \mathcal{A}} \pi_{\text{ref}}(a|s) \left[ f\left( \frac{\pi(a|s)}{\pi_{\text{ref}}(a|s)} \right) - f\left( \frac{\tilde{\pi}(a|s)}{\pi_{\text{ref}}(a|s)} \right) \right]$$

$$= \lambda \sum_{s \in \mathcal{S}} d_\rho^\pi(s) \sum_{a \in \mathcal{A}_\tau^\pi(s)} \pi_{\text{ref}}(a|s) \left[ f\left( \frac{\pi(a|s)}{\pi_{\text{ref}}(a|s)} \right) - f\left( \frac{\tilde{\pi}(a|s)}{\pi_{\text{ref}}(a|s)} \right) \right]$$

$$+ \lambda \sum_{s \in \mathcal{S}} \mathbb{1}(\mathcal{A}_\tau^\pi(s) \neq \emptyset) d_\rho^\pi(s) \pi_{\text{ref}}(a_{\max}^\pi(s)|s) \left[ f\left( \frac{\pi(a_{\max}^\pi(s)|s)}{\pi_{\text{ref}}(a_{\max}^\pi(s)|s)} \right) - f\left( \frac{\tilde{\pi}(a_{\max}^\pi(s)|s)}{\pi_{\text{ref}}(a_{\max}^\pi(s)|s)} \right) \right] \ .$$

Since $f$ is convex, for all $u, v \in [0; 1/\underline{\pi_{\text{ref}}}]$, $f(u) - f(v) \geq f'(v)(u-v)$, we have

$$\textbf{(III)} \geq \lambda \sum_{s \in \mathcal{S}} d_\rho^\pi(s) \sum_{a \in \mathcal{A}_\tau^\pi(s)} (\pi(a|s) - \tilde{\pi}(a|s)) f'(\tau_\lambda) \qquad \text{(since } \tilde{\pi}(a|s)/\pi_{\text{ref}}(a|s) = \tau_\lambda \text{)}$$

$$+ \lambda \sum_{s \in \mathcal{S}} \mathbb{1}(\mathcal{A}_\tau^\pi(s) \neq \emptyset) d_\rho^\pi(s) \left[ \pi(a_{\max}^\pi(s)|s) - \tilde{\pi}(a_{\max}^\pi(s)|s) \right] f'\left( \frac{\tilde{\pi}(a_{\max}^\pi(s)|s)}{\pi_{\text{ref}}(a_{\max}^\pi(s)|s)} \right) \ ,$$

Next, using that

$$\frac{\tilde{\pi}(a_{\max}^\pi(s)|s)}{\pi_{\text{ref}}(a_{\max}^\pi(s)|s)} \geq \frac{\pi(a_{\max}^\pi(s)|s) - \tau_\lambda}{\pi_{\text{ref}}(a_{\max}^\pi(s)|s)} \geq \frac{\pi(a_{\max}^\pi(s)|s) - \underline{\pi_{\text{ref}}}/2}{\pi_{\text{ref}}(a_{\max}^\pi(s)|s)} \geq 1 - \frac{1}{2} = \frac{1}{2} \ ,$$

combined with the monotonicity of $f'$ and the fact that $\pi(a_{\max}^\pi(s)|s) - \tilde{\pi}(a_{\max}^\pi(s)|s) \geq 0$ yields

$$\textbf{(III)} \geq \lambda \sum_{s \in \mathcal{S}} d_\rho^\pi(s) \sum_{a \in \mathcal{A}_\tau^\pi(s)} (\pi(a|s) - \tilde{\pi}(a|s)) f'(\tau_\lambda) \qquad \text{(since } \tilde{\pi}(a|s)/\pi_{\text{ref}}(a|s) = \tau_\lambda \text{)}$$

$$+ \lambda \sum_{s \in \mathcal{S}} 1(\mathcal{A}_\tau^\pi(s) \neq \emptyset) d_\rho^\pi(s) \left[ \pi(a_{\max}^\pi(s)|s) - \tilde{\pi}(a_{\max}^\pi(s)|s) \right] f' \left( \frac{\tilde{\pi}(a_{\max}^\pi(s)|s)}{\pi_{\mathrm{ref}}(a_{\max}^\pi(s)|s)} \right) \ ,$$

Additionally, since

$$0 \leq \pi(a_{\max}^\pi(s)|s) - \tilde{\pi}(a_{\max}^\pi(s)|s) \leq \sum_{a \in \mathcal{A}_\tau^\pi(s)} (\pi(a|s) - \tilde{\pi}(a|s)) \leq \tau_\lambda \sum_{a \in \mathcal{A}_\tau^\pi(s)} \pi_{\mathrm{ref}}(a|s) \ ,$$

implies

$$(\mathbf{III}) \geq -\frac{\lambda}{2} \sum_{s \in \mathcal{S}} d_\rho^\pi(s) 1(\mathcal{A}_\tau^\pi(s) \neq \emptyset) \left( \sum_{a \in \mathcal{A}_\tau^\pi(s)} \pi_{\mathrm{ref}}(a|s) \right) \tau_\lambda f'(\tau_\lambda)$$

$$+ \lambda \sum_{s \in \mathcal{S}} d_\rho^\pi(s) 1(\mathcal{A}_\tau^\pi(s) \neq \emptyset) \left( \sum_{a \in \mathcal{A}_\tau^\pi(s)} \pi_{\mathrm{ref}}(a|s) \right) \tau_\lambda f'(1/2) \ ,$$

$$\geq -\frac{\lambda}{4} \sum_{s \in \mathcal{S}} d_\rho^\pi(s) 1(\mathcal{A}_\tau^\pi(s) \neq \emptyset) \left( \sum_{a \in \mathcal{A}_\tau^\pi(s)} \pi_{\mathrm{ref}}(a|s) \right) \tau_\lambda f'(\tau_\lambda) \ ,$$

where in the last inequality, we used that $f'(\tau_\lambda) \leq -4|f'(1/2)|$. Hence, by using $\mathbf{A}_\rho$, we can lower bound this term as follows

$$(\mathbf{III}) \geq -\frac{\lambda}{4} (1 - \gamma) \min_{s \in \mathcal{S}} \{\rho(s)\} \max_{s \in \mathcal{S}} \left\{ \sum_{a \in A_\tau^\pi(s)} \pi_{\mathrm{ref}}(a|s) \right\} \tau_\lambda f'(\tau_\lambda) \ .$$

Collecting these lower bounds and using that

$$f'(\tau_\lambda) \leq -\frac{16 + 8\gamma\lambda \mathrm{d}_f}{\lambda(1 - \gamma)^2 \rho_{\min}}$$

concludes the proof. $\qquad \square$

Finally, we define the operator that maps each policy to one corresponding parameter

$$M^f \colon \Pi \ \to \ \mathbb{R}^{|\mathcal{S}||\mathcal{A}|}$$

by

$$M^f(\pi)(s,a) \ := \ f' \left( \frac{\pi(a \mid s)}{\pi_{\mathrm{ref}}(a \mid s)} \right) - f' \left( \frac{\pi(a_{|\mathcal{A}|}|s)}{\pi_{\mathrm{ref}}(a_{|\mathcal{A}|}|s)} \right), \quad \forall (s,a) \in \mathcal{S} \times \mathcal{A} \,. \qquad (59)$$

Finally, we define the improvement operator on the logitspace as

$$\mathcal{T}_\tau := M^f \circ \mathcal{U}_\tau \ .$$

The following lemma shows that $M^f$ successfully recovers a parameter that gives the policy and that $\mathcal{T}_\tau$ improves the value of the objective when $\lambda = \tau_\lambda$.

**Lemma D.2.** *Assume that, for some $\underline{\pi_{\mathrm{ref}}} > 0$, $f$ and $\pi_{\mathrm{ref}}$ satisfy $\mathbf{A}_f(\underline{\pi_{\mathrm{ref}}})$ and $\mathbf{P}(\underline{\pi_{\mathrm{ref}}})$ respectively. Assume in addition that the initial distribution $\rho$ satisfies $\mathbf{A}_\rho$. For any policy $\pi$, it holds that*

$$\pi_{M^f(\pi)}^f = \pi \ ,$$

*Additionally, for any $\theta \in \mathbb{R}^{|\mathcal{S}||\mathcal{A}|}$ and $(s,a) \in \mathcal{S} \times \mathcal{A}$, we have that*

$$v_{\mathcal{T}_{\tau_\lambda}(\theta)}^f \geq v_\theta^f \ , \quad \pi_{\mathcal{T}_{\tau_\lambda}(\theta)}^f \geq \underline{\pi_{\mathrm{ref}}} \tau_\lambda \ .$$

*Proof.* The proof follows immediately from a combination of equality (40) in Corollary B.7, (59), and Lemma D.1. $\qquad \square$

# E CONVERGENCE ANALYSIS OF STOCHASTIC POLICY GRADIENT

In this section, we aim to derive under $\mathbf{A}_f(\pi_{\mathrm{ref}})$ and $\mathbf{A}_\rho$ non-asymptotic convergence rates for $f-\mathrm{PG}$. First, we establish a bound on the bias and variance of the REINFORCE estimator defined in (11).

## E.1 BOUNDING THE BIAS AND VARIANCE OF THE STOCHASTIC ESTIMATOR

First, recall the expression of the stochastic estimator of the gradient

$$
\begin{aligned}
\mathrm{g}_z^f(\theta) = &\frac{1}{B} \sum_{b=0}^{B-1} \sum_{h=0}^{H-1} \sum_{\ell=0}^{h} \frac{\partial \log \pi_\theta^f(a_\ell|s_\ell)}{\partial\theta} \gamma^h \mathrm{r}(s_h, a_h) \\
&- \frac{\lambda}{B} \sum_{b=0}^{B-1} \sum_{h=0}^{H-1} \sum_{\ell=0}^{h-1} \frac{\partial \log \pi_\theta^f(a_\ell|s_\ell)}{\partial\theta} \gamma^h \mathrm{D}^f(\pi_\theta^f(\cdot|s_h)\|\pi_{\mathrm{ref}}(\cdot|s_h)) - \lambda\frac{1}{B} \sum_{b=0}^{B-1} \sum_{h=0}^{H-1} \gamma^h \mathrm{F}_\theta^f(s_h),
\end{aligned}
\tag{60}
$$

where $z = (s_{0:H-1}^b, a_{0:H-1}^b)_{b=0}^{B-1} \in (\mathcal{S} \cdot \mathcal{A})^{H \cdot B}$, and we recall that for any $s \in \mathcal{S}$, $\mathrm{F}_\theta^f(s)$ is a vector of size $|\mathcal{S}| \times |\mathcal{A}|$ defined as

$$
[\mathrm{F}_\theta^f(s)]_{(s',b)} = 1_s(s') \, \mathrm{W}_\theta^f(s) \, \mathrm{w}_\theta^f(b|s) \left[ f'\left(\frac{\pi_\theta^f(b|s)}{\pi_{\mathrm{ref}}(b|s)}\right) - \sum_{a\in\mathcal{A}} \mathrm{w}_\theta^f(a|s) f'\left(\frac{\pi_\theta^f(a|s)}{\pi_{\mathrm{ref}}(a|s)}\right) \right] \, ,
$$

and where $\mathrm{W}_\theta^f$, $\mathrm{Y}_\theta^f$, and $\mathrm{w}_\theta^f$ are defined in (38) and (39). Finally, define the expected gradient estimator as

$$
\mathrm{g}^f(\theta) := \mathbb{E}_{Z\sim[\nu(\theta)]^{\otimes B}} \left[ \mathrm{g}_Z^f(\theta) \right] \, ,
\tag{61}
$$

Before bounding the bias and the variance, we give an explicit expression of the derivative of the log probability that appears in the expression of our stochastic gradient estimator. We also provide a bound on the derivative of the log probabilities and on the matrix $\mathrm{F}_\theta^f(s)$ for any state $s \in \mathcal{S}$.

**Lemma E.1.** *Assume that, for some $\underline{\pi}_{\mathrm{ref}} > 0$, $f$ satisfy $\mathbf{A}_f(\underline{\pi}_{\mathrm{ref}})$. For any $\theta \in \mathbb{R}^{|\mathcal{S}||\mathcal{A}|}$, $(s, s', a, b) \in \mathcal{S}^2 \times \mathcal{A}^2$, we have*

$$
\frac{\partial \log \pi_\theta^f(a|s)}{\partial\theta(s',b)} = 1_{s'}(s) \frac{\mathrm{W}_\theta^f(s)}{\pi_\theta^f(a|s)} \left[ 1_b(a) \, \mathrm{w}_\theta^f(a|s) - \mathrm{w}_\theta^f(a|s) \, \mathrm{w}_\theta^f(b|s) \right] \, .
$$

*Additionally, we have that*

$$
\left\| \frac{\partial \log \pi_\theta^f(a|s)}{\partial\theta} \right\|_2 \le \frac{2 \, \mathrm{W}_\theta^f(s) \, \mathrm{w}_\theta^f(a|s)}{\pi_\theta^f(a|s)} \, , \quad \left\| \mathrm{F}_\theta^f(s) \right\|_2 \le 2 \, \mathrm{W}_\theta^f(s) \, \mathrm{Y}_\theta^f(s) \, .
$$

*Proof.* The proof follows from the log-derivative trick and the expression of the derivative of the policy provided in Corollary B.7. $\square$

Next, we establish a REINFORCE-type formula for the gradient of the objective.

**Lemma E.2.** *Assume that, for some $\underline{\pi}_{\mathrm{ref}} > 0$, $f$ satisfy $\mathbf{A}_f(\underline{\pi}_{\mathrm{ref}})$. It holds that*

$$
\begin{aligned}
\frac{\partial v_\theta^f(\rho)}{\partial\theta(s,b)} = &\mathbb{E}\left[ \sum_{t=0}^{\infty} \sum_{\ell=0}^{t} \frac{\partial \log \pi_\theta^f(A_\ell|S_\ell)}{\partial\theta(s,b)} \gamma^t \mathrm{r}(S_t, A_t) \right] \\
&- \lambda\mathbb{E}\left[ \sum_{t=0}^{\infty} \sum_{\ell=0}^{t-1} \frac{\partial \log \pi_\theta^f(A_\ell|S_\ell)}{\partial\theta(s,b)} \gamma^t \mathrm{D}^f(\pi_\theta^f(\cdot|S_t)\|\pi_{\mathrm{ref}}(\cdot|S_t)) \right] \\
&- \lambda\mathbb{E}\left[ \sum_{t=0}^{\infty} \gamma^t 1_s(S_t) \, \mathrm{W}_\theta^f(s) \, \mathrm{w}_\theta^f(b|s) \left[ f'\left(\frac{\pi_\theta^f(b|s)}{\pi_{\mathrm{ref}}(b|s)}\right) - \sum_{a\in\mathcal{A}} \mathrm{w}_\theta^f(a|s) f'\left(\frac{\pi_\theta^f(a|s)}{\pi_{\mathrm{ref}}(a|s)}\right) \right] \right] \, .
\end{aligned}
$$

*Proof.* Fix a parameter $\theta \in \mathbb{R}^{|\mathcal{S}||\mathcal{A}|}$, a horizon $T$, and a divergence generator $f$. For any truncated trajectory $z = (s_t, a_t)_{t=0}^{T-1} \in (\mathcal{S} \times \mathcal{A})^T$, we define its probability as

$$\nu_T^f(\theta; z) = \rho(s_0)\pi_\theta^f(a_0|s_0)\prod_{t=1}^{T-1}\mathsf{P}(s_t|s_{t-1},a_{t-1})\pi_\theta^f(a_t|s_t) ,$$

and the regularized return

$$R_{\theta,T}^f(z) = \sum_{t=0}^{T-1}\gamma^t\Big(\mathsf{r}(s_t,a_t) - \lambda\frac{\pi_{\text{ref}}(a_t|s_t)}{\pi_\theta^f(a_t|s_t)}f\big(\frac{\pi_\theta^f(a_t|s_t)}{\pi_{\text{ref}}(a_t|s_t)}\big)\Big). \tag{62}$$

The finite-horizon objective is

$$J_T^f(\theta) = \sum_{z\in(\mathcal{S}\times\mathcal{A})^T}\nu_T^f(\theta; z)R_{\theta,T}^f(z) .$$

Fix $(s, b) \in \mathcal{S} \times \mathcal{A}$. Differentiating this finite-horizon objective gives

$$\frac{\partial J_T^f(\theta)}{\partial\theta(s,b)} = \underbrace{\sum_{z\in(\mathcal{S}\times\mathcal{A})^T}\frac{\partial\nu_T^f(\theta; z)}{\partial\theta(s,b)}R_{\theta,T}^f(z)}_{(\mathbf{A})} + \underbrace{\sum_{z\in(\mathcal{S}\times\mathcal{A})^T}\nu_T^f(\theta; z)\frac{\partial R_{\theta,T}^f(z)}{\partial\theta(s,b)}}_{(\mathbf{B})} . \tag{63}$$

We now treat these two terms separately.

**Term (A).** Using the log-derivative trick and the fact that the only terms that depend on $\theta$ in $\nu_T^f(\theta; z)$ are the ones that depend on the policy itself, we obtain

$$\frac{\partial\nu_T^f(\theta; z)}{\partial\theta(s,b)} = \nu_T^f(\theta; z)\sum_{t=0}^{T-1}\frac{\partial\log\pi_\theta^f(a_t|s_t)}{\partial\theta(s,b)} . \tag{64}$$

Plugging expressions (62) and (64) in (**A**) then gives

$$(\mathbf{A}) = \sum_{z\in(\mathcal{S}\times\mathcal{A})^T}\nu_T^f(\theta; z)\sum_{t=0}^{T-1}\sum_{\ell=0}^{T-1}\frac{\partial\log\pi_\theta^f(a_\ell|s_\ell)}{\partial\theta(s,b)}\gamma^t\Big(\mathsf{r}(s_t,a_t) - \lambda\frac{\pi_{\text{ref}}(a_t|s_t)}{\pi_\theta^f(a_t|s_t)}f\Big(\frac{\pi_\theta^f(a_t|s_t)}{\pi_{\text{ref}}(a_t|s_t)}\Big)\Big) .$$

Now observe that for $\ell > t$, the sum of the log-gradient derivatives over $z \in (\mathcal{S} \times \mathcal{A})^T$ is 0. Therefore (**A**) reduces to

$$(\mathbf{A}) = \sum_{z\in(\mathcal{S}\times\mathcal{A})^T}\nu_T^f(\theta; z)\sum_{t=0}^{T-1}\sum_{\ell=0}^{t}\frac{\partial\log\pi_\theta^f(a_\ell|s_\ell)}{\partial\theta(s,b)}\gamma^t\Big(\mathsf{r}(s_t,a_t) - \lambda\frac{\pi_{\text{ref}}(a_t|s_t)}{\pi_\theta^f(a_t|s_t)}f\big(\frac{\pi_\theta^f(a_t|s_t)}{\pi_{\text{ref}}(a_t|s_t)}\big)\Big) . \tag{65}$$

**Term (B).** Taking the derivative of (62), we have

$$\frac{\partial R_{\theta,T}^f(z)}{\partial\theta(s,b)} = \lambda\sum_{t=0}^{T-1}\gamma^t\Big(\frac{\partial\pi_\theta^f(a_t|s_t)}{\partial\theta(s,b)}\frac{\pi_{\text{ref}}(a_t|s_t)}{\pi_\theta^f(a_t|s_t)^2}f\big(\frac{\pi_\theta^f(a_t|s_t)}{\pi_{\text{ref}}(a_t|s_t)}\big) - \frac{\partial\pi_\theta^f(a_t|s_t)}{\partial\theta(s,b)}\frac{f'\big(\frac{\pi_\theta^f(a_t|s_t)}{\pi_{\text{ref}}(a_t|s_t)}\big)}{\pi_\theta^f(a_t|s_t)}\Big).$$

Summing over $z \in (\mathcal{S} \times \mathcal{A})^T$ and using the log-derivative trick gives

$$(\mathbf{B}) = \lambda\sum_{z\in(\mathcal{S}\times\mathcal{A})^T}\nu_T^f(\theta; z)\sum_{t=0}^{T-1}\gamma^t\frac{\partial\log\pi_\theta^f(a_t|s_t)}{\partial\theta(s,b)}\Big[\frac{\pi_{\text{ref}}(a_t|s_t)}{\pi_\theta^f(a_t|s_t)}f\big(\frac{\pi_\theta^f(a_t|s_t)}{\pi_{\text{ref}}(a_t|s_t)}\big) - f'\big(\frac{\pi_\theta^f(a_t|s_t)}{\pi_{\text{ref}}(a_t|s_t)}\big)\Big]. \tag{66}$$

Plugging the expressions (65) and (66) in (63) gives

$$\frac{\partial J_T^f(\theta)}{\partial\theta(s,b)} = \mathbb{E}\left[\sum_{t=0}^{T-1}\sum_{\ell=0}^{t}\frac{\partial\log\pi_\theta^f(A_\ell|S_\ell)}{\partial\theta(s,b)}\gamma^t\left(\mathsf{r}(S_t,A_t) - \lambda\frac{\pi_{\text{ref}}(A_t|S_t)}{\pi_\theta^f(A_t|S_t)}f\left(\frac{\pi_\theta^f(A_t|S_t)}{\pi_{\text{ref}}(A_t|S_t)}\right)\right)\right]$$

$$+ \lambda \mathbb{E}\left[\sum_{t=0}^{T-1} \gamma^t \frac{\partial \log \pi_\theta^f(A_t|S_t)}{\partial \theta(s,b)}\left(\frac{\pi_{\mathrm{ref}}(A_t|S_t)}{\pi_\theta^f(A_t|S_t)} f\left(\frac{\pi_\theta^f(A_t|S_t)}{\pi_{\mathrm{ref}}(A_t|S_t)}\right) - f'\left(\frac{\pi_\theta^f(A_t|S_t)}{\pi_{\mathrm{ref}}(A_t|S_t)}\right)\right)\right].$$

The previous term can be rewritten as

$$\frac{\partial J_T^f(\theta)}{\partial \theta(s,b)} = \mathbb{E}_{Z \sim \nu_T^f(\theta)}\left[\sum_{t=0}^{T-1}\sum_{\ell=0}^{t} \frac{\partial \log \pi_\theta^f(A_\ell|S_\ell)}{\partial \theta(s,b)}\gamma^t \mathsf{r}(S_t, A_t)\right]$$

$$- \lambda \mathbb{E}_{Z \sim \nu_T^f(\theta)}\left[\sum_{t=0}^{T-1}\sum_{\ell=0}^{t-1} \frac{\partial \log \pi_\theta^f(A_\ell|S_\ell)}{\partial \theta(s,b)}\gamma^t \frac{\pi_{\mathrm{ref}}(A_t|S_t)}{\pi_\theta^f(A_t|S_t)} f\left(\frac{\pi_\theta^f(A_t|S_t)}{\pi_{\mathrm{ref}}(A_t|S_t)}\right)\right]$$

$$- \lambda \mathbb{E}_{Z \sim \nu_T^f(\theta)}\left[\sum_{t=0}^{T-1} \gamma^t \frac{\partial \log \pi_\theta^f(A_t|S_t)}{\partial \theta(s,b)} f'\left(\frac{\pi_\theta^f(A_t|S_t)}{\pi_{\mathrm{ref}}(A_t|S_t)}\right)\right].$$

Next, we apply the tower property by taking the conditional expectation with respect to $\mathcal{G}_t :=\sigma(S_0, A_0, \ldots, S_t)$ on the second expectation and using Lemma E.1 in the third expectation, gives

$$\frac{\partial J_T^f(\theta)}{\partial \theta(s,b)} = \mathbb{E}\left[\sum_{t=0}^{T-1}\sum_{\ell=0}^{t} \frac{\partial \log \pi_\theta^f(A_\ell|S_\ell)}{\partial \theta(s,b)}\gamma^t \mathsf{r}(S_t, A_t)\right]$$

$$- \lambda \mathbb{E}\left[\sum_{t=0}^{T-1}\sum_{\ell=0}^{t-1} \frac{\partial \log \pi_\theta^f(A_\ell|S_\ell)}{\partial \theta(s,b)}\gamma^t \mathrm{D}^f(\pi_\theta^f(\cdot|S_t)\|\pi_{\mathrm{ref}}(\cdot|S_t))\right]$$

$$- \lambda \mathbb{E}\left[\sum_{t=0}^{T-1} \gamma^t \mathbf{1}_s(S_t)\frac{\mathrm{W}_\theta^f(S_t)\,\mathrm{w}_\theta^f(b|S_t)}{\pi_\theta^f(A_t|S_t)}\left[\mathbf{1}_b(A_t) - \mathrm{w}_\theta^f(A_t|S_t)\right] f'\left(\frac{\pi_\theta^f(A_t|S_t)}{\pi_{\mathrm{ref}}(A_t|S_t)}\right)\right].$$

Applying the tower property again by taking the conditional expectation with respect to $\mathcal{G}_t$ in the third expectation gives

$$\frac{\partial J_T^f(\theta)}{\partial \theta(s,b)} = \mathbb{E}\left[\sum_{t=0}^{T-1}\sum_{\ell=0}^{t} \frac{\partial \log \pi_\theta^f(A_\ell|S_\ell)}{\partial \theta(s,b)}\gamma^t \mathsf{r}(S_t, A_t)\right]$$

$$- \lambda \mathbb{E}\left[\sum_{t=0}^{T-1}\sum_{\ell=0}^{t-1} \frac{\partial \log \pi_\theta^f(A_\ell|S_\ell)}{\partial \theta(s,b)}\gamma^t \mathrm{D}^f(\pi_\theta^f(\cdot|S_t)\|\pi_{\mathrm{ref}}(\cdot|S_t))\right]$$

$$- \lambda \mathbb{E}\left[\sum_{t=0}^{T-1} \gamma^t \mathbf{1}_s(S_t)\,\mathrm{W}_\theta^f(S_t)\,\mathrm{w}_\theta^f(b|S_t)\left[f'\left(\frac{\pi_\theta^f(b|S_t)}{\pi_{\mathrm{ref}}(b|S_t)}\right) - \sum_{a \in \mathcal{A}}\mathrm{w}_\theta^f(a|S_t) f'\left(\frac{\pi_\theta^f(a|S_t)}{\pi_{\mathrm{ref}}(a|S_t)}\right)\right]\right].$$

Taking $T \to +\infty$ and applying the dominated convergence theorem concludes the proof. $\qquad\square$

The following lemma establishes a bound on the variance and bias of the stochastic estimator.

**Lemma E.3.** *Assume that, for some $\pi_{\mathrm{ref}} > 0$, $f$ satisfy $\mathbf{A}_f(\pi_{\mathrm{ref}})$. There exists a constant $\beta_f \geq 0$ such that, for any parameter $\theta \in \mathbb{R}^{|\mathcal{S}||\mathcal{A}|}$, we have*

$$\left\|\mathrm{g}^f(\theta) - \frac{\partial v_\theta^f(\rho)}{\partial \theta}\right\|_2 \leq \beta_f \ ,$$

*where $\beta_f$ is an upper bound on the bias defined as*

$$\beta_f := \frac{2\gamma^H(H+1)}{(1-\gamma)^2}\omega_f\left[2 + 2\lambda\mathrm{d}_f + \lambda(1-\gamma)\mathrm{y}_f\right] \ ,$$

*where $\omega_f$ is defined in $\mathbf{A}_f(\pi_{\mathrm{ref}})$ and $\mathrm{d}_f$, and $\mathrm{y}_f$ are defined in* (6).

*Proof.* Using the expression of the gradient truncated at $H$ from (60) and (61), of the true gradient from Lemma E.2, and the triangle inequality, we have

$$\left\|\mathrm{g}^f(\theta) - \frac{\partial v_\theta^f(\rho)}{\partial \theta}\right\|_2 \leq \sum_{t=H}^{\infty}\sum_{\ell=0}^{t} \gamma^t \left\|\mathbb{E}_\rho^{\pi_\theta^f}\left[\frac{\partial \log \pi_\theta^f(A_\ell|S_\ell)}{\partial \theta}\mathsf{r}(S_t, A_t)\right]\right\|_2$$

$$+ \sum_{t=H}^{\infty} \sum_{\ell=0}^{t-1} \lambda \gamma^t \left\| \mathbb{E}_\rho^{\pi_\theta^f} \left[ \frac{\partial \log \pi_\theta^f(A_\ell|S_\ell)}{\partial \theta} \, \mathrm{D}^f(\pi_\theta^f(\cdot|S_t) \| \pi_{\mathrm{ref}}(\cdot|S_t)) \right] \right\|_2 + \lambda \sum_{t=H}^{\infty} \gamma^t \left\| \mathbb{E}_\rho^{\pi_\theta^f} \left[ \mathrm{F}_\theta^f(S_t) \right] \right\|_2 .$$

Next, applying Lemma E.1 combined with the triangle inequality yields

$$\left\| \mathrm{g}^f(\theta) - \frac{\partial v_\theta^f(\rho)}{\partial \theta} \right\|_2 \leq \sum_{t=H}^{\infty} \sum_{\ell=0}^{t} \gamma^t \mathbb{E}_\rho^{\pi_\theta^f} \left[ \frac{2 \, \mathrm{W}_\theta^f(S_\ell) \, \mathrm{w}_\theta^f(A_\ell|S_\ell)}{\pi_\theta^f(A_\ell|S_\ell)} \, |\mathrm{r}(S_t, A_t)| \right]$$
$$+ \sum_{t=H}^{\infty} \sum_{\ell=0}^{t-1} \lambda \gamma^t \mathbb{E}_\rho^{\pi_\theta^f} \left[ \frac{2 \, \mathrm{W}_\theta^f(S_\ell) \, \mathrm{w}_\theta^f(A_\ell|S_\ell)}{\pi_\theta^f(A_\ell|S_\ell)} \, \mathrm{D}^f(\pi_\theta^f(\cdot|S_t) \| \pi_{\mathrm{ref}}(\cdot|S_t)) \right]$$
$$+ \lambda \sum_{t=H}^{\infty} \gamma^t \mathbb{E}_\rho^{\pi_\theta^f} \left[ 2 \, \mathrm{W}_\theta^f(S_t) \, \mathrm{Y}_\theta^f(S_t) \right] .$$

We define the following filtration, for $t \geq 0$,

$$\mathcal{G}_t = \sigma(S_0, A_0, \ldots, S_t) ,$$

Next, applying the tower property of the conditional expectation by conditioning on $\mathcal{G}_t$, bounding the reward and the divergence respectively by 1, and $\max_{s \in \mathcal{S}} \sup_{\nu \in \mathcal{P}(\mathcal{A})} \mathrm{D}^f(\nu \| \pi_{\mathrm{ref}}(\cdot|s))\}$, and using that $\mathrm{w}_\theta^f(\cdot|s) \in \mathcal{P}(\mathcal{A})$ for any $s \in \mathcal{S}$ yields

$$\left\| \mathrm{g}^f(\theta) - \frac{\partial v_\theta^f(\rho)}{\partial \theta} \right\|_2 \leq 2 \sum_{t=H}^{\infty} \sum_{\ell=0}^{t} \gamma^t \left\| \mathrm{W}_\theta^f \right\|_\infty$$
$$+ 2\lambda \sum_{t=H}^{\infty} \sum_{\ell=0}^{t-1} \gamma^t \left\| \mathrm{W}_\theta^f \right\|_\infty \sup_{(s,\nu) \in \mathcal{S} \times \mathcal{P}(\mathcal{A})} \mathrm{D}^f(\nu \| \pi_{\mathrm{ref}}(\cdot|s))\} + 2\lambda \sum_{t=H}^{\infty} \gamma^t \left\| \mathrm{W}_\theta^f \right\|_\infty \left\| \mathrm{Y}_\theta^f \right\|_\infty .$$

Finally, using that

$$\sum_{t=H}^{\infty} \gamma^t \leq \frac{\gamma^H}{1-\gamma} , \quad \sum_{t=H}^{\infty} \gamma^t(t-1) \leq 2\gamma^H \frac{H}{(1-\gamma)^2} , \quad \sum_{t=H}^{\infty} \gamma^t t \leq 2\gamma^H \frac{H+1}{(1-\gamma)^2} ,$$

combined with Lemma B.6 completes the proof. $\qquad \square$

**Lemma E.4.** *Assume that, for some $\underline{\pi}_{\mathrm{ref}} > 0$, $f$ satisfy $\mathbf{A}_f(\underline{\pi}_{\mathrm{ref}})$. For any $\theta \in \mathbb{R}^{|\mathcal{S}||\mathcal{A}|}$, it holds that*

$$\mathbb{E}_{Z \sim [\nu(\theta)]^{\otimes B}} \left[ \left\| \mathrm{g}^f(\theta) - \mathrm{g}_Z^f(\theta) \right\|_2^2 \right] \leq \frac{\sigma_f^2}{B} ,$$

*where we have defined*

$$\sigma_f^2 := \frac{12}{(1-\gamma)^4} \left[ \omega_f^3 + \lambda^2 \gamma^2 \omega_f^3 \mathrm{d}_f^2 + \lambda^2 (1-\gamma)^2 \omega_f^2 \mathrm{y}_f^2 \right] ,$$

*and where $\omega_f$, $\mathrm{d}_f$, and $\mathrm{y}_f$ are defined in $\mathbf{A}_f(\underline{\pi}_{\mathrm{ref}})$ and (6).*

*Proof.* Firstly, define for $\xi = (s_h, a_h)_{h=0}^{H-1} \in (\mathcal{S} \times \mathcal{A})^H$

$$u_\xi(\theta) := \sum_{h=0}^{H-1} \sum_{\ell=0}^{h} \frac{\partial \log \pi_\theta^f(a_\ell|s_\ell)}{\partial \theta} \gamma^h \mathrm{r}(s_h, a_h)$$
$$- \lambda \sum_{h=0}^{H-1} \sum_{\ell=0}^{h-1} \frac{\partial \log \pi_\theta^f(a_\ell|s_\ell)}{\partial \theta} \gamma^h \, \mathrm{D}^f(\pi_\theta^f(\cdot|s_h) \| \pi_{\mathrm{ref}}(\cdot|s_h)) - \lambda \sum_{h=0}^{H-1} \gamma^h \mathrm{F}_\theta^f(s_h) ,$$

Importantly, for a given $Z \sim [\nu(\theta)]^{\otimes B}$, denoting by $Z = (Z_0, \ldots, Z_{B-1})$, it holds that

$$\mathrm{g}_Z^f(\theta) = \frac{1}{B} \sum_{b=0}^{B-1} u_{Z_b}(\theta) , \quad \text{and } \mathrm{g}^f(\theta) = \mathbb{E}_{\mathfrak{T} \sim \nu(\theta)} \left[ u_\mathfrak{T}(\theta) \right] .$$

Using that the variables $(Z_0, \ldots, Z_{B-1})$ are independent and identically distributed, we get

$$\mathbb{E}_{Z \sim [\nu(\theta)]^{\otimes B}}\left[\left\|\mathrm{g}^f(\theta) - \mathrm{g}_Z^f(\theta)\right\|_2^2\right] = \mathbb{E}_{Z \sim [\nu(\theta)]^{\otimes B}}\left[\left\|\frac{1}{B}\sum_{b=0}^{B-1} u_{Z_b}(\theta) - \mathrm{g}^f(\theta)\right\|_2^2\right]$$

$$= \frac{1}{B}\mathbb{E}_{\mathfrak{T} \sim \nu(\theta)}\left[\left\|u_{\mathfrak{T}}(\theta) - \mathrm{g}^f(\theta)\right\|_2^2\right]$$

$$\leq \frac{1}{B}\mathbb{E}_{\mathfrak{T} \sim \nu(\theta)}\left[\|u_{\mathfrak{T}}(\theta)\|_2^2\right] \quad , \tag{67}$$

where in the last inequality, we used that the second moment of a random variable dominates its variance. Next, using Jensen's inequality combined with the convexity of the square function, we have

$$\mathbb{E}_{\mathfrak{T} \sim \nu(\theta)}\left[\|u_{\mathfrak{T}}(\theta)\|_2^2\right] \leq 3\mathbb{E}_{\mathfrak{T}}\left[\left\|\sum_{h=0}^{H-1}\sum_{\ell=0}^{h}\frac{\partial \log \pi_\theta^f(A_\ell|S_\ell)}{\partial \theta}\gamma^h \mathrm{r}(S_h, A_h)\right\|_2^2\right]$$

$$+ 3\lambda^2 \mathbb{E}_{\mathfrak{T} \sim \nu(\theta)}\left[\left\|\sum_{h=0}^{H-1}\sum_{\ell=0}^{h-1}\frac{\partial \log \pi_\theta^f(A_\ell|S_\ell)}{\partial \theta}\gamma^h \mathrm{D}^f(\pi_\theta^f(\cdot|S_h)\|\pi_{\mathrm{ref}}(\cdot|S_h))\right\|_2^2\right]$$

$$+ 3\lambda^2 \mathbb{E}_{\mathfrak{T} \sim \nu(\theta)}\left[\left\|\sum_{h=0}^{H-1}\gamma^h \mathrm{F}_\theta^f(S_h)\right\|_2^2\right] \quad ,$$

Applying the triangle inequality and the fact that the reward and the divergence are positive yields

$$\mathbb{E}_{\mathfrak{T} \sim \nu(\theta)}\left[\|u_{\mathfrak{T}}(\theta)\|_2^2\right] \leq 3\mathbb{E}_{\mathfrak{T} \sim \nu(\theta)}\left[\left(\sum_{h=0}^{H-1}\sum_{\ell=0}^{h}\gamma^h\left\|\frac{\partial \log \pi_\theta^f(A_\ell|S_\ell)}{\partial \theta}\right\|_2 \mathrm{r}(S_h, A_h)\right)^2\right]$$

$$+ 3\lambda^2 \mathbb{E}_{\mathfrak{T} \sim \nu(\theta)}\left[\left(\sum_{h=0}^{H-1}\sum_{\ell=0}^{h-1}\gamma^h\left\|\frac{\partial \log \pi_\theta^f(A_\ell|S_\ell)}{\partial \theta}\right\|_2 \mathrm{D}^f(\pi_\theta^f(\cdot|S_h)\|\pi_{\mathrm{ref}}(\cdot|S_h))\right)^2\right]$$

$$+ 3\lambda^2 \mathbb{E}_{\mathfrak{T} \sim \nu(\theta)}\left[\left(\sum_{h=0}^{H-1}\gamma^h\left\|\mathrm{F}_\theta^f(S_h)\right\|_2\right)^2\right] \quad ,$$

Combining Lemma E.1 and the fact that the reward is bounded between $0$ and $1$ gives

$$\mathbb{E}_{\mathfrak{T}}\left[\|u_{\mathfrak{T}}(\theta)\|_2^2\right] \leq 3\mathbb{E}_{\mathfrak{T} \sim \nu(\theta)}\left[\left(\sum_{h=0}^{H-1}\sum_{\ell=0}^{h}2\gamma^{h/2} \cdot \gamma^{h/2}\frac{\mathrm{W}_\theta^f(S_\ell)\,\mathrm{w}_\theta^f(A_\ell|S_\ell)}{\pi_\theta^f(A_\ell|S_\ell)}\right)^2\right]$$

$$+ 3\lambda^2 \mathbb{E}_{\mathfrak{T} \sim \nu(\theta)}\left[\left(\sum_{h=0}^{H-1}\sum_{\ell=0}^{h-1}2\gamma^{h/2} \cdot \gamma^{h/2}\frac{\mathrm{W}_\theta^f(S_\ell)\,\mathrm{w}_\theta^f(A_\ell|S_\ell)}{\pi_\theta^f(A_\ell|S_\ell)}\sup_{(s,\nu) \in \mathcal{S} \times \mathcal{P}(\mathcal{A})}\mathrm{D}^f(\nu\|\pi_{\mathrm{ref}}(\cdot|s))\right)^2\right]$$

$$+ 3\lambda^2 \mathbb{E}_{\mathfrak{T} \sim \nu(\theta)}\left[\left(\sum_{h=0}^{H-1}2\gamma^{h/2} \cdot \gamma^{h/2}\left\|\mathrm{W}_\theta^f\right\|_\infty\left\|\mathrm{Y}_\theta^f\right\|_\infty\right)^2\right] \quad ,$$

Next, applying the Cauchy-Schwarz inequality gives

$$\mathbb{E}_{\mathfrak{T} \sim \nu(\theta)}\left[\|u_{\mathfrak{T}}(\theta)\|_2^2\right] \leq 3\mathbb{E}_{\mathfrak{T} \sim \nu(\theta)}\left[\left(\sum_{h=0}^{H-1}\sum_{\ell=0}^{h}4\gamma^h\right)\left(\sum_{h=0}^{H-1}\sum_{\ell=0}^{h}\gamma^h\frac{\mathrm{W}_\theta^f(S_\ell)^2\,\mathrm{w}_\theta^f(A_\ell|S_\ell)^2}{\pi_\theta^f(A_\ell|S_\ell)^2}\right)\right]$$

$$+ 3\lambda^2 \mathbb{E}_{\mathfrak{T} \sim \nu(\theta)}\left[\sum_{h=0}^{H-1}\sum_{\ell=0}^{h-1}4\gamma^h\left(\sum_{h=0}^{H-1}\sum_{\ell=0}^{h-1}\gamma^h\frac{\mathrm{W}_\theta^f(S_\ell)^2\,\mathrm{w}_\theta^f(A_\ell|S_\ell)^2}{\pi_\theta^f(A_\ell|S_\ell)^2}\sup_{(s,\nu) \in \mathcal{S} \times \mathcal{P}(\mathcal{A})}\mathrm{D}^f(\nu\|\pi_{\mathrm{ref}}(\cdot|s))^2\right)\right]$$

$$+ 3\lambda^2 \mathbb{E}_{\mathfrak{T} \sim \nu(\theta)}\left[\sum_{h=0}^{H-1}4\gamma^h\left(\sum_{h=0}^{H-1}\gamma^h\left\|\mathrm{W}_\theta^f\right\|_\infty^2\left\|\mathrm{Y}_\theta^f\right\|_\infty^2\right)\right] \quad ,$$

We define the following filtration, for $t \geq 0$,

$$\mathcal{G}_t = \sigma(S_0, A_0, \dots, S_t) \ ,$$

Next, applying the tower property of the conditional expectation by conditioning on $\mathcal{G}_t$, and using that $w_\theta^f(\cdot|s) \in \mathcal{P}(\mathcal{A})$ for any $s \in \mathcal{S}$ yields

$$\mathbb{E}_{\mathfrak{T} \sim \nu(\theta)} \left[ \|u_{\mathfrak{T}}(\theta)\|_2^2 \right] \leq 12 \left\| W_\theta^f \right\|_\infty^2 \max_{(s,a) \in \mathcal{S} \times \mathcal{A}} \left\{ \frac{w_\theta^f(a|s)}{\pi_\theta^f(a|s)} \right\} \left( \sum_{h=0}^{H-1} \gamma^h(h+1) \right)^2$$

$$+ 12\lambda^2 \left\| W_\theta^f \right\|_\infty^2 \max_{(s,a) \in \mathcal{S} \times \mathcal{A}} \left\{ \frac{w_\theta^f(a|s)}{\pi_\theta^f(a|s)} \right\} \sup_{(s,\nu) \in \mathcal{S} \times \mathcal{P}(\mathcal{A})} \left\{ D^f(\nu \| \pi_{\mathrm{ref}}(\cdot|s))^2 \right\} \left( \sum_{h=0}^{H-1} \gamma^h h \right)^2$$

$$+ 12\lambda^2 \left\| W_\theta^f \right\|_\infty^2 \left\| Y_\theta^f \right\|_\infty^2 \left( \sum_{h=0}^{H-1} \gamma^h \right)^2 \ .$$

Next using

$$\sum_{t=0}^{H-1} \gamma^t \leq \frac{1}{1-\gamma} \ , \quad \sum_{t=0}^{H-1} \gamma^t t \leq \frac{\gamma}{(1-\gamma)^2} \ , \sum_{t=0}^{H-1} \gamma^t(t+1) \leq \frac{1}{(1-\gamma)^2} \ ,$$

and plugging in the obtained bound in (67), combined with Lemma B.6 concludes the proof. $\square$

### E.2  SAMPLE COMPLEXITY OF STOCHASTIC $f$-PG

We now derive convergence rates for $f$-PG. First, we define the following quantity, which will be the Polyak-Łojasiewicz constant of our function over the optimization space, where policies are guaranteed not to be too ill-conditioned, i.e., all their entries are larger than the $\tau_\lambda$ defined in (58),

$$\underline{\mu}_f := \frac{\lambda(1-\gamma)\rho_{\min}^2 \zeta_f^2}{\omega_f^2} \underline{\pi_{\mathrm{ref}}}^2 \min_{x \in [\tau_\lambda, \frac{1}{\pi_{\mathrm{ref}}}]} f''(x)^{-2} \ . \tag{68}$$

As we will prove in this subsection, this quantity represents a lower bound of the non-uniform Łojasiewicz coefficient along the trajectory. In the following lemma, we give a simpler lower bound of $\underline{\mu}_f$ provided that $\lambda$ is not too large.

**Lemma E.5.** *Assume that, for some $\underline{\pi_{\mathrm{ref}}} > 0$, $f$ and $\pi_{\mathrm{ref}}$ satisfy $\mathbf{A}_f(\underline{\pi_{\mathrm{ref}}})$ and $\mathbf{P}(\underline{\pi_{\mathrm{ref}}})$ respectively. Assume in addition that the initial distribution $\rho$ satisfies $\mathbf{A}_\rho$ and that $\lambda$ satisfies*

$$\lambda \leq \frac{4}{(1-\gamma)^2 \rho_{\min}} \min \left( \frac{4}{|f'(\iota_f)|}, \frac{1}{|f'(\frac{1}{2})|}, \frac{4}{|f'(\frac{1}{2}\underline{\pi_{\mathrm{ref}}})|} \right) \ .$$

*In this case, it holds that*

$$\underline{\mu}_f = \frac{\lambda(1-\gamma)\rho_{\min}^2 \zeta_f^2}{\omega_f^2} \underline{\pi_{\mathrm{ref}}}^2 (f^\star)'' \left( -\frac{16 + 8\gamma\lambda d_f}{\lambda(1-\gamma)^2 \rho_{\min}} \right)^2 \ .$$

*Additionally if $\lambda \leq 1/d_f$, then it holds that*

$$\underline{\mu}_f \geq \frac{\lambda(1-\gamma)\rho_{\min}^2 \zeta_f^2}{\omega_f^2} \underline{\pi_{\mathrm{ref}}}^2 (f^\star)'' \left( -\frac{24}{\lambda(1-\gamma)^2 \rho_{\min}} \right)^2 \ .$$

*Proof.* First note that the first condition on $\lambda$ implies that $\tau_\lambda < \iota_f$, with $\iota_f$ defined in $\mathbf{A}_f(\underline{\pi_{\mathrm{ref}}})$, and thus

$$\min_{x \in [\tau_\lambda, \frac{1}{\pi_{\mathrm{ref}}}]} f''(x)^{-2} = f''(\tau_\lambda)^{-2}.$$

Additionally, the second and third conditions on $\lambda$ guarantee that the minimum of $\tau_\lambda$ in (58) is attained in the first term, that is

$$\tau_\lambda = [f']^{-1}\left(-\frac{16 + 8\gamma\lambda d_f}{\lambda(1-\gamma)^2 \rho_{\min}}\right) \ .$$

Finally, we recall that the convex conjugate of the divergence generator $f$ defined in (25) satisfies, for any $y \in (-\infty, f'(\frac{1}{\pi_{\mathrm{ref}}}))$,

$$(f^\star)''(y) = \frac{1}{f''([f']^{-1}(y))} \ .$$

Thus, we obtain that $f''(\tau_\lambda)^{-2} = (f^\star)''(\frac{-16 + 8\gamma\lambda d_f}{\lambda(1-\gamma)^2 \rho_{\min}})^2$ which concludes the proof. $\qquad\square$

In the following, we define the filtration adapted to the iterates of $f$-PG as

$$\mathcal{F}_t := \sigma\Big(Z_t : t \in \{0, \ldots, T-1\}\Big) \ .$$

The following theorem gives convergence rates of $f$-PG.

**Theorem E.6.** *Assume that, for some $\pi_{\mathrm{ref}} > 0$, $f$ and $\pi_{\mathrm{ref}}$ satisfy $\mathbf{A}_f(\pi_{\mathrm{ref}})$ and $\mathbf{P}(\pi_{\mathrm{ref}})$ respectively. Assume in addition that the initial distribution $\rho$ satisfies $\mathbf{A}_\rho$. Fix $\eta \leq 1/2L_f$, a given temperature $\lambda$, and consider the iterates $(\theta_t)_{t=0}^\infty$ of the algorithm $f$-PG. It holds almost surely that*

$$\inf_{t \geq 0} \mu_f(\theta_t) \geq \underline{\mu}_f \ . \tag{69}$$

*Additionally, for any $t \geq 0$ we have that*

$$v_\star^f(\rho) - \mathbb{E}\left[v_{\theta_t}^f(\rho)\right] \leq (1 - \underline{\mu}_f \eta/4)^t (v_\star^f(\rho) - v_{\theta_0}^f(\rho)) + \frac{6\eta\sigma_f^2}{B\underline{\mu}_f} + \frac{6\beta_f^2}{\underline{\mu}_f} \ .$$

*Proof.* Recall that for any $t \in [T]$, we have that

$$\theta_t = \mathcal{T}_{\tau_\lambda}(\bar{\theta}_t) \ .$$

Hence, by Lemma D.2, for any $t \in [T]$ it holds that

$$\pi_{\theta_t}^f \geq \tau_\lambda \underline{\pi_{\mathrm{ref}}} \ .$$

Combining the previous inequality with the expression of the coefficient $\mu_f$ provided in Theorem C.1 proves the first statement of the lemma. Next, using Theorem B.12 gives

$$v_{\theta_{t+1}}^f(\rho) \geq v_{\theta_t}^f(\rho) + 2\eta\langle\nabla v_{\theta_t}^f(\rho), \mathrm{g}_{Z_t}^f(\theta_t)\rangle - \frac{\eta^2 L_f}{2}\left\|\mathrm{g}_{Z_t}^f(\theta_t)\right\|_2^2 \ .$$

Next, taking the conditional expectation with respect to $\mathcal{F}_t$ and adding and subtracting $\nabla v_{\theta_t}^f(\rho)$ in the dot product gives

$$\mathbb{E}\left[v_{\theta_{t+1}}^f(\rho)\Big|\mathcal{F}_t\right] \geq v_{\theta_t}^f(\rho) + 2\eta\left\|\nabla v_{\theta_t}^f(\rho)\right\|^2 + \underbrace{2\eta\langle\nabla v_{\theta_t}^f(\rho), \mathrm{g}^f(\theta_t) - \nabla v_{\theta_t}^f(\rho)\rangle}_{(\mathbf{K_1})}$$
$$- \frac{\eta^2 L_f}{2}\underbrace{\mathbb{E}\left[\left\|\mathrm{g}_{Z_t}^f(\theta_t)\right\|_2^2\Big|\mathcal{F}_t\right]}_{(\mathbf{K_2})} \ . \tag{70}$$

We now bound each of these terms separately.

**Bounding $\mathbf{K_1}$.** Using the Cauchy-Schwarz inequality, yields

$$2\eta\langle\nabla v_{\theta_t}^f(\rho), \mathrm{g}^f(\theta_t) - \nabla v_{\theta_t}^f(\rho)\rangle \geq -2\eta\left\|\nabla v_{\theta_t}^f(\rho)\right\|_2\left\|\mathrm{g}^f(\theta_t) - \nabla v_{\theta_t}^f(\rho)\right\|_2$$
$$= -2 \cdot \eta^{1/2}\left\|\nabla v_{\theta_t}^f(\rho)\right\|_2 \cdot \eta^{1/2}\beta_f \ ,$$

where in the last inequality we used Lemma E.3. Next, using Young's inequality gives

$$2\eta\langle\nabla v_{\theta_t}^f(\rho), \mathrm{g}^f(\theta_t) - \nabla v_{\theta_t}^f(\rho)\rangle \geq -\eta\left\|\nabla v_{\theta_t}^f(\rho)\right\|_2^2 - \eta\beta_f^2 \ . \tag{71}$$

**Bounding $K_2$.** Using the convexity of the square function with Jensen's inequality gives

$$\mathbb{E}\left[\left\|g_{Z_t}^f(\theta_t)\right\|_2^2 \bigg| \mathcal{F}_t\right] = \mathbb{E}\left[\left\|g_{Z_t}^f(\theta_t) - g^f Z_t(\theta_t) + g^f Z_t(\theta_t) - \nabla v_{\theta_t}^f(\rho) + \nabla v_{\theta_t}^f(\rho)\right\|_2^2 \bigg| \mathcal{F}_t\right]$$

$$\leq 3\beta_f^2 + 3\left\|\nabla v_{\theta_t}^f(\rho)\right\|_2^2 + \frac{3\sigma_f^2}{B} \quad, \tag{72}$$

where we used Lemma E.3 and Lemma E.4. Plugging in the bounds (71) on $K_1$ and (72) on $K_2$ in (70) gives

$$\mathbb{E}\left[v_{\theta_{t+1}}^f(\rho)\bigg|\mathcal{F}_t\right] \geq v_{\theta_t}^f(\rho) + \eta\left\|\nabla v_{\theta_t}^f(\rho)\right\|^2 - \left(\eta + \frac{3\eta^2 L_f}{2}\right)\beta_f^2 - \frac{3\eta^2 L_f}{2}\left\|\nabla v_{\theta_t}^f(\rho)\right\|_2^2 - \frac{3\eta^2 L_f \sigma_f^2}{2B} \quad.$$

Taking the expectation with respect to all the stochasticity, multiplying both sides by $-1$, and adding $v_\star^f(\rho)$ gives

$$v_\star^f(\rho) - \mathbb{E}\left[v_{\theta_{t+1}}^f(\rho)\right] \leq v_\star^f(\rho) - \mathbb{E}\left[v_{\theta_t}^f(\rho)\right] - \eta\left(1 - \frac{3\eta L_f}{2}\right)\left\|\nabla v_{\theta_t}^f(\rho)\right\|^2$$

$$+ \left(\eta + \frac{3\eta^2 L_f}{2}\right)\beta_f^2 + \frac{3\eta^2 L_f \sigma_f^2}{2B} \quad.$$

Next using Theorem C.1 combined with (69) yields

$$v_\star^f(\rho) - \mathbb{E}\left[v_{\theta_{t+1}}^f(\rho)\right] \leq \left(1 - \eta\underline{\mu}_f\left(1 - \frac{3\eta L_f}{2}\right)\right)\left(v_\star^f(\rho) - \mathbb{E}\left[v_{\theta_t}^f(\rho)\right]\right)$$

$$+ 2\eta\beta_f^2 + \frac{3\eta^2 L_f \sigma_f^2}{2B} \quad,$$

where we used $3\eta L_f/2 \leq 1$. Finally, using that $\eta \leq 1/2L_f$ to bound $1 - 3\eta L_f/2$ and unrolling the recursion concludes the proof. $\qquad \square$

Next, we provide the sample complexity of $f$-`PG` for solving the $f$-regularized objective.

**Corollary E.7.** *Assume that, for some $\pi_{\mathrm{ref}} > 0$, $f$ and $\pi_{\mathrm{ref}}$ satisfy $A_f(\pi_{\mathrm{ref}})$ and $P(\pi_{\mathrm{ref}})$ respectively. Assume in addition that the initial distribution $\rho$ satisfies $A_\rho$. Fix any $\epsilon > 0$ and $\lambda > 0$. Setting*

$$H \geq \frac{4}{(1-\gamma)^2} + \frac{1}{1-\gamma}\log\left(\frac{216\omega_f^2}{\epsilon\underline{\mu}_f(1-\gamma)^4}\left[4 + 4\lambda^2 d_f^2 + \lambda^2(1-\gamma)^2 y_f^2\right]\right), \tag{73}$$

*and*

$$\eta \leq \min\left(\frac{1}{2L_f}, \frac{\epsilon B\underline{\mu}_f}{18\sigma_f^2}\right) \quad, \tag{74}$$

*and*

$$T \geq \frac{4}{\underline{\mu}_f}\max\left(2L_f, \frac{18\sigma_f^2}{\epsilon B\underline{\mu}_f}\right)\cdot\log\left(\frac{3(v_\star^f(\rho) - v_{\theta_0}^f(\rho))}{\epsilon}\right) \quad, \tag{75}$$

*guarantees that*

$$v_\star^f(\rho) - \mathbb{E}\left[v_{\theta_t}^f(\rho)\right] \leq \epsilon \quad.$$

*Proof.* As $\eta \leq 1/2L_f$, then by using Theorem E.6, it holds that

$$v_\star^f(\rho) - \mathbb{E}\left[v_{\theta_T}^f(\rho)\right] \leq \underbrace{(1 - \underline{\mu}_f\eta/4)^T(v_\star^f(\rho) - v_{\theta_0}^f(\rho))}_{(\mathbf{U})} + \underbrace{\frac{6\eta\sigma_f^2}{B\underline{\mu}_f}}_{(\mathbf{V})} + \underbrace{\frac{6\beta_f^2}{\underline{\mu}_f}}_{(\mathbf{W})} \quad.$$

Next, we aim to show that under our conditions on $T$, $H$, and $\eta$, each of these terms is smaller than $\epsilon/3$.

**Bounding V.** We start with the term **V**, which gives a condition on the step-size. In particular, setting

$$\eta \leq \frac{\epsilon B \underline{\mu}_f}{18 \sigma_f^2} \ ,$$

guarantees that $\mathbf{V} \leq \epsilon/3$, which, together with $\eta \leq 1/(2L_f)$, gives the condition (74).

**Bounding U.** In order to ensure that **U** is smaller then $\epsilon/3$, we need $T$ to satisfy

$$T \geq \frac{4}{\eta \underline{\mu}_f} \log \left( \frac{\epsilon}{3 \left( v_\star^f(\rho) - v_{\theta_0}^f(\rho) \right)} \right) \ ,$$

which, combined with the inequality $\log(1+x) \leq x$ for $x > -1$, ensures that **U** is smaller than $\epsilon/3$, and the condition (75) follows from (74).

**Bounding W.** Using Lemma E.3 and Jensen's inequality, it holds that

$$\mathbf{W} \leq \frac{6}{\underline{\mu}_f} \cdot \frac{4\gamma^{2H}(H+1)^2}{(1-\gamma)^4} \omega_f^2 \left[ 12 + 12\lambda^2 \mathrm{d}_f^2 + 3\lambda^2(1-\gamma)^2 \mathrm{y}_f^2 \right] \ ,$$

Next, we remark that for any $a > 0$, we have $a\gamma^{2H}(H+1)^2 \leq \epsilon/3$ for

$$H \geq \frac{1}{1-\gamma} \max \left( \frac{4}{1-\gamma}, \log \left( \frac{3a}{\epsilon} \right) \right) \ .$$

Taking $a = \frac{6}{\underline{\mu}_f} \cdot \frac{4}{(1-\gamma)^4} \omega_f^2 \left[ 12 + 12\lambda^2 \mathrm{d}_f^2 + 3\lambda^2(1-\gamma)^2 \mathrm{y}_f^2 \right]$ gives $\mathbf{W} < \epsilon/3$, provided that (73) holds. $\qquad \square$

### E.3 GUARANTEES ON THE NON-REGULARIZED PROBLEM

A key criterion for evaluating the quality of a reinforcement learning algorithm is its sample efficiency in solving the original, unregularized objective. To this end, we recall a result from Geist et al. (2019), which establishes a connection between regularized and unregularized value functions, and further characterizes the performance of the optimal $f$-regularized policy when evaluated in the original unregularized MDP.

**Lemma E.8** (Proposition 3 and Theorem 2 of Geist et al. (2019))**.** *For any policy $\pi$, and state $s \in \mathcal{S}$ it holds that*

$$\left| v_\pi^f(s) - v_\pi(s) \right| \leq \frac{\lambda \mathrm{d}_f}{1-\gamma} \ .$$

*Additionally, denote by $\pi_\star^f$ the optimal regularized policy. For any state $s \in \mathcal{S}$, It holds that*

$$v_\star^f(s) - v_{\pi_\star^f} \leq \frac{\lambda \mathrm{d}_f}{1-\gamma} \ .$$

The following theorem gives the convergence rate for the non-regularised problem.

**Theorem E.9.** *Assume that, for some $\pi_{\mathrm{ref}} > 0$, $f$ and $\pi_{\mathrm{ref}}$ satisfy $\mathbf{A}_f(\pi_{\mathrm{ref}})$ and $\mathbf{P}(\pi_{\mathrm{ref}})$ respectively. Assume in addition that the initial distribution $\rho$ satisfies $\mathbf{A}_\rho$. Fix $\eta \leq 1/2L_f$, a given temperature $\lambda$, and consider the iterates $(\theta)_{t=0}^\infty$ of the algorithm $f\text{-PG}$. For any $t \geq 0$ we have that*

$$\mathbb{E}\left[ v_\star(\rho) - v_{\theta_t}(\rho) \right] \leq (1 - \underline{\mu}_f \eta/4)^t (v_\star^f(\rho) - v_{\theta_0}^f(\rho)) + \frac{6\eta\sigma_f^2}{B\underline{\mu}_f} + \frac{6\beta_f^2}{\underline{\mu}_f} + \frac{2\lambda \mathrm{d}_f}{1-\gamma} \ .$$

*Proof.* The proof holds from Theorem E.6 and Lemma E.8. $\qquad \square$

Finally, we give the sample complexity of $f\text{-PG}$ to solve the unregularised problem.

**Corollary E.10.** *Assume that, for some $\pi_{\mathrm{ref}} > 0$, $f$ and $\pi_{\mathrm{ref}}$ satisfy $\mathbf{A}_f(\pi_{\mathrm{ref}})$ and $\mathbf{P}(\pi_{\mathrm{ref}})$ respectively. Assume in addition that the initial distribution $\rho$ satisfies $\mathbf{A}_\rho$. Consider any constant $\mathrm{c}_f > 0$ such that*

$$\mathrm{c}_f \leq \min\left(\frac{1}{\mathrm{d}_f}, \frac{1}{\mathrm{y}_f}, 1\right) \ .$$

*Fix any $(1-\gamma)^{-1} > \epsilon > 0$ such that*

$$\epsilon < \frac{16}{(1-\gamma)^3 \rho_{\min}} \min\left(\frac{4}{|f'(\iota_f)|}, \frac{1}{|f'(\frac{1}{2})|}, \frac{4}{|f'(\frac{1}{2}\pi_{\mathrm{ref}})|}\right) \ , \quad \text{and set } \lambda = \frac{(1-\gamma)\epsilon}{4}\mathrm{c}_f \ . \tag{76}$$

*Define*

$$d(\epsilon) = (f^\star)''\left(\frac{-96}{\epsilon \mathrm{c}_f(1-\gamma)^3 \rho_{\min}}\right)^2 \ , \quad \ell(\epsilon) = \log\left(\frac{6(v_\star^f(\rho) - v_{\theta_0}^f(\rho))}{\epsilon}\right) \ . \tag{77}$$

*Additionally, define the three following constants which depend only on $f$ as*

$$C_f^{(1)} = \frac{16\omega_f^2}{\mathrm{c}_f \zeta_f^2} \ , \quad C_f^{(2)} = 48\omega_f\left(\gamma\omega_f + (1-\gamma)\kappa_f\right) \ , \quad C_f^{(3)} = \frac{3456\omega_f^5}{\zeta_f^2 \mathrm{c}_f} \ . \tag{78}$$

*Setting*

$$H \geq \frac{4}{(1-\gamma)^2} + \frac{1}{1-\gamma}\log\left(\frac{297\omega_f^2 C_1^f}{\epsilon d(\epsilon)(1-\gamma)^6 \rho_{\min}^2 \pi_{\mathrm{ref}}^2}\right) \ ,$$

*and*

$$\eta \leq \min\left(\frac{(1-\gamma)^3}{C_f^{(2)}}, \frac{\epsilon^2 d(\epsilon)(1-\gamma)^6 B\rho_{\min}^2 \pi_{\mathrm{ref}}^2}{C_f^{(3)}}\right) \ , \tag{79}$$

*and*

$$T \geq \frac{16 C_f^{(1)}\ell(\epsilon)}{\epsilon d(\epsilon)(1-\gamma)^2 \rho_{\min}^2 \pi_{\mathrm{ref}}^2}\max\left(\frac{C_f^{(2)}}{(1-\gamma)^3}, \frac{C_f^{(3)}}{\epsilon^2 d(\epsilon)(1-\gamma)^6 B\rho_{\min}^2 \pi_{\mathrm{ref}}^2}\right) \ , \tag{80}$$

*guarantees that*

$$v_\star(\rho) - \mathbb{E}\left[v_{\theta_t}(\rho)\right] \leq \epsilon \ .$$

*Proof.* First, note that the conditions (76) on $\epsilon$ and $\lambda$ guarantee that

$$\lambda \leq \frac{4}{(1-\gamma)^2 \rho_{\min}}\min\left(\frac{4}{|f'(\iota_f)|}, \frac{1}{|f'(\frac{1}{2})|}, \frac{4}{|f'(\frac{1}{2}\pi_{\mathrm{ref}})|}\right) \ ,$$

Thus, using Lemma E.5, and the fact that $\epsilon < (1-\gamma)^{-1}$, we have that

$$\underline{\mu}_f \geq \frac{\lambda(1-\gamma)\rho_{\min}^2 \zeta_f^2}{\omega_f^2}\pi_{\mathrm{ref}}^2(f^\star)''\left(\frac{-24}{\lambda(1-\gamma)^2 \rho_{\min}}\right)^2 \ .$$

Plugging in the expression of $\lambda$ from (76) yields the following simplified expression

$$\underline{\mu}_f \geq \frac{\epsilon \mathrm{c}_f(1-\gamma)^2 \rho_{\min}^2 \zeta_f^2}{4\omega_f^2}\pi_{\mathrm{ref}}^2(f^\star)''\left(\frac{-96}{\epsilon \mathrm{c}_f(1-\gamma)^3 \rho_{\min}}\right)^2 = \frac{4\epsilon(1-\gamma)^2 \rho_{\min}^2 \pi_{\mathrm{ref}}^2}{C_f^{(1)}}d(\epsilon) \ , \tag{81}$$

where $d(\epsilon)$ and $C_f^{(1)}$ are defined respectively in (77) and (78). Additionally, combining Lemma E.3, Lemma E.4 and Theorem B.12 and the expression of $\lambda$ yields

$$\sigma_f^2 \leq \frac{24}{(1-\gamma)^4}\omega_f^3 \ , \quad \beta_f \leq \frac{6\gamma^H H}{(1-\gamma)^2}\omega_f \ , \quad L_f = \frac{13\omega_f\left(\omega_f + (1-\gamma)\kappa_f\right)}{(1-\gamma)^3} \ , \tag{82}$$

where we used that under $\mathbf{A}_f(\pi_{\mathrm{ref}})$, we have $\omega_f \geq 1$. Using Theorem E.9 and the fact that $\lambda \leq (1-\gamma)\epsilon/4\mathrm{d}_f$, we have that

$$v_\star(\rho) - \mathbb{E}\left[v_{\theta_t}(\rho)\right] \leq \underbrace{(1 - \underline{\mu}_f \eta/4)^t(v_\star^f(\rho) - v_{\theta_0}^f(\rho))}_{(\mathbf{U}')} + \underbrace{\frac{6\eta\sigma_f^2}{B\underline{\mu}_f}}_{(\mathbf{V}')} + \underbrace{\frac{6\beta_f^2}{\underline{\mu}_f}}_{(\mathbf{W}')} + \frac{\epsilon}{2} \ .$$

Next, we aim to show that under our conditions on $T$, $H$, and $\eta$, each of these terms is smaller than $\epsilon/6$.

**Bounding V'.** We start with the term **V'** which gives a condition on the step-size. Using (81) and (82), it holds that

$$\mathbf{V'} \le \frac{144\eta\omega_f^3}{(1-\gamma)^4 B\underline{\mu}_f} \le \frac{576\eta\omega_f^5}{(1-\gamma)^6 B} \frac{1}{\epsilon c_f \rho_{\min}^2 \zeta_f^2 \underline{\pi_{\mathrm{ref}}}^2 (f^\star)'' \left(\frac{-96}{\epsilon c_f (1-\gamma)^3 \rho_{\min}}\right)^2}$$

$$= \frac{C_f^{(3)}\eta}{6\epsilon d(\epsilon)(1-\gamma)^6 B\rho_{\min}^2 \underline{\pi_{\mathrm{ref}}}^2},$$

where $C_f^{(3)}$ is defined in (78). In particular, setting

$$\eta \le \frac{\epsilon^2 d(\epsilon)(1-\gamma)^6 B\rho_{\min}^2 \underline{\pi_{\mathrm{ref}}}^2}{C_f^{(3)}},$$

guarantees that $\mathbf{V''} \le \epsilon/6$, which together with the condition $\eta \le 1/(2L_f)$, gives the condition (79).

**Bounding U'.** In order to ensure that **U'** is smaller then $\epsilon/6$, we need $T$ to satisfy

$$T \ge \frac{1}{\log(1-\eta\underline{\mu}_f/4)} \log\left(\frac{\epsilon}{6\left(v_\star^f(\rho) - v_{\theta_0}^f(\rho)\right)}\right),$$

which combined with the inequality $\log(1+x) \le x$ for $x > -1$, ensures that $\mathbf{V''} < \epsilon/6$ and the condition (80) follows from (79).

**Bounding W'.** Using (82), it holds that

$$\mathbf{W'} \le \frac{6}{\underline{\mu}_f} \cdot \frac{36\gamma^{2H}(H+1)^2}{(1-\gamma)^4}\omega_f^2 \ . \ ,$$

Next, using that for any $a > 0$, we have $a\gamma^{2H}(H+1)^2 \le \epsilon/6$ for

$$H \ge \frac{1}{1-\gamma}\max\left(\frac{4}{1-\gamma}, \log\left(\frac{6a}{\epsilon}\right)\right),$$

shows that under our condition on $H$, we have $\mathbf{W'} < \epsilon/6$. $\qquad\square$

## F  APPLICATION TO COMMON $f$-DIVERGENCES

In this section, we apply the results of the preceding section to two commonly used $f$-divergences, which are Kullback-Leibler and $\alpha$-Tsallis.

### F.1  KULLBACK-LEIBLER

**Lemma F.1.** *Assume that, for some $\underline{\pi_{\mathrm{ref}}} > 0$, $\pi_{\mathrm{ref}}$ satisfy $\mathbf{P}(\underline{\pi_{\mathrm{ref}}})$. The function $f$ defined by $f(u) = u\log(u)$ satisfies $\mathbf{A}_f(\underline{\pi_{\mathrm{ref}}})$, with*

$$\omega_f = 1 \ , \quad \kappa_f = 1 \ , \quad \iota_f = 1 \ .$$

*Additionally, under the condition that*

$$\lambda \le \frac{4}{(1-\gamma)^2 \rho_{\min}(\log(2/\underline{\pi_{\mathrm{ref}}}) + 1)} \ , \tag{83}$$

*we have*

$$\zeta_f = 1, \ \mathrm{d}_f \le |\log(\underline{\pi_{\mathrm{ref}}})|, \ \mathrm{y}_f \le 1 + 2|\log(\underline{\pi_{\mathrm{ref}}})|, \ L_f = \frac{8}{(1-\gamma)^3} + 4\lambda\frac{3 + 4|\log(\underline{\pi_{\mathrm{ref}}})|}{(1-\gamma)^3} \ ,$$

$$\beta_f = \frac{2\gamma^H(H+1)}{(1-\gamma)^2}\left[2 + \lambda + 4\lambda|\log(\underline{\pi_{\mathrm{ref}}})|\right] \ , \quad \sigma_f^2 \le \frac{12}{(1-\gamma)^4}\left[1 + \lambda^2\left(2 + 5|\log(\underline{\pi_{\mathrm{ref}}})|^2\right)\right] \ ,$$

$$\underline{\mu}_f \ge \lambda(1-\gamma)\rho_{\min}^2 \underline{\pi_{\mathrm{ref}}}^2 \exp\left(-\frac{32 + 16\gamma\lambda|\log(\underline{\pi_{\mathrm{ref}}})|}{\lambda(1-\gamma)^2 \rho_{\min}}\right)/9 \ .$$

*Proof.* Firstly, note that we have

$$f(u) = u\log(u) \ , \quad f'(u) = \log(u) + 1 \ , \quad f''(u) = u^{-1} \ , \quad f'''(u) = -u^{-2} \ .$$

**Satisfying $\mathbf{A}_f(\underline{\pi_{\text{ref}}})$.** Observe that $(i)$ and $(ii)$ are immediately valid from the expression above of the derivatives of $f$. Moreover, we have

$$1/uf''(u) = 1 \ , \quad \frac{|f'''(u)|}{f''(u)^2} = 1$$

showing that $(iii)$ of $\mathbf{A}_f(\underline{\pi_{\text{ref}}})$, is satisfied with $\omega_f = \kappa_f = 1$. Finally, as $f''$ is a strictly decreasing function on $\mathbb{R}_+$ then $(iv)$ is valid with $\iota_f = 1$.

**Bounding the constants.** Next, we bound sequentially each of the constants that appear in the statement of the lemma. For any $s \in \mathcal{S}$ and $\nu \in \mathcal{P}(\mathcal{A})$, we have that

$$\sum_{a\in\mathcal{A}} \frac{\pi_{\text{ref}}(a|s)}{f''(\frac{\nu(a)}{\pi_{\text{ref}}(a|s)})} = \sum_{a\in\mathcal{A}} \nu(a) = 1 \ ,$$

Thus using (7), we have that $\zeta_f = 1$. It holds that

$$\begin{aligned}
\mathrm{d}_f &= \max_{(s,\nu)\in\mathcal{S}\times\mathcal{P}(\mathcal{A})} \sum_{a\in\mathcal{A}} \pi_{\text{ref}}(a|s) f\Big(\frac{\nu(a)}{\pi_{\text{ref}}(a|s)}\Big) \\
&= \max_{(s,\nu)\in\mathcal{S}\times\mathcal{P}(\mathcal{A})} \sum_{a\in\mathcal{A}} \nu(a)\log\Big(\frac{\nu(a)}{\pi_{\text{ref}}(a|s)}\Big) \\
&\leq \max_{(s,\nu)\in\mathcal{S}\times\mathcal{P}(\mathcal{A})} \sum_{a\in\mathcal{A}} \nu(a)\log(\nu(a)) - \nu(a)\log(\underline{\pi_{\text{ref}}}) \ ,
\end{aligned}$$

which gives $\mathrm{d}_f \leq -\log(\underline{\pi_{\text{ref}}})$. Next, we have

$$\begin{aligned}
\mathrm{y}_f &= \max_{(s,\nu)\in\mathcal{S}\times\mathcal{P}(\mathcal{A})} \sum_{a\in\mathcal{A}} \frac{\pi_{\text{ref}}(a|s)}{f''\left(\frac{\nu(a)}{\pi_{\text{ref}}(a|s)}\right)} \left| f'\left(\frac{\nu(a)}{\pi_{\text{ref}}(a|s)}\right) \right| \\
&= \max_{(s,\nu)\in\mathcal{P}(\mathcal{A})} \sum_{a\in\mathcal{A}} \nu(a) \left| \log\left(\frac{\nu(a)}{\pi_{\text{ref}}(a|s)}\right) + 1 \right| \\
&= 1 - \log(\underline{\pi_{\text{ref}}}) + \max_{\nu\in\mathcal{P}(\mathcal{A})} -\sum_{a\in\mathcal{A}} \nu(a)\log(\nu(a)) \\
&\leq 1 + 2|\log(\underline{\pi_{\text{ref}}})| \ ,
\end{aligned}$$

where in the last inequality, we used that the entropy of a distribution on $\mathcal{A}$ is bounded by $\log(|\mathcal{A}|)$ and the fact that $\underline{\pi_{\text{ref}}} \leq 1/|\mathcal{A}|$. Next, using Theorem B.12 and the bounds above, we have

$$\begin{aligned}
L_f &= \frac{8\omega_f\left(\gamma\omega_f+(1-\gamma)\kappa_f\right)}{(1-\gamma)^3} + 4\lambda \frac{2\gamma^2\omega_f^2\mathrm{d}_f + 2\gamma(1-\gamma)\omega_f\left[\kappa_f\mathrm{d}_f + \mathrm{y}_f\right] + (1-\gamma)^2\left[\omega_f + 2\kappa_f\mathrm{y}_f\right]}{(1-\gamma)^3} \\
&\leq \frac{8}{(1-\gamma)^3} + 4\lambda \frac{3 + 4|\log(\underline{\pi_{\text{ref}}})|}{(1-\gamma)^3} \ .
\end{aligned}$$

Using Lemma E.3 gives

$$\beta_f = \frac{2\gamma^H(H+1)}{(1-\gamma)^2}\omega_f\left[2 + 2\lambda\mathrm{d}_f + \lambda(1-\gamma)\mathrm{y}_f\right] \leq \frac{2\gamma^H(H+1)}{(1-\gamma)^2}\left[2 + \lambda + 4\lambda|\log(\underline{\pi_{\text{ref}}})|\right] \ .$$

Using Lemma E.4 gives

$$\sigma_f^2 = \frac{12}{(1-\gamma)^4}\left[\omega_f^3 + \lambda^2\gamma^2\omega_f^3\mathrm{d}_f^2 + \lambda^2(1-\gamma)^2\omega_f^2\mathrm{y}_f^2\right] \leq \frac{12}{(1-\gamma)^4}\left[1 + \lambda^2(2 + 5|\log(\underline{\pi_{\text{ref}}})|^2))\right] \ .$$

Next, note that (83), guarantees that we have

$$\lambda \leq \frac{4}{(1-\gamma)^2\rho_{\min}} \min\left(\frac{4}{|f'(\iota_f)|}, \frac{1}{|f'(\frac{1}{2})|}, \frac{4}{|f'(\frac{1}{2}\pi_{\text{ref}})|}\right)$$

Thus, using Lemma E.5, we have

$$\underline{\mu}_f = \frac{\lambda(1-\gamma)\rho_{\min}^2\zeta_f^2}{\omega_f^2}\underline{\pi_{\mathrm{ref}}}^2(f^\star)''\left(-\frac{16+8\gamma\lambda\mathrm{d}_f}{\lambda(1-\gamma)^2\rho_{\min}}\right)^2$$

$$\geq \lambda(1-\gamma)\rho_{\min}^2\underline{\pi_{\mathrm{ref}}}^2\exp\left(-\frac{32+16\gamma\lambda|\log(\underline{\pi_{\mathrm{ref}}})|}{\lambda(1-\gamma)^2\rho_{\min}}\right)/9 \ ,$$

where in the last equality, we used that the convex conjugate of $f(u) = u\log(u)$ is $f^\star(y) = \exp(y-1)$ and that $\exp(-2) \geq 1/9$. $\qquad\square$

In the next two corollaries, we apply Corollary E.7 and Corollary E.10 to get more explicitly the sample complexity of $f\text{-PG}$ with entropy regularization.

**Corollary F.2.** *Assume that, for some $1/4 \geq \underline{\pi_{\mathrm{ref}}} > 0$, $\pi_{\mathrm{ref}}$ satisfy $\mathbf{P}(\pi_{\mathrm{ref}})$. Assume in addition that the initial distribution $\rho$ satisfies $\mathbf{A}_\rho$. Fix any $(1-\gamma)^{-1} \geq \epsilon > 0$, $\lambda > 0$ and $B$ such that*

$$\lambda \leq \min\left(\frac{4}{(1-\gamma)^2\rho_{\min}(\log(2/\underline{\pi_{\mathrm{ref}}})+1)}, 1\right), \ \ B \leq \frac{216|\log(\underline{\pi_{\mathrm{ref}}})|\exp\left(\frac{48|\log(\underline{\pi_{\mathrm{ref}}})|}{\lambda(1-\gamma)^2\rho_{\min}}\right)}{\epsilon\lambda(1-\gamma)^2\underline{\pi_{\mathrm{ref}}}^2\rho_{\min}^2}.$$

*Setting*

$$H \geq \frac{1}{1-\gamma}\log\left(\frac{29160\log(\underline{\pi_{\mathrm{ref}}})^2}{\epsilon\lambda(1-\gamma)^5\rho_{\min}^2\underline{\pi_{\mathrm{ref}}}^2}\right) + \frac{52|\log(\underline{\pi_{\mathrm{ref}}})|}{\lambda(1-\gamma)^3\rho_{\min}}, \tag{84}$$

*and*

$$\eta \leq \frac{\epsilon B\lambda(1-\gamma)^5\rho_{\min}^2\underline{\pi_{\mathrm{ref}}}^2}{15552|\log(\underline{\pi_{\mathrm{ref}}})|^2}\exp\left(-\frac{48|\log(\underline{\pi_{\mathrm{ref}}})|}{\lambda(1-\gamma)^2\rho_{\min}}\right) \ ,$$

*and*

$$T \geq \frac{559872|\log(\underline{\pi_{\mathrm{ref}}})|^2}{\lambda^2\epsilon B(1-\gamma)^6\rho_{\min}^4\underline{\pi_{\mathrm{ref}}}^4}\exp\left(\frac{96|\log(\underline{\pi_{\mathrm{ref}}})|}{\lambda(1-\gamma)^2\rho_{\min}}\right) \ ,$$

*guarantees that*

$$v_\star^f(\rho) - \mathbb{E}\left[v_{\theta_t}^f(\rho)\right] \leq \epsilon \ ,$$

*where $f$ is the Kullback-Leibler divergence generator. Thus, the sample complexity of $f\text{-PG}$ to learn an $\epsilon$-solution of the entropy regularized problem is*

$$TBH \approx \frac{1}{\lambda^3\epsilon B}\frac{|\log(\underline{\pi_{\mathrm{ref}}})|^3}{(1-\gamma)^9\rho_{\min}^5\underline{\pi_{\mathrm{ref}}}^4}\exp\left(\frac{|\log(\underline{\pi_{\mathrm{ref}}})|}{\lambda(1-\gamma)^2\rho_{\min}}\right)$$

*Proof.* To prove this corollary, we show that the assumptions of Corollary E.7 hold. First, $\mathbf{A}_f(\underline{\pi_{\mathrm{ref}}})$ holds as a consequence of Lemma F.1. Then, we show that the condition (73) in Corollary E.7 holds, that is $H \leq \frac{4}{(1-\gamma)^2} + \frac{1}{1-\gamma}\log\left(\frac{216\omega_f^2}{\epsilon\underline{\mu}_f(1-\gamma)^4}\left[4 + 4\lambda^2\mathrm{d}_f^2 + \lambda^2(1-\gamma)^2\mathrm{y}_f^2\right]\right)$. To this end, we remark that

$$\frac{4}{(1-\gamma)^2} + \frac{1}{1-\gamma}\log\left(\frac{216\omega_f^2}{\epsilon\underline{\mu}_f(1-\gamma)^4}\left[4 + 4\lambda^2\mathrm{d}_f^2 + \lambda^2(1-\gamma)^2\mathrm{y}_f^2\right]\right)$$

$$\leq \frac{4}{(1-\gamma)^2} + \frac{1}{1-\gamma}\log\left(\frac{216}{\epsilon\underline{\mu}_f(1-\gamma)^4}\left[4 + \lambda^2(5 + 6\log(\underline{\pi_{\mathrm{ref}}})^2)\right]\right)$$

$$\leq \frac{4}{(1-\gamma)^2} + \frac{1}{1-\gamma}\log\left(\frac{1944\left[4 + \lambda^2(5 + 6\log(\underline{\pi_{\mathrm{ref}}})^2)\right]}{\epsilon\lambda(1-\gamma)^5\rho_{\min}^2\underline{\pi_{\mathrm{ref}}}^2}\right) + \frac{32+16\gamma\lambda|\log(\underline{\pi_{\mathrm{ref}}})|}{\lambda(1-\gamma)^3\rho_{\min}}$$

$$\leq \frac{1}{1-\gamma}\log\left(\frac{29160\log(\underline{\pi_{\mathrm{ref}}})^2}{\epsilon\lambda(1-\gamma)^5\rho_{\min}^2\underline{\pi_{\mathrm{ref}}}^2}\right) + \frac{52|\log(\underline{\pi_{\mathrm{ref}}})|}{\lambda(1-\gamma)^3\rho_{\min}} \leq H \ ,$$

where we used the lower bound on $\underline{\mu}_f$ provided in Lemma F.1 in the second inequality, as well as $\lambda < 1$ and $\underline{\pi}_{\text{ref}} \leq 1/4$ in the last two inequalities. Furthermore, Lemma F.1 with $\lambda \leq 1$ gives

$$L_f \leq \frac{36|\log(\underline{\pi}_{\text{ref}})|}{(1-\gamma)^3} \quad , \tag{85}$$

$$\sigma_f^2 \leq \frac{96|\log(\underline{\pi}_{\text{ref}})|^2}{(1-\gamma)^4} \quad , \tag{86}$$

$$\underline{\mu}_f \geq \frac{\lambda(1-\gamma)\rho_{\min}^2 \underline{\pi}_{\text{ref}}^2}{9} \exp\left(-\frac{48|\log(\underline{\pi}_{\text{ref}})|}{\lambda(1-\gamma)^2\rho_{\min}}\right) \quad . \tag{87}$$

Using these three bounds on smoothness, variance and Polyak-Łojasiewicz coefficients, we obtain

$$\min\left(\frac{1}{2L_f}, \frac{\epsilon B \underline{\mu}_f}{18\sigma_f^2}\right) \geq \min\left(\frac{(1-\gamma)^3}{72|\log(\underline{\pi}_{\text{ref}})|}, \frac{\epsilon B \lambda (1-\gamma)^5 \rho_{\min}^2 \underline{\pi}_{\text{ref}}^2 \exp\left(-\frac{48|\log(\underline{\pi}_{\text{ref}})|}{\lambda(1-\gamma)^2\rho_{\min}}\right)}{15552|\log(\underline{\pi}_{\text{ref}})|^2}\right)$$

$$= \frac{\epsilon B \lambda (1-\gamma)^5 \rho_{\min}^2 \underline{\pi}_{\text{ref}}^2}{15552|\log(\underline{\pi}_{\text{ref}})|^2} \exp\left(-\frac{48|\log(\underline{\pi}_{\text{ref}})|}{\lambda(1-\gamma)^2\rho_{\min}}\right) \quad , \tag{88}$$

where in the last identity, we used the fact that $\epsilon < (1-\gamma)^{-1}$ and that

$$B \leq \frac{216|\log(\underline{\pi}_{\text{ref}})|}{\epsilon\lambda(1-\gamma)^2\underline{\pi}_{\text{ref}}^2\rho_{\min}^2} \exp\left(\frac{48|\log(\underline{\pi}_{\text{ref}})|}{\lambda(1-\gamma)^2\rho_{\min}}\right) \quad .$$

This shows that our condition on $\eta$ guarantees that the one set in Corollary E.7 is satisfied. Finally using (88) and (87), we have

$$\frac{4}{\underline{\mu}_f}\max\left(2L_f, \frac{18\sigma_f^2}{\epsilon B \underline{\mu}_f}\right) \leq \frac{4}{\underline{\mu}_f}\frac{15552|\log(\underline{\pi}_{\text{ref}})|^2}{\epsilon B \lambda (1-\gamma)^5 \rho_{\min}^2 \underline{\pi}_{\text{ref}}^2} \exp\left(\frac{48|\log(\underline{\pi}_{\text{ref}})|}{\lambda(1-\gamma)^2\rho_{\min}}\right)$$

$$\leq \frac{559872|\log(\underline{\pi}_{\text{ref}})|^2}{\lambda^2\epsilon B(1-\gamma)^6\rho_{\min}^4\underline{\pi}_{\text{ref}}^4} \exp\left(\frac{96|\log(\underline{\pi}_{\text{ref}})|}{\lambda(1-\gamma)^2\rho_{\min}}\right) \quad ,$$

which concludes the proof. $\qquad\square$

**Corollary F.3.** *Assume that, for some $\underline{\pi}_{\text{ref}} \in (0, 1/4]$, $\pi_{\text{ref}}$ satisfy $P(\underline{\pi}_{\text{ref}})$. Assume in addition that the initial distribution $\rho$ satisfies $A_\rho$ and fix $f$ to be the Kullback-Leibler divergence generator, i.e. $f(u) = u\log(u)$. Fix any $\epsilon \in (0, (1-\gamma)^{-1}]$, such that*

$$\epsilon < \frac{16}{(1-\gamma)^3\rho_{\min}(\log(2/\underline{\pi}_{\text{ref}}) + 1)} \quad , \quad \text{and set } \lambda = \frac{(1-\gamma)\epsilon}{12\log(|\underline{\pi}_{\text{ref}}|)} \quad . \tag{89}$$

*Additionally set any $B$ such that*

$$B \leq \frac{1}{\epsilon^2(1-\gamma)^3\rho_{\min}^2\underline{\pi}_{\text{ref}}^2} \exp\left(\frac{576|\log(\underline{\pi}_{\text{ref}})|}{\epsilon(1-\gamma)^3\rho_{\min}}\right) \quad . \tag{90}$$

*Setting*

$$H \geq \frac{1}{1-\gamma}\log\left(\frac{|\log(\underline{\pi}_{\text{ref}})|}{\epsilon(1-\gamma)^6\rho_{\min}^2\underline{\pi}_{\text{ref}}^2}\right) + \frac{586|\log(\underline{\pi}_{\text{ref}})|}{\epsilon(1-\gamma)^4\rho_{\min}} \quad ,$$

*and*

$$\eta \leq \frac{\epsilon^2(1-\gamma)^6 B\rho_{\min}^2\underline{\pi}_{\text{ref}}^2}{93312|\log(\underline{\pi}_{\text{ref}})|} \exp\left(\frac{-576|\log(\underline{\pi}_{\text{ref}})|}{\epsilon(1-\gamma)^3\rho_{\min}}\right) \quad , \tag{91}$$

*and*

$$T \geq \frac{644972544|\log(\underline{\pi}_{\text{ref}})|^2}{\epsilon^3(1-\gamma)^8\rho_{\min}^4\underline{\pi}_{\text{ref}}^4 B} \exp\left(\frac{1152|\log(\underline{\pi}_{\text{ref}})|}{\epsilon(1-\gamma)^3\rho_{\min}}\right)\log\left(\frac{6(v_\star^f(\rho) - v_{\theta_0}^f(\rho))}{\epsilon}\right) \quad , \tag{92}$$

*guarantees that*

$$v_\star(\rho) - \mathbb{E}\left[v_{\theta_t}(\rho)\right] \leq \epsilon \ .$$

*Thus, the sample complexity of $f$-PG, where $f$ is the Kullback-Leibler divergence generator, to learn an $\epsilon$-solution of the non-regularized problem is*

$$TBH \approx \frac{|\log(\underline{\pi_{\mathrm{ref}}})|^3}{\epsilon^4(1-\gamma)^{12}\rho_{\min}^5 \underline{\pi_{\mathrm{ref}}}^4} \exp\left(\frac{|\log(\underline{\pi_{\mathrm{ref}}})|}{\epsilon(1-\gamma)^3\rho_{\min}}\right)$$

*Proof.* This result follows from Corollary E.10, whose assumptions we check now. First, note that (89) implies that

$$\epsilon < \frac{16}{(1-\gamma)^3\rho_{\min}} \min\left(\frac{4}{|f'(\iota_f)|}, \frac{1}{|f'(\frac{1}{2})|}, \frac{4}{|f'(\frac{1}{2}\underline{\pi_{\mathrm{ref}}})|}\right) \ .$$

Additionally, we can rewrite the constants from Corollary E.10 using Lemma F.1, which gives

$$C_f^{(1)} \leq 48|\log(\underline{\pi_{\mathrm{ref}}})|, \qquad C_f^{(2)} \leq 48, \qquad C_f^{(3)} = 10368|\log(\underline{\pi_{\mathrm{ref}}})|,$$

$$d(\epsilon) \geq \exp\left(\frac{-576|\log(\underline{\pi_{\mathrm{ref}}})|}{\epsilon(1-\gamma)^3\rho_{\min}}\right)/9. \tag{93}$$

where $C_f^{(1)}, C_f^{(2)}, C_f^{(3)}$, and $d(\epsilon)$ are defined in (78) and (77). Next, the condition on $H$ in Corollary E.10 holds since

$$\frac{4}{(1-\gamma)^2} + \frac{1}{1-\gamma}\log\left(\frac{297\omega_f^2 C_1^f}{\epsilon d(\epsilon)(1-\gamma)^6\rho_{\min}^2\underline{\pi_{\mathrm{ref}}}^2}\right)$$

$$\leq \frac{4}{(1-\gamma)^2} + \frac{1}{1-\gamma}\log\left(\frac{128304|\log(\underline{\pi_{\mathrm{ref}}})|}{\epsilon(1-\gamma)^6\rho_{\min}^2\underline{\pi_{\mathrm{ref}}}^2}\right) + \frac{576|\log(\underline{\pi_{\mathrm{ref}}})|}{\epsilon(1-\gamma)^4\rho_{\min}}$$

$$\leq \frac{1}{1-\gamma}\log\left(\frac{|\log(\underline{\pi_{\mathrm{ref}}})|}{\epsilon(1-\gamma)^6\rho_{\min}^2\underline{\pi_{\mathrm{ref}}}^2}\right) + \frac{586|\log(\underline{\pi_{\mathrm{ref}}})|}{\epsilon(1-\gamma)^4\rho_{\min}} \leq H \ ,$$

where in the second to last inequality, we used that $\underline{\pi_{\mathrm{ref}}} < 1/4$ and that $\log(128304) \leq 6$. Using (93), we have

$$\min\left(\frac{(1-\gamma)^3}{C_f^{(2)}}, \frac{\epsilon^2 d(\epsilon)(1-\gamma)^6 B\rho_{\min}^2\underline{\pi_{\mathrm{ref}}}^2}{C_f^{(3)}}\right)$$

$$\geq \min\left(\frac{(1-\gamma)^3}{48}, \frac{\epsilon^2(1-\gamma)^6 B\rho_{\min}^2\underline{\pi_{\mathrm{ref}}}^2}{93312|\log(\underline{\pi_{\mathrm{ref}}})|}\exp\left(\frac{-576|\log(\underline{\pi_{\mathrm{ref}}})|}{\epsilon(1-\gamma)^3\rho_{\min}}\right)\right)$$

$$\geq \frac{\epsilon^2(1-\gamma)^6 B\rho_{\min}^2\underline{\pi_{\mathrm{ref}}}^2}{93312|\log(\underline{\pi_{\mathrm{ref}}})|}\exp\left(\frac{-576|\log(\underline{\pi_{\mathrm{ref}}})|}{\epsilon(1-\gamma)^3\rho_{\min}}\right) \ , \tag{94}$$

where in the last inequality, we used the condition on $B$ introduced in (90). Hence our condition on the step-size ensures that the one assumed in Corollary E.10 is satisfied. Next, using (93) and (94) yields

$$\frac{16 C_f^{(1)}\ell(\epsilon)}{\epsilon d(\epsilon)(1-\gamma)^2\rho_{\min}^2\underline{\pi_{\mathrm{ref}}}^2}\max\left(\frac{C_f^{(2)}}{(1-\gamma)^3}, \frac{C_f^{(3)}}{\epsilon^2 d(\epsilon)(1-\gamma)^6 B\rho_{\min}^2\underline{\pi_{\mathrm{ref}}}^2}\right)$$

$$\leq \frac{16 C_f^{(1)}\ell(\epsilon)}{\epsilon d(\epsilon)(1-\gamma)^2\rho_{\min}^2\underline{\pi_{\mathrm{ref}}}^2} \cdot \frac{93312|\log(\underline{\pi_{\mathrm{ref}}})|}{\epsilon^2(1-\gamma)^6 B\rho_{\min}^2\underline{\pi_{\mathrm{ref}}}^2}\exp\left(\frac{576|\log(\underline{\pi_{\mathrm{ref}}})|}{\epsilon(1-\gamma)^3\rho_{\min}}\right)$$

$$\leq \frac{6912|\log(\underline{\pi_{\mathrm{ref}}})|\ell(\epsilon)}{\epsilon(1-\gamma)^2\rho_{\min}^2\underline{\pi_{\mathrm{ref}}}^2} \cdot \frac{93312|\log(\underline{\pi_{\mathrm{ref}}})|}{\epsilon^2(1-\gamma)^6 B\rho_{\min}^2\underline{\pi_{\mathrm{ref}}}^2}\exp\left(\frac{1152|\log(\underline{\pi_{\mathrm{ref}}})|}{\epsilon(1-\gamma)^3\rho_{\min}}\right)$$

$$\leq \frac{644972544|\log(\underline{\pi_{\mathrm{ref}}})|^2}{\epsilon^3(1-\gamma)^8\rho_{\min}^4\underline{\pi_{\mathrm{ref}}}^4 B}\exp\left(\frac{1152|\log(\underline{\pi_{\mathrm{ref}}})|}{\epsilon(1-\gamma)^3\rho_{\min}}\right)\log\left(\frac{6(v_\star^f(\rho) - v_{\theta_0}^f(\rho))}{\epsilon}\right) \ ,$$

which proves that under our condition on $T$ the one assumed by Corollary E.10 is satisfied. $\qquad\square$

## F.2 $\alpha$-TSALLIS

**Lemma F.4.** *Assume that, for some $\underline{\pi}_{\text{ref}} \in (0, 1/4]$, $\pi_{\text{ref}}$ satisfy $\boldsymbol{P}(\underline{\pi}_{\text{ref}})$. For any $\alpha \in (0;1)$, the function $f_\alpha$ defined by*

$$f_\alpha(u) = \frac{u^\alpha - \alpha u + \alpha - 1}{\alpha(\alpha - 1)} \; ,$$

*satisfies $\boldsymbol{A}_f(\underline{\pi}_{\text{ref}})$, with*

$$\omega_f = \underline{\pi}_{\text{ref}}^{\,\alpha-1} \; , \quad \kappa_f = 2\underline{\pi}_{\text{ref}}^{\,\alpha-1} \; , \quad \iota_f = 1 \; .$$

*Additionally, under the condition that*

$$\lambda \le \frac{4}{(1-\gamma)^2 \rho_{\min}} \cdot \frac{1-\alpha}{(\underline{\pi}_{\text{ref}}/2)^{\alpha-1} - 1} \; , \tag{95}$$

*we have*

$$\zeta_f = 1, \;\; \mathrm{d}_f \le \frac{4|\log(\underline{\pi}_{\text{ref}})|}{\alpha^2}, \;\; \mathrm{y}_f \le 4|\log(\underline{\pi}_{\text{ref}})|, \;\; L_f = \frac{16\underline{\pi}_{\text{ref}}^{\,\alpha-1}}{(1-\gamma)^3} + 180\lambda \frac{\underline{\pi}_{\text{ref}}^{\,2\alpha-2}|\log(\underline{\pi}_{\text{ref}})|}{\alpha^2(1-\gamma)^3} \; ,$$

$$\beta_f = \frac{4\gamma^H(H+1)}{(1-\gamma)^2}\left[1 + 6\lambda \frac{|\log(\underline{\pi}_{\text{ref}})|}{\alpha^2}\right] \; , \quad \sigma_f^2 \le \frac{12\underline{\pi}_{\text{ref}}^{\,3\alpha-3}}{(1-\gamma)^4}\left[1 + \frac{16\lambda^2}{\alpha^4}\log(\underline{\pi}_{\text{ref}}|^2)\right] \; ,$$

$$\underline{\mu}_f \ge \lambda(1-\gamma)\underline{\pi}_{\text{ref}}^{\,2-2\alpha}\rho_{\min}^2 \underline{\pi}_{\text{ref}}^{\,2} \exp_\alpha\left(-\frac{16 + 32\gamma\lambda|\log(\underline{\pi}_{\text{ref}})|/\alpha^2}{\lambda(1-\gamma)^2\rho_{\min}}\right)^{4-2\alpha} \; .$$

*Proof.* Fix any $\alpha \in (0,1)$ and set $f = f_\alpha$. Firstly, note that we have

$$f(u) = \frac{u^\alpha - \alpha u + \alpha - 1}{\alpha(\alpha-1)}, \;\; f'(u) = \frac{u^{\alpha-1} - 1}{\alpha - 1}, \;\; f''(u) = u^{\alpha-2} \; , \;\; f'''(u) = (\alpha-2)u^{\alpha-3} \; .$$

**Satisfying $\boldsymbol{A}_f(\underline{\pi}_{\text{ref}})$.** Observe that $(i)$ and $(ii)$ are immediately valid from the expression above of the derivatives of $f$. Moreover, we have

$$1/(uf_\alpha''(u)) = u^{1-\alpha} \; , \quad \frac{|f'''(u)|}{f''(u)^2} = |\alpha - 2|u^{1-\alpha} \; ,$$

showing that $(iii)$ of $\boldsymbol{A}_f(\underline{\pi}_{\text{ref}})$, is satisfied with $\omega_f = \underline{\pi}_{\text{ref}}^{\,\alpha-1}$, and $\kappa_f = 2\underline{\pi}_{\text{ref}}^{\,\alpha-1}$. Finally, as $f''$ is a strictly decreasing function on $\mathbb{R}_+$ then $(iv)$ is valid with $\iota_f = 1$. Next, we bound sequentially each of the constants that appear in the statement of the lemma.

**Bounding $\zeta_f$.** For any $s \in \mathcal{S}$ and $\nu \in \mathcal{P}(\mathcal{A})$, using Jensen's inequality we have that

$$\sum_{a \in \mathcal{A}} \frac{\pi_{\text{ref}}(a|s)}{f''\left(\frac{\nu(a)}{\pi_{\text{ref}}(a|s)}\right)} = \sum_{a \in \mathcal{A}} \pi_{\text{ref}}(a|s)\left(\frac{\nu(a)}{\pi_{\text{ref}}(a|s)}\right)^{2-\alpha} \ge \left(\sum_{a \in \mathcal{A}} \pi_{\text{ref}}(a|s)\frac{\nu(a)}{\pi_{\text{ref}}(a|s)}\right)^{2-\alpha} = 1 \; ,$$

Thus using (7), we have that $\zeta_f = 1$.

**Bounding $\mathrm{d}_f$.** For any state $s \in \mathcal{S}$ and $\nu \in \mathcal{P}(\mathcal{A})$, it holds that

$$\mathrm{D}^f(\nu\|\pi_{\text{ref}}(\cdot|s)) = \frac{1}{\alpha(\alpha-1)}\sum_{a \in \mathcal{A}} \pi_{\text{ref}}(a|s)\left[\left(\frac{\nu(a)}{\pi_{\text{ref}}(a|s)}\right)^\alpha - \alpha\frac{\nu(a)}{\pi_{\text{ref}}(a|s)} - (1-\alpha)\right] \; .$$

Next, for $y \in ]0; \frac{1}{\underline{\pi}_{\text{ref}}}]$, define the function $p_y \colon [\alpha; 1] \to \mathbb{R}$ which satisfies

$$p_y(\beta) = y^\beta - \beta y - (1-\beta) \; , \quad p_y'(\beta) = \log(y)y^\beta - y + 1 \; .$$

It holds that

$$\frac{y^\alpha - \alpha y - (1-\alpha)}{\alpha - 1} = \frac{p_y(\alpha) - p_y(1)}{\alpha - 1} \le \sup_{\beta \in [\alpha,1]}|p_y'(\beta)| \le y+1+y|\log(y)|\mathbf{1}_{y\ge 1} + |\log(y)|y^\alpha\mathbf{1}_{y\le 1}.$$

Applying the previous inequality with $y = \nu(a)/\pi_{\text{ref}}(a|s)$ yields

$$D^f(\nu\|\pi_{\text{ref}}(\cdot|s)) \leq \frac{1}{\alpha}\sum_{a\in\mathcal{A}}\pi_{\text{ref}}(a|s)\left[1 + \frac{\nu(a)}{\pi_{\text{ref}}(a|s)} + \frac{\nu(a)}{\pi_{\text{ref}}(a|s)}|\log(\frac{\nu(a)}{\pi_{\text{ref}}(a|s)})|1_{\nu(a)/\pi_{\text{ref}}(a|s)\geq 1}\right.$$

$$\left. + |\log(\frac{\nu(a)}{\pi_{\text{ref}}(a|s)})|\left(\frac{\nu(a)}{\pi_{\text{ref}}(a|s)}\right)^\alpha 1_{\nu(a)/\pi_{\text{ref}}(a|s)\leq 1}\right]$$

$$= \frac{1}{\alpha}\sum_{a\in\mathcal{A}}\nu(a)\log(\nu(a))1_{\nu(a)/\pi_{\text{ref}}(a|s)\geq 1} - \frac{1}{\alpha}\sum_{a\in\mathcal{A}}\nu(a)\log(\pi_{\text{ref}}(a|s))1_{\nu(a)/\pi_{\text{ref}}(a|s)\geq 1}$$

$$+ \frac{1}{\alpha}\max_{a\in\mathcal{A}}|\log(\frac{\nu(a)}{\pi_{\text{ref}}(a|s)})|\left(\frac{\nu(a)}{\pi_{\text{ref}}(a|s)}\right)^\alpha 1_{\nu(a)/\pi_{\text{ref}}(a|s)\leq 1} + \frac{2}{\alpha} \ ,$$

where in the last equality, we used that if $x \geq 1$, then $\log(x) \geq 0$. Next using that $\log(\nu(a)) \leq 0$ yields

$$D^f(\nu\|\pi_{\text{ref}}(\cdot|s)) \leq -\frac{1}{\alpha}\sum_{a\in\mathcal{A}}\nu(a)\log(\pi_{\text{ref}}(a|s)) + \frac{1}{\alpha}\max_{x\in[0;1]}|\log(x)|x^\alpha + \frac{2}{\alpha} \ ,$$

$$\leq \frac{1}{\alpha}|\log(\underline{\pi_{\text{ref}}})| + \frac{1}{\alpha}\max_{x\in[0;1]}|\log(x)|x^\alpha + \frac{2}{\alpha} \ ,$$

where in the last inequality, we used that the entropy of a probability measure on $\mathcal{A}$ is bounded by $|\log(\underline{\pi_{\text{ref}}})|$. Next using that $\max_{x\in[0;1]}|\log(x)|x^\alpha \leq e^{-1}/\alpha$ combined with $\max(1, \log(|\mathcal{A}|)) \leq |\log(\underline{\pi_{\text{ref}}})|$ gives

$$D^f(\nu\|\pi_{\text{ref}}(\cdot|s)) \leq \frac{3|\log(\underline{\pi_{\text{ref}}})|}{\alpha} + \frac{1}{\alpha^2} \leq \frac{4|\log(\underline{\pi_{\text{ref}}})|}{\alpha^2} \ .$$

Thus, it holds that

$$d_f \leq \frac{4|\log(\underline{\pi_{\text{ref}}})|}{\alpha^2} \ .$$

**Bounding $y_f$.** For any policy $\nu \in \mathcal{P}(\mathcal{A})$ and $s \in \mathcal{S}$, we have

$$\sum_{a\in\mathcal{A}}\frac{\pi_{\text{ref}}(a|s)}{f''(\frac{\nu(a)}{\pi_{\text{ref}}(a|s)})}\left|f'\left(\frac{\nu(a)}{\pi_{\text{ref}}(a|s)}\right)\right| = \left|\frac{1}{1-\alpha}\right|\left|\sum_{a\in\mathcal{A}}\frac{\nu(a)^{2-\alpha}}{\pi_{\text{ref}}(a|s)^{1-\alpha}}\left|\left(\frac{\nu(a)}{\pi_{\text{ref}}(a|s)}\right)^{\alpha-1} - 1\right| \ .$$

Next, define the function $g_y$ for $y \in ]0; \frac{1}{\underline{\pi_{\text{ref}}}}]$ which satisfies

$$g_y(\beta) = y^{\beta-1} \ , \quad g_y'(\beta) = \log(y)y^{\beta-1} \ .$$

It holds that

$$\left|\frac{y^{\alpha-1}-1}{1-\alpha}\right| = \left|\frac{g_y(\alpha) - g_y(1)}{\alpha - 1}\right| \leq \sup_{\beta\in[\alpha,1]}|g_y'(\beta)| \leq |\log(y)|\,1_{y\leq 1} + \left|\log(y)y^{\alpha-1}\right|1_{y\geq 1} \ ,$$

Hence, applying the previous inequality with $y = \nu(a)/\pi_{\text{ref}}(a|s)$ gives

$$\sum_{a\in\mathcal{A}}\frac{\pi_{\text{ref}}(a|s)}{f''(\frac{\nu(a)}{\pi_{\text{ref}}(a|s)})}\left|f'\left(\frac{\nu(a)}{\pi_{\text{ref}}(a|s)}\right)\right|$$

$$\leq \sum_{a\in\mathcal{A}}\frac{\nu(a)^{2-\alpha}}{\pi_{\text{ref}}(a|s)^{1-\alpha}}\left[\left|\log\left(\frac{\nu(a)}{\pi_{\text{ref}}(a|s)}\right)\right|1_{\nu(a)\leq\pi_{\text{ref}}(a|s)} + \left|\log\left(\frac{\pi(a|s)}{\pi_{\text{ref}}(a|s)}\right)\right|\left(\frac{\pi_{\text{ref}}(a|s)}{\nu(a)}\right)^{1-\alpha}\right]$$

$$\leq \sum_{a\in\mathcal{A}}\left(\frac{\nu(a)}{\pi_{\text{ref}}(a)}\right)^{1-\alpha}\nu(a)\left|\log\left(\frac{\nu(a)}{\pi_{\text{ref}}(a|s)}\right)\right|1_{\nu(a)\leq\pi_{\text{ref}}(a|s)} + \sum_{a\in\mathcal{A}}\nu(a)\left|\log\left(\frac{\pi(a|s)}{\pi_{\text{ref}}(a|s)}\right)\right|$$

$$\leq \sum_{a\in\mathcal{A}}\nu(a)\left|\log\left(\frac{\nu(a)}{\pi_{\text{ref}}(a|s)}\right)\right| + 2|\log(\underline{\pi_{\text{ref}}})| \ ,$$

where in the last inequality, we used for the first term that for any $u \in \mathbb{R}, u1_{u\leq 1} \leq 1$ that the entropy of a probability distribution on $\mathcal{A}$ is bounded by $\log(|\mathcal{A}|)$, the fact that $\underline{\pi_{\text{ref}}} \leq 1/|\mathcal{A}|$. Using the same argument again to bound the first term gives

$$y_f \leq 4|\log(\underline{\pi_{\text{ref}}})| \ .$$

.

**Bounding $L_f$.** Next, using Theorem B.12 and the bound on the constants previously computed, we have

$$L_f = \frac{8\omega_f\left(\gamma\omega_f + (1-\gamma)\kappa_f\right)}{(1-\gamma)^3} + 4\lambda\frac{2\gamma^2\omega_f^2 d_f + 2\gamma(1-\gamma)\omega_f\left[\kappa_f d_f + y_f\right] + (1-\gamma)^2\left[\omega_f + 2\kappa_f y_f\right]}{(1-\gamma)^3}$$

$$\leq \frac{16\pi_{\mathrm{ref}}^{\alpha-1}}{(1-\gamma)^3} + 180\lambda\frac{\pi_{\mathrm{ref}}^{2\alpha-2}|\log(\pi_{\mathrm{ref}})|}{\alpha^2(1-\gamma)^3} \ ,$$

where in the last inequality, we used that $\underline{\pi}_{\mathrm{ref}} < 1/4$.

**Bounding $\beta_f$.** Using Lemma E.3 gives

$$\beta_f = \frac{2\gamma^H(H+1)}{(1-\gamma)^2}\omega_f\left[2 + 2\lambda d_f + \lambda(1-\gamma)y_f\right] \leq \frac{4\gamma^H(H+1)}{(1-\gamma)^2}\left[1 + 6\lambda\frac{|\log(\underline{\pi}_{\mathrm{ref}})|}{\alpha^2}\right] \ .$$

**Bounding $\sigma_f$.** Using Lemma E.4 gives

$$\sigma_f^2 = \frac{12}{(1-\gamma)^4}\left[\omega_f^3 + \lambda^2\gamma^2\omega_f^3 d_f^2 + \lambda^2(1-\gamma)^2\omega_f^2 y_f^2\right] \leq \frac{12\pi_{\mathrm{ref}}^{3\alpha-3}}{(1-\gamma)^4}\left[1 + \frac{16\lambda^2}{\alpha^4}\log(\underline{\pi}_{\mathrm{ref}}|^2)\right] \ .$$

**Bounding $\underline{\mu}_f$.** Next, note that as $f_\alpha'$ is an increasing function then $f_\alpha'(\underline{\pi}_{\mathrm{ref}}/2) \leq f_\alpha'(1/2) \leq f_\alpha'(\iota_f) = f_\alpha'(1) = 0$. Thus, we have $|f_\alpha'(\iota_f)| \leq |f_\alpha'(1/2)| \leq |f_\alpha'(\underline{\pi}_{\mathrm{ref}}/2)|$. This proves that (95), guarantees that

$$\lambda \leq \frac{4}{(1-\gamma)^2\rho_{\min}}\min\left(\frac{4}{|f'(\iota_f)|}, \frac{1}{|f'(\frac{1}{2})|}, \frac{4}{|f'(\frac{1}{2}\underline{\pi}_{\mathrm{ref}})|}\right)$$

Thus, using Lemma E.5, we have

$$\underline{\mu}_f = \frac{\lambda(1-\gamma)\rho_{\min}^2\zeta_f^2}{\omega_f^2}\underline{\pi}_{\mathrm{ref}}^2(f^\star)''\left(-\frac{16 + 8\gamma\lambda d_f}{\lambda(1-\gamma)^2\rho_{\min}}\right)^2 \ . \tag{96}$$

Next, recall from proposition 8 of Roulet et al. (2025) that

$$f_\alpha^*(x) = \frac{(1 + (\alpha-1)x)^{\frac{\alpha}{\alpha-1}} - 1}{\alpha} \ , \quad \text{for } x \leq \frac{1}{1-\alpha} \ .$$

Thus, we have

$$(f_\alpha^\star)'(x) = \left(1 + (\alpha-1)x\right)^{\frac{1}{\alpha-1}} = \exp_\alpha(x) \ ,$$

where we have originally defined $\exp_\alpha$ in Section 2. Finally, it holds that

$$(f_\alpha^\star)''(x) = \exp_\alpha(x)^{2-\alpha} \ .$$

Thus,

$$\underline{\mu}_f \geq \lambda(1-\gamma)\underline{\pi}_{\mathrm{ref}}^{2-2\alpha}\rho_{\min}^2\underline{\pi}_{\mathrm{ref}}^2\exp_\alpha\left(-\frac{16 + 32\gamma\lambda|\log(\underline{\pi}_{\mathrm{ref}})|/\alpha^2}{\lambda(1-\gamma)^2\rho_{\min}}\right)^{4-2\alpha} \ ,$$

$\square$

In the next two corollaries, we apply Corollary E.7 and Corollary E.10 to get more explicitly the sample complexity of $f$-PG with entropy regularization.

**Corollary F.5.** *Assume that, for some $1/4 \geq \underline{\pi}_{\mathrm{ref}} > 0$, $\pi_{\mathrm{ref}}$ satisfy $\boldsymbol{P}(\underline{\pi}_{\mathrm{ref}})$. Assume in addition that the initial distribution $\rho$ satisfies $\boldsymbol{A}_\rho$. Fix any $(1-\gamma)^{-1} \geq \epsilon > 0$, $\alpha \in (0,1)$, $\lambda > 0$, and $B$ such that*

$$\lambda \leq \min\left(\frac{4}{(1-\gamma)^2\rho_{\min}} \cdot \frac{1-\alpha}{(\pi_{\mathrm{ref}}/2)^{\alpha-1} - 1}, 1\right), \quad B \leq \frac{\exp_\alpha\left(-\frac{48|\log(\underline{\pi}_{\mathrm{ref}})|\max(\lambda,\alpha^2)}{\lambda\alpha^2(1-\gamma)^2\rho_{\min}}\right)^{2\alpha-4}}{\underline{\pi}_{\mathrm{ref}}^2\rho_{\min}^2} \ .$$

*Setting*

$$H \geq \frac{1}{1-\gamma} \log \left( \frac{28152\underline{\pi_{\mathrm{ref}}}^{4\alpha-4} |\log(\underline{\pi_{\mathrm{ref}}})|^2}{\epsilon \lambda \alpha^4 (1-\gamma)^5 \rho_{\min}^2 \underline{\pi_{\mathrm{ref}}}^2} \max(\alpha^2, \lambda)^2 \right) + \frac{196 |\log(\underline{\pi_{\mathrm{ref}}})|}{\lambda \alpha^2 (1-\gamma)^3 \rho_{\min}} \max(\alpha^2, \lambda) \tag{97}$$

*and*

$$\eta \leq \frac{\epsilon B \lambda (1-\gamma)^5 \alpha^4 \underline{\pi_{\mathrm{ref}}}^{5-5\alpha} \rho_{\min}^2 \underline{\pi_{\mathrm{ref}}}^2}{3672 \log(\underline{\pi_{\mathrm{ref}}})^2 \max(\alpha^2, \lambda)^2} \exp_\alpha \left( -\frac{48 |\log(\underline{\pi_{\mathrm{ref}}})|}{\lambda \alpha^2 (1-\gamma)^2 \rho_{\min}} \max(\lambda, \alpha^2) \right)^{4-2\alpha},$$

*and*

$$T \geq \frac{14688 \log(\underline{\pi_{\mathrm{ref}}})^2 \max(\alpha^2, \lambda)^2}{\lambda^2 \epsilon B (1-\gamma)^6 \alpha^4 \underline{\pi_{\mathrm{ref}}}^{7-7\alpha} \rho_{\min}^4 \underline{\pi_{\mathrm{ref}}}^4} \exp_\alpha \left( -\frac{48 |\log(\underline{\pi_{\mathrm{ref}}})|}{\lambda \alpha^2 (1-\gamma)^2 \rho_{\min}} \max(\lambda, \alpha^2) \right)^{4\alpha-8},$$

*guarantees that*

$$v_\star^f(\rho) - \mathbb{E}\left[ v_{\theta_t}^f(\rho) \right] \leq \epsilon ,$$

*where $f = f_\alpha$ is the $\alpha$-Csiszár–Cressie–Read divergence generator. Thus, the sample complexity of $f\text{-PG}$ to learn an $\epsilon$-solution of the $\alpha$-Tsallis regularized problem is*

$$TBH \approx \frac{|\log(\underline{\pi_{\mathrm{ref}}})|^3 \max(\alpha^{-6}, \lambda^{-3})}{\epsilon B (1-\gamma)^9 \underline{\pi_{\mathrm{ref}}}^{7-7\alpha} \rho_{\min}^5 \underline{\pi_{\mathrm{ref}}}^4} \exp_\alpha \left( -\frac{|\log(\underline{\pi_{\mathrm{ref}}})|}{\lambda \alpha^2 (1-\gamma)^2 \rho_{\min}} \max(\lambda, \alpha^2) \right)^{4\alpha-8} .$$

*Proof.* To prove this corollary, we will show that under the conditions of this corollary, the assumptions of Theorem E.6 holds. Firstly, note that by using Lemma F.4, the assumption $\mathbf{A}_f(\pi_{\mathrm{ref}})$ holds. Secondly, using Lemma F.4 note that

$$\frac{4}{(1-\gamma)^2} + \frac{1}{1-\gamma} \log \left( \frac{216 \omega_f^2}{\epsilon \underline{\mu}_f (1-\gamma)^4} \left[ 4 + 4\lambda^2 \mathrm{d}_f^2 + \lambda^2 (1-\gamma)^2 \mathrm{y}_f^2 \right] \right)$$

$$\leq \frac{4}{(1-\gamma)^2} + \frac{1}{1-\gamma} \log \left( \frac{216 \underline{\pi_{\mathrm{ref}}}^{2\alpha-2}}{\epsilon \underline{\mu}_f (1-\gamma)^4} \left[ 4 + 32\lambda^2 \frac{|\log(\underline{\pi_{\mathrm{ref}}})|^2}{\alpha^4} \right] \right)$$

$$\leq \frac{4}{(1-\gamma)^2} + \frac{1}{1-\gamma} \log \left( \frac{864 \underline{\pi_{\mathrm{ref}}}^{4\alpha-4} \left[ 1 + 32\lambda^2 |\log(\underline{\pi_{\mathrm{ref}}})|^2 / \alpha^4 \right]}{\epsilon \lambda (1-\gamma)^5 \rho_{\min}^2 \underline{\pi_{\mathrm{ref}}}^2} \right)$$

$$- \frac{1}{1-\gamma} \log \left( \exp_\alpha \left( -\frac{16 + 32\gamma\lambda |\log(\underline{\pi_{\mathrm{ref}}})|}{\lambda \alpha^2 (1-\gamma)^2 \rho_{\min}} \right)^{4-2\alpha} \right),$$

where in the last inequality, we used the lower bound on $\underline{\mu}_f$ provided in Lemma F.4. Next using the definition of $\exp_\alpha$ (see Section 2) and the fact that $\lambda \leq 1$, we have

$$\frac{4}{(1-\gamma)^2} + \frac{1}{1-\gamma} \log \left( \frac{216 \omega_f^2}{\epsilon \underline{\mu}_f (1-\gamma)^4} \left[ 4 + 4\lambda^2 \mathrm{d}_f^2 + \lambda^2 (1-\gamma)^2 \mathrm{y}_f^2 \right] \right)$$

$$\leq \frac{4}{(1-\gamma)^2} + \frac{1}{1-\gamma} \log \left( \frac{28152 \underline{\pi_{\mathrm{ref}}}^{4\alpha-4} |\log(\underline{\pi_{\mathrm{ref}}})|^2}{\epsilon \lambda \alpha^4 (1-\gamma)^5 \rho_{\min}^2 \underline{\pi_{\mathrm{ref}}}^2} \max(\alpha^2, \lambda)^2 \right)$$

$$+ \frac{4-2\alpha}{1-\alpha} \frac{1}{1-\gamma} \log \left( 1 + (1-\alpha) \left( \frac{48 |\log(\underline{\pi_{\mathrm{ref}}})|}{\lambda \alpha^2 (1-\gamma)^2 \rho_{\min}} \max(\alpha^2, \lambda) \right) \right)$$

$$\leq \frac{1}{1-\gamma} \log \left( \frac{28152 \underline{\pi_{\mathrm{ref}}}^{4\alpha-4} |\log(\underline{\pi_{\mathrm{ref}}})|^2}{\epsilon \lambda \alpha^4 (1-\gamma)^5 \rho_{\min}^2 \underline{\pi_{\mathrm{ref}}}^2} \max(\alpha^2, \lambda)^2 \right) + \frac{196 |\log(\underline{\pi_{\mathrm{ref}}})|}{\lambda \alpha^2 (1-\gamma)^3 \rho_{\min}} \max(\alpha^2, \lambda) ,$$

where in the last inequality, we used that for $x \geq 0$, we have $\log(1+x) \leq x$. This shows that our condition on $H$ guarantees that the one set in Theorem E.9 is satisfied. Next, using again Lemma F.4 observe that

$$L_f = 196 \frac{\underline{\pi_{\mathrm{ref}}}^{2\alpha-2} |\log(\underline{\pi_{\mathrm{ref}}})|}{\alpha^2 (1-\gamma)^3} \max(\alpha^2, \lambda) , \quad \sigma_f^2 \leq \frac{204 \underline{\pi_{\mathrm{ref}}}^{3\alpha-3} \log(\underline{\pi_{\mathrm{ref}}})^2}{\alpha^4 (1-\gamma)^4} \max(\alpha^2, \lambda)^2 \tag{98}$$

$$\underline{\mu}_f \geq \lambda(1-\gamma)\underline{\pi_{\text{ref}}}^{2-2\alpha}\rho_{\min}^2\underline{\pi_{\text{ref}}}^2\exp_\alpha\left(-\frac{48|\log(\underline{\pi_{\text{ref}}})|}{\lambda\alpha^2(1-\gamma)^2\rho_{\min}}\max(\lambda,\alpha^2)\right)^{4-2\alpha} . \tag{99}$$

Hence, we have that

$$
\begin{aligned}
&\min\left(\frac{1}{2L_f},\frac{\epsilon B\underline{\mu}_f}{18\sigma_f^2}\right)\\
&\geq \min\left(\frac{\alpha^2(1-\gamma)^3\min(\alpha^{-2},\lambda^{-1})}{392\underline{\pi_{\text{ref}}}^{2\alpha-2}|\log(\underline{\pi_{\text{ref}}})|},\right.\\
&\qquad\left.\frac{\epsilon B\lambda(1-\gamma)^5\alpha^4\underline{\pi_{\text{ref}}}^{5-5\alpha}\rho_{\min}^2\underline{\pi_{\text{ref}}}^2}{3672\log(\underline{\pi_{\text{ref}}})^2\max(\alpha^2,\lambda)^2}\exp_\alpha\left(-\frac{48|\log(\underline{\pi_{\text{ref}}})|}{\lambda\alpha^2(1-\gamma)^2\rho_{\min}}\max(\lambda,\alpha^2)\right)^{4-2\alpha}\right)
\end{aligned}
\tag{100}
$$

$$= \frac{\epsilon B\lambda(1-\gamma)^5\alpha^4\underline{\pi_{\text{ref}}}^{5-5\alpha}\rho_{\min}^2\underline{\pi_{\text{ref}}}^2}{3672\log(\underline{\pi_{\text{ref}}})^2\max(\alpha^2,\lambda)^2}\exp_\alpha\left(-\frac{48|\log(\underline{\pi_{\text{ref}}})|}{\lambda\alpha^2(1-\gamma)^2\rho_{\min}}\max(\lambda,\alpha^2)\right)^{4-2\alpha} , \tag{101}$$

where in the last identity, we used the fact that $\epsilon < (1-\gamma)^{-1}$ and that

$$B \leq \frac{1}{\underline{\pi_{\text{ref}}}^2\rho_{\min}^2}\exp_\alpha\left(-\frac{48|\log(\underline{\pi_{\text{ref}}})|}{\lambda\alpha^2(1-\gamma)^2\rho_{\min}}\max(\lambda,\alpha^2)\right)^{2\alpha-4} .$$

This shows that our condition on $\eta$ guarantees that the one set in Theorem E.9 is satisfied. Finally using (101) and (99), we have

$$
\begin{aligned}
&\frac{4}{\underline{\mu}_f}\max\left(2L_f,\frac{18\sigma_f^2}{\epsilon B\underline{\mu}_f}\right)\\
&\leq \frac{4}{\underline{\mu}_f}\frac{3672\log(\underline{\pi_{\text{ref}}})^2\max(\alpha^2,\lambda)^2}{\epsilon B\lambda(1-\gamma)^5\alpha^4\underline{\pi_{\text{ref}}}^{5-5\alpha}\rho_{\min}^2\underline{\pi_{\text{ref}}}^2}\exp_\alpha\left(-\frac{48|\log(\underline{\pi_{\text{ref}}})|}{\lambda\alpha^2(1-\gamma)^2\rho_{\min}}\max(\lambda,\alpha^2)\right)^{2\alpha-4}\\
&\leq \frac{14688\log(\underline{\pi_{\text{ref}}})^2\max(\alpha^2,\lambda)^2}{\lambda^2\epsilon B(1-\gamma)^6\alpha^4\underline{\pi_{\text{ref}}}^{7-7\alpha}\rho_{\min}^4\underline{\pi_{\text{ref}}}^4}\exp_\alpha\left(-\frac{48|\log(\underline{\pi_{\text{ref}}})|}{\lambda\alpha^2(1-\gamma)^2\rho_{\min}}\max(\lambda,\alpha^2)\right)^{4\alpha-8} ,
\end{aligned}
$$

which concludes the proof. $\qquad\square$

**Corollary F.6.** *Assume that, for some $\pi_{\text{ref}} \in (0,1/4]$, $\pi_{\text{ref}}$ satisfy $\boldsymbol{P}(\pi_{\text{ref}})$. Assume in addition that the initial distribution $\rho$ satisfies $\boldsymbol{A}_\rho$ and fix $f$ to be the $\alpha$-Csiszár–Cressie–Read divergence generator, i.e.*

$$f(u) = f_\alpha(u) = \frac{u^\alpha - \alpha u + \alpha - 1}{\alpha(\alpha-1)} .$$

*Fix any $\epsilon \in (0,(1-\gamma)^{-1}]$, such that*

$$\epsilon < \frac{16}{(1-\gamma)^3\rho_{\min}}\cdot\frac{1-\alpha}{(\pi_{\text{ref}}/2)^{\alpha-1}-1} , \quad \text{and set } \lambda = \frac{(1-\gamma)\alpha^2\epsilon}{16|\log(\underline{\pi_{\text{ref}}})|} . \tag{102}$$

*Additionally set any $B$ such that*

$$B \leq \frac{1}{\epsilon^2\alpha^2(1-\gamma)^3\rho_{\min}^2\underline{\pi_{\text{ref}}}^2}\exp_\alpha\left(-\frac{384|\log(\underline{\pi_{\text{ref}}})|}{\epsilon\alpha^2(1-\gamma)^3\rho_{\min}}\right)^{2\alpha-4} \tag{103}$$

*Setting*

$$H \geq \frac{1}{1-\gamma}\log\left(\frac{19008\underline{\pi_{\text{ref}}}^{4\alpha-2}|\log(\underline{\pi_{\text{ref}}})|}{\epsilon\alpha^2(1-\gamma)^6\rho_{\min}^2\underline{\pi_{\text{ref}}}^2}\right) + \frac{1540|\log(\underline{\pi_{\text{ref}}})|}{\epsilon\alpha^2(1-\gamma)^4\rho_{\min}} ,$$

*and*

$$\eta \leq \frac{\epsilon^2\alpha^2(1-\gamma)^6 B\rho_{\min}^2\underline{\pi_{\text{ref}}}^2}{13824\underline{\pi_{\text{ref}}}^{5\alpha-5}|\log(\underline{\pi_{\text{ref}}})|}\exp_\alpha\left(-\frac{384|\log(\underline{\pi_{\text{ref}}})|}{\epsilon\alpha^2(1-\gamma)^3\rho_{\min}}\right)^{4-2\alpha} , \tag{104}$$

*and*

$$T \geq \frac{12^7 \underline{\pi_{\mathrm{ref}}}^{7\alpha-7} |\log(\underline{\pi_{\mathrm{ref}}})|^2}{\epsilon^3 \alpha^4 (1-\gamma)^8 \rho_{\min}^4 \underline{\pi_{\mathrm{ref}}}^4 B} \exp_\alpha \left( -\frac{384 |\log(\underline{\pi_{\mathrm{ref}}})|}{\epsilon \alpha^2 (1-\gamma)^3 \rho_{\min}} \right)^{4\alpha-8} \log \left( \frac{6(v_\star^f(\rho) - v_{\theta_0}^f(\rho))}{\epsilon} \right) ,$$
(105)

*guarantees that*

$$v_\star(\rho) - \mathbb{E}\left[ v_{\theta_t}(\rho) \right] \leq \epsilon .$$

*Thus, the sample complexity of $f$-PG, where $f$ is the $\alpha$-Csiszár–Cressie–Read divergence generator, to learn an $\epsilon$-solution of the non-regularized problem is*

$$TBH \approx \frac{\underline{\pi_{\mathrm{ref}}}^{7\alpha-7} |\log(\underline{\pi_{\mathrm{ref}}})|^3}{\epsilon^4 \alpha^6 (1-\gamma)^{12} \rho_{\min}^5 \underline{\pi_{\mathrm{ref}}}^4} \exp_\alpha \left( -\frac{384 |\log(\underline{\pi_{\mathrm{ref}}})|}{\epsilon \alpha^2 (1-\gamma)^3 \rho_{\min}} \right)^{4\alpha-8} .$$

*Proof.* To prove this corollary, we will show that under the conditions of this corollary, the assumptions of Theorem E.9 holds. Firstly, note that (102) implies that

$$\epsilon < \frac{16}{(1-\gamma)^3 \rho_{\min}} \min \left( \frac{4}{|f'(\iota_f)|}, \frac{1}{|f'(\frac{1}{2})|}, \frac{4}{|f'(\frac{1}{2}\underline{\pi_{\mathrm{ref}}})|} \right) , \quad \lambda = \frac{(1-\gamma)\epsilon}{4} \mathrm{c}_f ,$$

with $\mathrm{c}_f = \alpha^2/4 |\log(\underline{\pi_{\mathrm{ref}}})|$. Additionally, observe using Lemma F.4 that we have

$$C_f^{(1)} \leq \frac{64 \underline{\pi_{\mathrm{ref}}}^{2\alpha-2} |\log(\underline{\pi_{\mathrm{ref}}})|}{\alpha^2}, \quad C_f^{(2)} \leq 96 \underline{\pi_{\mathrm{ref}}}^{2\alpha-2}, \quad C_f^{(3)} \leq \frac{13824 \underline{\pi_{\mathrm{ref}}}^{5\alpha-5} |\log(\underline{\pi_{\mathrm{ref}}})|}{\alpha^2},$$

$$d(\epsilon) = \exp_\alpha \left( -\frac{384 |\log(\underline{\pi_{\mathrm{ref}}})|}{\epsilon \alpha^2 (1-\gamma)^3 \rho_{\min}} \right)^{4-2\alpha} ,$$
(106)

where $C_f^{(1)}, C_f^{(2)}, C_f^{(3)}$, and $d(\epsilon)$ are defined in (78) and (77). Next, observe that

$$\frac{4}{(1-\gamma)^2} + \frac{1}{1-\gamma} \log \left( \frac{297 \omega_f^2 C_1^f}{\epsilon d(\epsilon)(1-\gamma)^6 \rho_{\min}^2 \underline{\pi_{\mathrm{ref}}}^2} \right)$$

$$\leq \frac{4}{(1-\gamma)^2} + \frac{1}{1-\gamma} \log \left( \frac{19008 \underline{\pi_{\mathrm{ref}}}^{4\alpha-2} |\log(\underline{\pi_{\mathrm{ref}}})|}{\epsilon \alpha^2 (1-\gamma)^6 \rho_{\min}^2 \underline{\pi_{\mathrm{ref}}}^2} \right)$$

$$+ \frac{4-2\alpha}{1-\gamma} \log \left( \exp_\alpha \left( -\frac{384 |\log(\underline{\pi_{\mathrm{ref}}})|}{\epsilon \alpha^2 (1-\gamma)^3 \rho_{\min}} \right) \right)$$

$$\leq \frac{4}{(1-\gamma)^2} + \frac{1}{1-\gamma} \log \left( \frac{19008 \underline{\pi_{\mathrm{ref}}}^{4\alpha-2} |\log(\underline{\pi_{\mathrm{ref}}})|}{\epsilon \alpha^2 (1-\gamma)^6 \rho_{\min}^2 \underline{\pi_{\mathrm{ref}}}^2} \right) + \frac{1536 |\log(\underline{\pi_{\mathrm{ref}}})|}{\epsilon \alpha^2 (1-\gamma)^4 \rho_{\min}} ,$$

where in the last inequality, we used that $\log(1+u) \leq x$ for $u > -1$. This shows that our condition on $H$ implies the one assumed in Theorem E.9. Using (106), we have

$$\min \left( \frac{(1-\gamma)^3}{C_f^{(2)}}, \frac{\epsilon^2 d(\epsilon)(1-\gamma)^6 B \rho_{\min}^2 \underline{\pi_{\mathrm{ref}}}^2}{C_f^{(3)}} \right)$$

$$\geq \min \left( \frac{(1-\gamma)^3}{96 \underline{\pi_{\mathrm{ref}}}^{2\alpha-2}}, \frac{\epsilon^2 \alpha^2 (1-\gamma)^6 B \rho_{\min}^2 \underline{\pi_{\mathrm{ref}}}^2}{13824 \underline{\pi_{\mathrm{ref}}}^{5\alpha-5} |\log(\underline{\pi_{\mathrm{ref}}})|} \exp_\alpha \left( -\frac{384 |\log(\underline{\pi_{\mathrm{ref}}})|}{\epsilon \alpha^2 (1-\gamma)^3 \rho_{\min}} \right)^{4-2\alpha} \right)$$

$$\geq \frac{\epsilon^2 \alpha^2 (1-\gamma)^6 B \rho_{\min}^2 \underline{\pi_{\mathrm{ref}}}^2}{13824 \underline{\pi_{\mathrm{ref}}}^{5\alpha-5} |\log(\underline{\pi_{\mathrm{ref}}})|} \exp_\alpha \left( -\frac{384 |\log(\underline{\pi_{\mathrm{ref}}})|}{\epsilon \alpha^2 (1-\gamma)^3 \rho_{\min}} \right)^{4-2\alpha} ,$$
(107)

where in the last inequality, we used the condition on $B$ introduced in (103). Hence our condition on the step-size ensures that the one assumed in Theorem E.9 is satisfied. Next, using (106) and (107) yields

$$\frac{16 C_f^{(1)} \ell(\epsilon)}{\epsilon d(\epsilon)(1-\gamma)^2 \rho_{\min}^2 \underline{\pi_{\mathrm{ref}}}^2} \max \left( \frac{C_f^{(2)}}{(1-\gamma)^3}, \frac{C_f^{(3)}}{\epsilon^2 d(\epsilon)(1-\gamma)^6 B \rho_{\min}^2 \underline{\pi_{\mathrm{ref}}}^2} \right)$$

$$\leq \frac{16 C_f^{(1)}\ell(\epsilon)}{\epsilon d(\epsilon)(1-\gamma)^2\rho_{\min}^2\underline{\pi_{\text{ref}}}^2}\cdot\frac{13824\underline{\pi_{\text{ref}}}^{5\alpha-5}|\log(\underline{\pi_{\text{ref}}})|}{\epsilon^2\alpha^2(1-\gamma)^6 B\rho_{\min}^2\underline{\pi_{\text{ref}}}^2}\exp_\alpha\left(-\frac{384|\log(\underline{\pi_{\text{ref}}})|}{\epsilon\alpha^2(1-\gamma)^3\rho_{\min}}\right)^{2\alpha-4}$$

$$\leq \frac{1024\underline{\pi_{\text{ref}}}^{2\alpha-2}|\log(\underline{\pi_{\text{ref}}})|\ell(\epsilon)}{\epsilon\alpha^2(1-\gamma)^2\rho_{\min}^2\underline{\pi_{\text{ref}}}^2}\cdot\frac{13824\underline{\pi_{\text{ref}}}^{5\alpha-5}|\log(\underline{\pi_{\text{ref}}})|}{\epsilon^2\alpha^2(1-\gamma)^6 B\rho_{\min}^2\underline{\pi_{\text{ref}}}^2}\exp_\alpha\left(-\frac{384|\log(\underline{\pi_{\text{ref}}})|}{\epsilon\alpha^2(1-\gamma)^3\rho_{\min}}\right)^{4\alpha-8}$$

$$\leq \frac{12^7\underline{\pi_{\text{ref}}}^{7\alpha-7}|\log(\underline{\pi_{\text{ref}}})|^2}{\epsilon^3\alpha^4(1-\gamma)^8\rho_{\min}^4\underline{\pi_{\text{ref}}}^4 B}\exp_\alpha\left(-\frac{384|\log(\underline{\pi_{\text{ref}}})|}{\epsilon\alpha^2(1-\gamma)^3\rho_{\min}}\right)^{4\alpha-8}\log\left(\frac{6(v_\star^f(\rho)-v_{\theta_0}^f(\rho))}{\epsilon}\right) \,,$$

which proves that under our condition on $T$ the one assumed by Theorem E.9 is satisfied. $\qquad\square$

**Corollary F.7.** *Assume the same condition of Corollary F.6. For any $(1-\gamma)^{-1} > \epsilon > 0$ and $\alpha \in (0,1)$, denote respectively by $T(\epsilon,\alpha)$, $B(\epsilon,\alpha)$, and $H(\epsilon,\alpha)$, the thresholds set in Corollary F.6 on $T$, $B$, and $H$, to learn an $\epsilon$-solution of the unregularized problem. Addtionnaly, denote by $\alpha^\star(\epsilon)$ the minimizer of $T(\epsilon,\alpha)B(\epsilon,\alpha)H(\epsilon,\alpha)$. It holds that*

$$\alpha^\star(\epsilon) = \frac{11}{2}\cdot\frac{1}{\log(1/\epsilon)}+o\left(\frac{1}{\log(1/\epsilon)}\right) \,.$$

*Additionally, for $\epsilon < e^{-11}$, it holds that*

$$T(\epsilon,\alpha^\star(\epsilon))B(\epsilon,\alpha^\star(\epsilon))H(\epsilon,\alpha^\star(\epsilon)) = \widetilde{O}\left(\frac{|\log(\underline{\pi_{\text{ref}}})|^{11}}{\epsilon^{12}(1-\gamma)^{36}\rho_{\min}^{14}\underline{\pi_{\text{ref}}}^{11}}\log\left(\frac{(v_\star^f(\rho)-v_{\theta_0}^f(\rho))}{\epsilon}\right)\right) \,.$$

*Proof.* We first provide an equivalent of the $\alpha$ that optimises the sample complexity provided in Corollary F.6 and then bound the sample complexity obtained by using this $\alpha$.

**Finding the best $\alpha$.** Firstly, note that using Corollary F.6, we have

$$S(\epsilon,\alpha) := \log\left(T(\epsilon,\alpha)B(\epsilon,\alpha)H(\epsilon,\alpha)\right)$$

$$= \log\left(\frac{12^7\underline{\pi_{\text{ref}}}^{7\alpha-7}|\log(\underline{\pi_{\text{ref}}})|^2}{\epsilon^3\alpha^4(1-\gamma)^8\rho_{\min}^4\underline{\pi_{\text{ref}}}^4}\right)$$

$$+ \frac{4\alpha-8}{\alpha-1}\log\left(1+(1-\alpha)\left(\frac{384|\log(\underline{\pi_{\text{ref}}})|}{\epsilon\alpha^2(1-\gamma)^3\rho_{\min}}\right)\right)$$

$$+ \log\left(\log\left(\frac{6(v_\star^f(\rho)-v_{\theta_0}^f(\rho))}{\epsilon}\right)\right)$$

$$+ \log\left(\frac{1}{1-\gamma}\log\left(\frac{19008\underline{\pi_{\text{ref}}}^{4\alpha-2}|\log(\underline{\pi_{\text{ref}}})|}{\epsilon\alpha^2(1-\gamma)^6\rho_{\min}^2\underline{\pi_{\text{ref}}}^2}\right)+\frac{1540|\log(\underline{\pi_{\text{ref}}})|}{\epsilon\alpha^2(1-\gamma)^4\rho_{\min}}\right) \,.$$

Firstly, observe that for any function $k(\epsilon)$ which does converge to a different value from 0, we have

$$\lim_{\varepsilon\to0}\frac{S(\varepsilon,1/\log(1/\varepsilon))}{S(\varepsilon,k(\varepsilon))} < 1 \,,$$

which establishes that $\alpha^\star(\epsilon)\to0$. This allows, to rewrite $S(\epsilon,\alpha)$ as

$$S(\epsilon,\alpha) = \log\left(\frac{1}{\epsilon^4\alpha^6}\right)+\frac{8-4\alpha}{1-\alpha}\log\left(\frac{1}{\epsilon\alpha^2}\right)+\psi(\alpha,\epsilon) \,,$$

where $\psi(\alpha,\epsilon)$ is defined as

$$\psi(\alpha,\epsilon) = \log\left(\frac{12^7\underline{\pi_{\text{ref}}}^{7\alpha-7}|\log(\underline{\pi_{\text{ref}}})|^2}{(1-\gamma)^8\rho_{\min}^4\underline{\pi_{\text{ref}}}^4}\right)$$

$$+ \frac{4\alpha-8}{\alpha-1}\left[\log\left(1+(1-\alpha)\left(\frac{384|\log(\underline{\pi_{\text{ref}}})|}{\epsilon\alpha^2(1-\gamma)^3\rho_{\min}}\right)\right)-\log\left(\frac{1}{\epsilon\alpha^2}\right)\right]$$

$$+ \log\left(\log\left(\frac{6(v_\star^f(\rho)-v_{\theta_0}^f(\rho))}{\epsilon}\right)\right)$$

$$+ \log \left( \frac{\epsilon \alpha^2}{1 - \gamma} \log \left( \frac{19008 {\pi_{\text{ref}}}^{4\alpha - 2} |\log(\pi_{\text{ref}})|}{\epsilon \alpha^2 (1 - \gamma)^6 \rho_{\min}^2 \underline{\pi_{\text{ref}}}^2} \right) + \frac{1540 |\log(\pi_{\text{ref}})|}{(1 - \gamma)^4 \rho_{\min}} \right) \ .$$

Importantly, observe that $\psi(\alpha, \epsilon)$ is dominated by $\log(1/\epsilon)$ when $(\alpha, \epsilon) \to 0$ and that

$$\frac{\partial \psi(\alpha, \epsilon)}{\partial \alpha} = o \left( \frac{1}{\alpha \log(\frac{1}{\epsilon \alpha})} \right) \ .$$

Computing the derivative of this function with respect to $\alpha$ yields

$$\frac{\partial S(\epsilon, \alpha)}{\partial \alpha} = \frac{-6}{\alpha} + 4 \frac{1}{(1 - \alpha)^2} \log \left( \frac{1}{\epsilon} \right) + 4 \frac{1}{(1 - \alpha)^2} \log \left( \frac{1}{\alpha^2} \right) - \frac{8}{\alpha} \left( 1 + \frac{1}{1 - \alpha} \right) + \frac{\partial \psi(\epsilon, \alpha)}{\partial \alpha} \tag{108}$$

$$= \frac{-22}{\alpha} + 4 \log \left( \frac{1}{\epsilon} \right) + o \left( \frac{1}{\alpha \log(\frac{1}{\epsilon \alpha})} \right) \ . \tag{109}$$

As

$$\left. \frac{\partial S(\epsilon, \alpha)}{\partial \alpha} \right|_{\alpha = \alpha^\star(\epsilon)} = 0 \ ,$$

Then this implies that

$$\alpha^\star(\epsilon) = \frac{11}{2} \cdot \frac{1}{\log(1/\epsilon)} + o \left( \frac{1}{\log(1/\epsilon)} \right) \ .$$

Next, we provide a bound on the sample complexity given by this $\alpha^\star(\epsilon)$.

**Computing the sample complexity.** Firstly, note that for $\alpha \leq 1/2$ and $\epsilon < 1$, we have

$$T(\epsilon, \alpha) B(\epsilon, \alpha) H(\epsilon, \alpha)$$

$$\leq \frac{12^{11} {\pi_{\text{ref}}}^{7\alpha - 7} |\log(\pi_{\text{ref}})|^3}{\epsilon^4 \alpha^6 (1 - \gamma)^{12} 2 \rho_{\min}^5 \underline{\pi_{\text{ref}}}^4} \exp_\alpha \left( -\frac{384 |\log(\pi_{\text{ref}})|}{\epsilon \alpha^2 (1 - \gamma)^3 \rho_{\min}} \right)^{4\alpha - 8} \log \left( \frac{6 (v_\star^f(\rho) - v_{\theta_0}^f(\rho))}{\epsilon} \right)$$

$$\leq \frac{{\pi_{\text{ref}}}^{7\alpha - 7} |\log(\pi_{\text{ref}})|^3}{\epsilon^4 \alpha^6 (1 - \gamma)^{12} \rho_{\min}^5 \underline{\pi_{\text{ref}}}^4} \left( \frac{|\log(\pi_{\text{ref}})|}{\epsilon \alpha^2 (1 - \gamma)^3 \rho_{\min}} \right)^{(8 - 4\alpha)/(1 - \alpha)} \log \left( \frac{6 (v_\star^f(\rho) - v_{\theta_0}^f(\rho))}{\epsilon} \right) \ . \tag{110}$$

Next, note that for $\epsilon < e^{-11}$, we have $\alpha = \frac{11}{2 \log(1/\epsilon)} \leq 1/2$. Thus, using that $1/(1 - \alpha) \leq 1 + 2\alpha$ yield

$$\left( \frac{1}{\epsilon \alpha^2} \right)^{(8 - 4\alpha)/(1 - \alpha)} = \exp \left( \frac{8 - 4\alpha}{1 - \alpha} \cdot \log \left( \frac{1}{\epsilon \alpha^2} \right) \right) \leq \exp \left( (8 + 8\alpha) \cdot \log \left( \frac{1}{\epsilon \alpha^2} \right) \right) \lesssim \frac{1}{\epsilon^8 \alpha^{16}} \ .$$

Plugging in the previous bound in (110) concludes the proof. $\qquad\square$

# G  DISCUSSION ON UNREGULARIZED POLICY GRADIENT WITH $f$-SOFTARGMAX PARAMETERIZATION

In this section, we show that it is possible to derive Non-Uniform Łojasiewicz inequalities on the unregularized objective $v_\theta(\rho) := v_{\pi_\theta^f}(\rho)$ under $f$-SoftArgmax Parameterization in the bandit setting. We expect the analysis to extend to the RL setting using similar arguments to those of Mei et al. (2020b;a); Liu et al. (2025).

**Theorem G.1.** *Consider the bandit case, i.e. $|\mathcal{S}| = 1$. Assume that, for some $\underline{\pi_{\text{ref}}} > 0$, $f$ and $\pi_{\text{ref}}$ satisfy $\mathbf{A}_f(\underline{\pi_{\text{ref}}})$ and $\mathbf{P}(\underline{\pi_{\text{ref}}})$ respectively. We also assume that the function $1/\overline{f''}$ is convex and that the initial distribution $\rho$ satisfies $\mathbf{A}_\rho$. Then, it holds that*

$$\|\nabla v_\theta(\rho)\|_2^2 \geq \frac{\mathrm{w}_\theta^f(a^\star)^2}{\zeta_f} \frac{1}{v_\star(\rho) - v_{\pi_{\text{ref}}}(\rho)} \cdot f'' \left( \frac{v_\star(\rho) - v_\theta(\rho)}{v_\star(\rho) - v_{\pi_{\text{ref}}}(\rho)} \right) \ ,$$

*where $v_\star(\rho) = \max\limits_{\pi \in \mathcal{P}(\mathcal{A})^\mathcal{S}} v_\pi(\rho)$.*

*Proof.* Subsequently, we drop the dependency on the state for more clarity. Using Lemma C.2 (with $\gamma = 0$ and $|\mathcal{S}| = 1$), for any $a \in \mathcal{A}$, we have

$$\frac{\partial v_\theta(\rho)}{\partial \theta(a)} = \mathrm{w}_\theta^f(a) \cdot \left(\mathrm{r}(a) - \sum_b \mathrm{w}_\theta^f(b)\mathrm{r}(b)\right).$$

Denote by $a^\star$ any optimal action, and $\mathrm{r}^\star = \mathrm{r}(a^\star)$. We have

$$\|\nabla v_\theta(\rho)\|_2^2 = \sum_a \mathrm{w}_\theta^f(a)^2 \cdot \left(\mathrm{r}(a) - \sum_b \mathrm{w}_\theta^f(b)\mathrm{r}(b)\right)^2 \geq \mathrm{w}_\theta^f(a^\star)^2 \left(\sum_b \mathrm{w}_\theta^f(b)(\mathrm{r}^\star - \mathrm{r}(b))\right)^2,$$

Using that $\Delta(a) = \mathrm{r}^\star - \mathrm{r}(a) \geq 0$, and also

$$\mathrm{w}_\theta^f(a) = \frac{1}{\mathrm{W}_\theta^f} \cdot \frac{\pi_{\mathrm{ref}}(a)}{f''(\pi_\theta^f(a)/\pi_{\mathrm{ref}}(a))} \quad , \quad \mathrm{W}_\theta^f := \sum_{b \in \mathcal{A}} \frac{\pi_{\mathrm{ref}}(a)}{f''(\pi_\theta^f(b)/\pi_{\mathrm{ref}}(b))} \quad ,$$

yields

$$\sum_b \mathrm{w}_\theta^f(b)\Delta(b) = \frac{1}{\mathrm{W}_\theta^f} \cdot \sum_{b \in \mathcal{A}} \Delta(b)\pi_{\mathrm{ref}}(b) \cdot \frac{1}{f''(\pi_\theta^f(b)/\pi_{\mathrm{ref}}(b))}.$$

Let's introduce a distribution $\gamma(a) = \Delta(a)\pi_{\mathrm{ref}}(a)/(\sum_{b \in \mathcal{A}} \Delta(b)\pi_{\mathrm{ref}}(b))$, then we have

$$\sum_{b \in \mathcal{A}} \mathrm{w}_\theta^f(b)\Delta(b) = \frac{1}{\mathrm{W}_\theta^f \cdot \mathbb{E}_{c \sim \pi_{\mathrm{ref}}}[\Delta(c)]} \mathbb{E}_{c \sim \gamma} \left[1/(f''(\pi_\theta^f(c)/\pi_{\mathrm{ref}}(c)))\right] .$$

Next, we use that a map $x \mapsto 1/f''(x)$ is convex, and thus was have

$$\mathbb{E}_{c \sim \gamma} \left[1/(f''(\pi_\theta^f(c)/\pi_{\mathrm{ref}}(c)))\right] \geq 1/f'' \left(\mathbb{E}_{c \sim \gamma} \left[\pi_\theta^f(c)/\pi_{\mathrm{ref}}(c)\right]\right)$$

$$= 1/f'' \left(\frac{\sum_{c \in \mathcal{A}} \Delta(c)\pi_\theta^f(c)}{\sum_{c \in \mathcal{A}} \Delta(c)\pi_{\mathrm{ref}}(c)}\right).$$

Overall, we have

$$\sum_{b \in \mathcal{A}} \pi_\theta^f(b)\Delta(b) \geq \frac{1}{\mathrm{W}_\theta^f} \frac{1}{\left(\sum_{c \in \mathcal{A}} \Delta(c)\pi_{\mathrm{ref}}(c)\right) f'' \left(\frac{\sum_{c \in \mathcal{A}} \Delta(c)\pi_\theta^f(c)}{\sum_{c \in \mathcal{A}} \Delta(c)\pi_{\mathrm{ref}}(c)}\right)} .$$

Finally, using that $\mathrm{W}_\theta^f \leq \zeta_f$ concludes the proof. $\qquad\square$

Similarly, to Mei et al. (2020b); Liu et al. (2024; 2025), this Łojasiewicz inequality depends on the probability of the optimal action, which is very restrictive. Although extending the analysis of Mei et al. (2020b); Liu et al. (2024; 2025) in the deterministic setting is possible, addressing the stochastic setting for this type of Łojasiewicz inequality appears very challenging. This justifies adding a regularizer to the objective to ensure better PL inequalities, and on which the minimal coefficient can be lower bounded on the trajectory by leveraging a proper projection operator.

## H LINKS WITH MIRROR DESCENT

We stress that our proposed method is fundamentally different from mirror descent. Let us define a mapping $\Phi(\pi) = \sum_{s \in \mathcal{S}} \mathrm{D}^f(\pi(\cdot|s)\|\pi_{\mathrm{ref}}(\cdot|s))$. For the functions $f$ that we consider, $\Phi$ is Legendre on the positive orthant and separable across states (Bubeck et al., 2015). In this case, the $f$-regularized value function $v_\pi^f(\rho)$ can be optimized directly in the policy space via the Lazy Mirror Descent algorithm (or dual averaging; see Nesterov (2009); Xiao (2009); Juditsky et al. (2023)) with $\Phi$ as mirror map. Denoting by $\pi_t$ the policy at step $t$ and $\widetilde{\pi}_t$ by the unnormalized policy at step $t$, the lazy MD updates reads:

$$\nabla\Phi(\widetilde{\pi}_{t+1}) = \nabla\Phi(\widetilde{\pi}_t) + \eta\nabla_\pi v_\pi^f(\rho)|_{\pi=\pi_t} , \quad \pi_{t+1} = \arg\min_{\pi \in \Pi} B_\Phi(\pi\|\widetilde{\pi}_{t+1}) . \tag{111}$$

where $\Pi = \mathcal{P}(\mathcal{A})^{|\mathcal{S}|}$ is a policy space and $B_\Phi(\pi\|\pi') = \Phi(\pi) - \Phi(\pi') - \langle \nabla\Phi(\pi'), \pi - \pi'\rangle$ is the corresponding Bregman divergence. Since $\Phi$ is separable over states, the Bregman projection can be written state-wise as $\pi_{t+1}(\cdot|s) = \text{f-softargmax}(\nabla\Phi(\widetilde{\pi}_{t+1})(s,\cdot), \pi_{\text{ref}}(\cdot|s))$.

By denoting $\theta_t = \nabla\Phi(\widetilde{\pi}_t)$, one obtains updates that resemble those of (10) (after the removal of $\mathcal{T}$), with one important difference: the gradient in (111) is taken with respect to the policy $\pi$ whereas in (10) it is computed w.r.t the "dual" parameter $\theta$ (in the MD terminology). Even more important, the update (111) can be expressed as, by the chain rule

$$\theta_{t+1} = \theta_t + \eta \left[ \frac{\partial \pi_\theta^f}{\partial \theta}\Big|_{\theta=\theta_t} \right]^{-1} \nabla_\theta J^f(\theta_t),$$

which have an additional preconditioning term given by the inverse of the policy Jacobian.

A crucial feature of (10) is that it performs a gradient ascent in the "dual" space directly. This algorithm can be extended in the non-tabular setting directly, by parameterizing the function $\theta(s,a)$, allowing extensions to deep RL. This is in contrast with Lazy-MD methods (Nesterov, 2009; Xiao, 2009; Juditsky et al., 2023), due to preconditioning, which cannot be expressed as direct parameter-space gradient steps. This remark has several important implications, which we leave for future work.

## I  TECHNICAL LEMMAS

**Lemma I.1** (Lemma 1.2.3 in Nesterov (2004)). *Let $f : \mathbb{R}^d \to \mathbb{R}$ be twice continuously differentiable. Suppose there exists $L \geq 0$ such that for all $x \in \mathbb{R}^d$ and $v \in \mathbb{R}^d$,*

$$|v^\top \nabla^2 f(x)\, v| \;\leq\; L\|v\|^2.$$

*Then $f$ has an $L$-Lipschitz continuous gradient (i.e., $f$ is $L$-smooth); in particular,*

$$\|\nabla f(y) - \nabla f(x)\| \;\leq\; L\|y - x\|,$$

*and*

$$f(y) \;\geq\; f(x) + \langle \nabla f(x), y - x\rangle - \tfrac{L}{2}\|y - x\|^2$$

*for all $x, y \in \mathbb{R}^d$.*

**Lemma I.2.** *Consider any two policies $\pi_i$, $i = 1, 2$. It holds that*

$$\left\| d_\rho^{\pi_1} - d_\rho^{\pi_2} \right\|_1 \leq \frac{\gamma}{1-\gamma} \sup_{s\in\mathcal{S}} \|\pi_1(\cdot|s) - \pi_2(\cdot|s)\|_1 \;.$$

*Proof.* Let us start from the definition of flow conservation constraints for the discounted state visitation (Puterman, 1994), for $i \in \{1,2\}$, we have

$$d_\rho^{\pi_i}(s) = (1-\gamma)\rho(s) + \gamma \sum_{s'} \mathsf{P}_{\pi_i}(s|s') d_\rho^{\pi_i}(s') \;.$$

Then, we have

$$\sum_{s\in\mathcal{S}} |d_\rho^{\pi_2}(s) - d_\rho^{\pi_1}(s)| \leq \gamma \sum_{(s',a')} \sum_s \left| \mathsf{P}(s|s',a')\pi_2(a'|s')d_\rho^{\pi_2}(s') - \mathsf{P}(s|s',a')\pi_1(a'|s')d_\rho^{\pi_1}(s') \right|$$

$$\leq \gamma \sum_{s',a'} \sum_s \mathsf{P}(s|s',a')\,|\pi_2(a'|s') - \pi_1(a'|s')|\,d_\rho^{\pi_2}(s')$$

$$+ \gamma \sum_{s',a'} \sum_s \mathsf{P}(s|s',a')\pi_1(a'|s') \left| d_\rho^{\pi_1}(s') - d_\rho^{\pi_2}(s') \right|$$

$$\leq \gamma \sup_{s\in\mathcal{S}} \|\pi_1(\cdot|s) - \pi_2(\cdot|s)\|_1 + \gamma \sum_{s'} |d_\rho^{\pi_1}(s') - d_\rho^{\pi_2}(s')| \;,$$

which concludes the proof. $\qquad\square$

**Lemma I.3** (Performance Difference Lemma). *It holds that*

$$v_\star^f(\rho) - v_\theta^f(\rho) = \frac{1}{1-\gamma} \sum_{s\in\mathcal{S}} d_\rho^{\pi_\star^f}(s) \left[ \sum_{a\in\mathcal{A}} \pi_\star^f(a|s) q_\theta^f(s,a) - \lambda\, \mathrm{D}^f(\pi_\star^f(\cdot|s)\|\pi_{\text{ref}}(\cdot|s)) - v_{\pi_\theta}^f(s) \right].$$

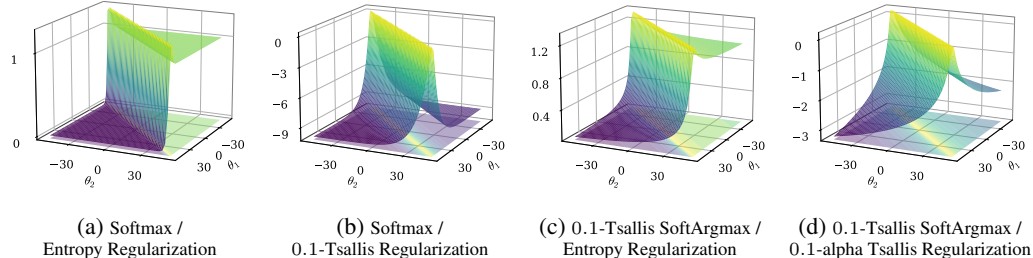

(a) Softmax / Entropy Regularization

(b) Softmax / 0.1-Tsallis Regularization

(c) 0.1-Tsallis SoftArgmax / Entropy Regularization

(d) 0.1-Tsallis SoftArgmax / 0.1-alpha Tsallis Regularization

Figure 3: Regularized value landscapes for a one-state, two-action MDP (rewards $0, 1$) for different coupling between parameterizations and regularizations.

*Proof.* Fix $\theta \in \mathbb{R}^{|\mathcal{S}||\mathcal{A}|}$ and any state $s \in \mathcal{S}$. It holds that

$$
\begin{aligned}
v_\star^f(s) - v_{\pi_\theta}^f(s) &= \sum_{a \in \mathcal{A}} \pi_\star^f(a|s) q_\star^f(s, a) - \sum_{a \in \mathcal{A}} \pi_\theta^f(a|s) q_\theta^f(s, a) \\
&\quad - \lambda \mathrm{D}^f(\pi_\star^f(\cdot|s)\|\pi_{\mathrm{ref}}(\cdot|s)) + \lambda \mathrm{D}^f(\pi_\theta^f(\cdot|s)\|\pi_{\mathrm{ref}}(\cdot|s)) \\
&= \sum_{a \in \mathcal{A}} \pi_\star^f(a|s) \left( q_\star^f(s, a) - q_\theta^f(s, a) \right) + \sum_{a \in \mathcal{A}} \left( \pi_\star^f(a|s) - \pi_\theta^f(a|s) \right) q_\theta^f(s, a) \\
&\quad - \lambda \mathrm{D}^f(\pi_\star^f(\cdot|s)\|\pi_{\mathrm{ref}}(\cdot|s)) + \lambda \mathrm{D}^f(\pi_\theta^f(\cdot|s)\|\pi_{\mathrm{ref}}(\cdot|s)) \\
&= \gamma \sum_{a \in \mathcal{A}} \pi_\star^f(a|s) \sum_{s' \in \mathcal{S}} \mathrm{P}(s'|s, a) \left( v_\star^f(s') - v_\theta^f(s') \right) + \sum_{a \in \mathcal{A}} \left( \pi_\star^f(a|s) - \pi_\theta^f(a|s) \right) q_\theta^f(s, a) \\
&\quad - \lambda \mathrm{D}^f(\pi_\star^f(\cdot|s)\|\pi_{\mathrm{ref}}(\cdot|s)) + \lambda \mathrm{D}^f(\pi_\theta^f(\cdot|s)\|\pi_{\mathrm{ref}}(\cdot|s)) \ ,
\end{aligned}
$$

where in the last equality, we used the definition of the regularized Q-function (2). Expanding the recursion yields

$$
\begin{aligned}
v_\star^f(s) - v_{\pi_\theta}^f(s) &= \frac{1}{1-\gamma} \sum_{s' \in \mathcal{S}} d_s^\star(s') \left[ \sum_{a \in \mathcal{A}} \left( \pi_\star^f(a|s') - \pi_\theta^f(a|s') \right) q_\theta^f(s', a) \right] \\
&\quad + \frac{1}{1-\gamma} \sum_{s' \in \mathcal{S}} d_s^\star(s') \left[ \lambda \mathrm{D}^f(\pi_\theta^f(\cdot|s')\|\pi_{\mathrm{ref}}(\cdot|s')) - \lambda \mathrm{D}^f(\pi_\star^f(\cdot|s')\|\pi_{\mathrm{ref}}(\cdot|s')) \right] \\
&= \frac{1}{1-\gamma} \sum_{s' \in \mathcal{S}} d_s^\star(s') \left[ \sum_{a \in \mathcal{A}} \pi_\star^f(a|s') q_\theta^f(s', a) - \lambda \mathrm{D}^f(\pi_\star^f(\cdot|s')\|\pi_{\mathrm{ref}}(\cdot|s')) \right] \\
&\quad - \frac{1}{1-\gamma} \sum_{s' \in \mathcal{S}} d_s^\star(s') \left[ \sum_{a \in \mathcal{A}} \pi_\theta^f(a|s') q_\theta^f(s', a) - \lambda \mathrm{D}^f(\pi_\theta^f(\cdot|s')\|\pi_{\mathrm{ref}}(\cdot|s')) \right] \ ,
\end{aligned}
$$

which concludes the proof. $\qquad \square$

**Lemma I.4** (Lemma 23 of Mei et al. (2020b) ). *Let $\pi \in \mathcal{P}(\mathcal{A})$. Denote $H(\pi) = \mathrm{diag}(\pi) - \pi\pi^\top$. For any vector $x \in \mathbb{R}^{|\mathcal{A}|}$*

$$
\left\| H(\pi) \left( x - \frac{\langle x, \mathbb{1}_{|\mathcal{A}|}\rangle}{|\mathcal{A}|} \mathbb{1}_{|\mathcal{A}|} \right) \right\|_2 \geq \min_{a \in \mathcal{A}} \pi(a) \cdot \left\| x - \frac{\langle x, \mathbb{1}_{|\mathcal{A}|}\rangle}{|\mathcal{A}|} \mathbb{1}_{|\mathcal{A}|} \right\|_2 \ .
$$

**Lemma I.5** (Danskin 1966). *Let $Z \subset \mathbb{R}^m$ be compact and let $\phi : \mathbb{R}^n \times Z \to \mathbb{R}$ be continuous. Define*

$$
f(x) = \max_{z \in Z} \phi(x, z), \qquad Z_0(x) = \arg\max_{z \in Z} \phi(x, z).
$$

*Assume that for each fixed $z \in Z$, the map $x \mapsto \phi(x, z)$ is differentiable. If $Z_0(x) = \{\bar{z}\}$ and $x \mapsto \phi(x, \bar{z})$ is differentiable at $x$, then $f$ is differentiable at $x$ with*

$$
\frac{\partial f(x)}{\partial x} = \frac{\partial \phi(x, \bar{z})}{\partial x}.
$$

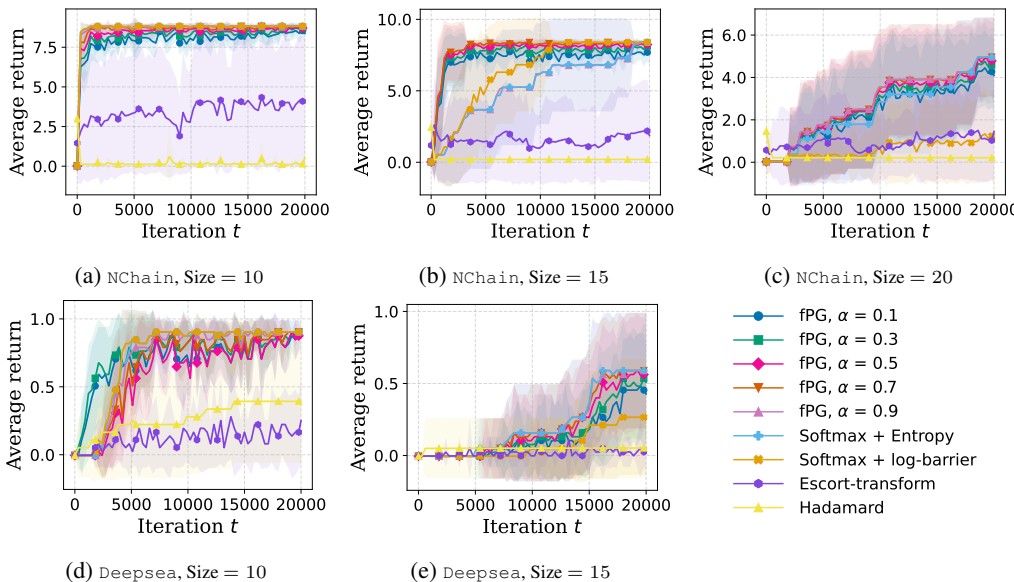

Figure 4: Average return as a function of training iterations on `NChain` (top row: sizes 10, 15, 20) and `DeepSea` (bottom row: sizes 10, 15). For $f$-`PG`, we report the best configuration for each divergence parameter $\alpha$; for all other methods, we report the best-performing configuration over their respective hyperparameters. Curves show the mean performance over 15 independent seeds, with shaded regions indicating one standard deviation.

## J    EXPERIMENTS

### J.1    UNCOUPLING THE PARAMETERIZATION AND THE REGULARIZATION

Figure 3 compares the regularized value landscapes induced by different couplings between policy parameterizations and regularizers. Both Softmax / Entropy Regularization (Figure 3a) and Softmax / 0.1-Tsallis Regularization (Figure 3b) produce highly ill–conditioned objectives, characterized by wide flat plateaus separated by extremely sharp ridges. These geometries create large regions with vanishing gradients together with nearly singular directions, which are known to slow down and destabilize policy–gradient methods. Switching to the Tsallis SoftArgmax parameterization already improves the situation: under Entropy Regularization (Figure 3c), the flat directions are reduced and the basin around the optimum becomes more pronounced. However, the most favorable geometry is obtained when Tsallis SoftArgmax is coupled with Tsallis regularization (Figure 3d). In this matched Tsallis–Tsallis regime, the landscape becomes smooth, strongly curved, and well–conditioned, with a single broad basin leading to the optimum and no spurious flat regions or steep barriers. This alignment between the geometry induced by the parameterization and that of the regularizer yields an almost quadratic objective in logits, explaining why the Tsallis–Tsallis coupling provides the best convergence behavior.

### J.2    TABULAR EXPERIMENTS

We evaluate the empirical performance of $f$-`PG` equipped with $\alpha$-Tsallis regularization, with the goal of assessing how our coupled parameterization–regularization framework compares to the baselines summarized in Table 2. All methods are evaluated on the *unregularized* return objective, and we report learning curves as a function of training iterations. For each value of the Tsallis parameter $\alpha$, we tune both the temperature parameter $\lambda$ and the step-size $\eta$ over the grid

$$\lambda \in \{10^{-3}, 10^{-2}, 10^{-1}, 1.0\}, \qquad \eta \in \{10^{-4}, 3 \times 10^{-4}, 10^{-3}\}.$$

For the baseline methods, we analogously select the best-performing configuration over their respective hyperparameter grids. All curves are averaged over 15 independent random seeds, and shaded regions indicate one standard deviation.

**NChain and DeepSea.**    We consider two canonical tabular exploration benchmarks. `NChain` is a long-horizon chain environment in which the agent must repeatedly move in one direction to reach a terminal state with a large $+1$ reward, while a small immediate reward equal to $+0.01$ is available for moving in the opposite direction. As the chain length increases, the probability of discovering the optimal policy decays exponentially unless sufficient structured exploration is induced. The `DeepSea` environment, described in Section 5, is a two-dimensional sparse-reward navigation task in which the agent must follow a precise sequence of actions to reach a distant rewarding state. Both environments therefore test the ability of a policy-gradient method to propagate credit over long horizons and through sparse feedback.

**Results.**    Figure 4 reports learning curves on both environments. On `NChain` (top row, Figures 4a to 4c), the standard softmax–entropy policy gradient baseline performs competitively for the smallest instance (Size 10), but its performance degrades markedly as the chain length increases. In particular, for Size 15 a substantial performance gap opens up, and for Size 20 softmax converges slowly and remains far from optimal. In contrast, $f$-`PG` with $\alpha < 1$ achieves substantially higher returns and converges much faster for intermediate horizons, most notably for Size 1,5 where a clear performance gap with the softmax entropy-regularized policy gradient emerges. For the longest chain (Size 20), Tsallis regularization continues to outperform the other baselines but exhibits a similarly slow convergence trend to softmax–entropy, reflecting the difficulty of the problem at this scale. Moreover, alternative regularization schemes fail completely on the longest `NChain` of size 20. Additionally, Escort policy gradients and Hadamard parameterizations plateau at very low returns across all chain lengths, indicating the need for additional exploration.

A similar pattern is observed in `DeepSea` (bottom row, Figures 4d and 4e). For Size 10, Tsallis-regularized policies perform on par with the softmax–entropy baseline and in some cases converge faster. For the more challenging Size 15 instance, Tsallis policies with $\alpha < 1$ remain competitive. In this environment, $f$-`PG` dominates all other baselines, which fail to make meaningful progress toward high-return policies.

Although no single Tsallis parameter $\alpha < 1$ is uniformly optimal across all problem sizes, a clear and robust trend emerges across both `NChain` and `DeepSea`: *for every environment, there exists an $\alpha < 1$ that strictly outperforms all alternative schemes and matches or exceeds the performance of softmax–entropy*. These results corroborate the observations of Section 5 and confirm that jointly tailoring the policy parameterization and the regularization to the structure of the problem leads to improved empirical performance.

### J.3    FULL DESCRIPTION OF THE DEEPRL EXPERIMENTS

We provide here a full description of the $\alpha$-`Tsallis` `PPO` algorithm and the experimental setup of Section 5.

**Algorithm description.**    $\alpha$-`Tsallis` `PPO` follows the same overall algorithmic structure as standard `PPO`, with only minor modifications: the softmax policy parameterization and entropy regularization are replaced by their Tsallis $\alpha$-softargmax and Tsallis $\alpha$-regularization counterparts. Specifically, the policy network outputs unnormalised action logits, which are mapped to a probability distribution via the Tsallis $\alpha$-softargmax instead of a usual softmax (see Section 3). At each interaction step, we compute the Tsallis divergence between the current policy and the uniform distribution over actions using the same coefficient $\alpha$, and subtract $\lambda \cdot \mathrm{D}^{f_\alpha}(\pi(\cdot|s)\|\mathrm{uniform})$ (see Table 2 for the expression of $f_\alpha$) from the received reward.[2] All other components, including the clipped surrogate objective, value loss, and advantage normalization, remain unchanged.

**Training pipeline.**    At each update, we collect trajectories from 16 parallel environments for 32 steps, followed by 16 epochs of PPO optimisation over 4 minibatches. We use a discounting factor

---

[2]This regularized reward is used in the computation of advantages and value targets for the `DeepSea` environment. The reason is that adding entropy to the rewards turns out to be critical for the method's final performance, since the original softmax-PPO fails even in `DeepSea` of size 20. Notice that it contrasts with an original PPO that adds entropy regularization only to a loss function and not to a reward. For `Noisy Carpole`, we use a standard implementation of PPO as a baseline.

$\gamma = 0.99$ and GAE $\lambda = 0.95$, and apply gradient clipping at norm $0.5$. Both actor and critic are two-layer multilayer perceptrons with $64$ hidden units and `tanh` activations. Optimization is performed using `Adam` (Kingma, 2014), and we perform a grid search over

$$\lambda \in \{10^{-3},\, 10^{-2},\, 10^{-1},\, 1.0\}, \quad \eta \in \{10^{-4},\, 3 \times 10^{-4},\, 10^{-3}\}.$$

Each configuration is evaluated across $25$ seeds. Episode returns are aggregated per configuration and reported as mean $\pm$ standard error. For each $\alpha$, we select the configuration that produces the highest last reward on average.

