# OpenReview forum: "Beyond Softmax and Entropy: Convergence Rates of Policy Gradients with $\boldsymbol{f}$-SoftArgmax Parameterization $\&$ Coupled Regularization"
_ICLR.cc/2026/Conference — ICLR 2026 Poster_

### Official Review · Reviewer_ri98 · 2025-10-25

**Soundness:** 2
**Presentation:** 2
**Contribution:** 2
**Rating:** 4
**Confidence:** 2

**Summary:**

The authors propose a general f-regularized policy gradient method with coupled parametrization, which generalizes the popular KL regularization with softmax parametrization. Theoretically, they demonstrate global convergence and show an improved trade-off between sample complexity and regularization bias compared to the classical entropy-softmax combination.

**Strengths:**

The authors propose a general framework for studying policy gradient methods in f-divergence-regularized RL with coupled parametrization, providing strong theoretical guarantees.

**Weaknesses:**

- The experiments are conducted on a 5×5 GridWorld, which limits the generalizability of the proposed policy gradient method.
- Apart from the theoretical contributions, which I do not feel qualified to fully assess, it is unclear to me what practical advantage the proposed algorithm offers compared to using KL regularization with softmax parametrization.

**Questions:**

- What are the main factors limiting the proposed algorithm from being evaluated on a broader set of environments, such as Atari?
- How does this approach compare to prior work [1] showing that Mirror Descent, with different choices of mirror map and optimization space (e.g., logits or policy), can lead to novel regularized policy gradient objectives?

[1] Vaswani, S., Bachem, O., Totaro, S., Müller, R., Garg, S., Geist, M., Machado, M.C., Castro, P.S. and Roux, N.L., 2021. A general class of surrogate functions for stable and efficient reinforcement learning. arXiv preprint arXiv:2108.05828.

---

> ### Author Response · Authors · 2025-11-21
>
> We thank the reviewer for their careful reading and are pleased that they highlighted our "strong theoretical guarantees", which are precisely the focus of our work.
>
> Below, we address the concerns and questions raised by the reviewer.
>
> > **``The experiments are conducted on a $5\times5$ GridWorld, which limits the generalizability of the proposed policy gradient method.''**, **``...it is unclear to me what practical advantage the proposed algorithm offers compared to using KL regularization with softmax parametrization.''**.
>
> Thank you very much for these remarks. In the revised version of the papers, we extend the experimental section and, instead of GridWorld, we use a more well-established exploration benchmark, DeepSea, of different sizes. In particular, we observe that on a task that does not require deep exploration (i.e., when size is equal to 5), the standard softmax-based method outperforms Tsallis-based methods. However, when the exploration component of the problem becomes more critical, Tsallis performs better and discovers the hidden reward more effectively. We connect this phenomenon to the fact that a corresponding parametrization propagates a signal about suboptimal reward less actively than a softmax policy, allowing the agent to spend more time finding a bigger reward before starting exploitation.
>
>
> > **``How does this approach compare to prior work [1] showing that Mirror Descent, with different choices of mirror map and optimization space (e.g., logits or policy), can lead to novel regularized policy gradient objectives?''**
>
> We thank the reviewer for bringing this prior work to our attention.  Indeed, the work [1] studies the choice of general divergence in the MDPO-style method and experimentally verifies it for a KL setting.
>
> Although the main distinction between our work and [1] is that *they study only direct and softmax parameterizations* and plug-in them to derive novel surrogate policy-improvement losses. In our work, the main focus is on altering the parameterization within a standard policy gradient method and providing a comprehensive theoretical analysis of its influence.
>
> > **``What are the main factors limiting the proposed algorithm from being evaluated on a broader set of environments, such as Atari?''**
>
> The main limiting factor is that our algorithm is formally defined for tabular environments. While it is definitely possible to just plug-in our new parameterization into any existing method, such as PPO, we note that, to achieve actual improvement, it is necessary to adapt existing surrogate losses to the new parameterization, following common practice (see, e.g., [1]).
> This is a very promising direction for future research, although we would like to emphasize that this is outside the scope of the current work, and our primary focus is on establishing theoretical guarantees.
>
> **Thank you again for your careful review, and please do not hesitate to let us know if you have any further questions.**

---

> > ### Comment · Reviewer_ri98 · 2025-11-25
> >
> > Thank you for your responses. What is the rationale for fixing the step size to 0.1 in the DeepSea experiments? I expect the choice of step size to substantially influence the conclusions.
> >
> > While the theoretical development is interesting and may motivate new algorithmic ideas, it is difficult to extrapolate from tabular settings to more complex domains such as Atari or MuJoCo without careful empirical study and a clear step by step description of how the method would need to be adapted. In its current form, the work’s impact feels limited.

---

> ### Author Response · Authors · 2025-12-02
>
> We thank the reviewer for the valuable feedback.
>
> To address the reviewer’s concern regarding practical relevance beyond the tabular setting, we have
> *added deep RL experiments* on the JAX $\texttt{DeepSea}$ environments (sizes $10$, $20$, $30$, $40$, and $50$).  In these experiments, we do *not* apply the tabular f-PG algorithm.  Instead, following the discussion in our paper on extending the coupled $f$-softargmax parametrization to deep RL (see Section~6), we implement the *PPO extension that we propose*, obtained by replacing the classical softmax policy with its $f$-softargmax counterpart while keeping the standard PPO surrogate. This forms a direct deep-RL instantiation of our coupled parametrization framework.
> In addition, for these deep RL experiments we conduct a *full hyperparameter sweep* to ensure that the observed performance is not an artifact of a specific configuration.
>
> Our results show that the qualitative advantages highlighted in the tabular experiments, notably the effect of the coupled parametrization on exploration and stability, *persist in the deep RL setting*.
> These new experiments are included in the revised appendix and strengthen the empirical support for our method.

---

### Official Review · Reviewer_zseX · 2025-11-02

**Soundness:** 3
**Presentation:** 3
**Contribution:** 3
**Rating:** 6
**Confidence:** 3

**Summary:**

This paper proposes generalization of entropy regularized policy gradient methods using f-divergences, as well as corresponding parameterizations.

The authors shows global convergence results the proposed methods, as well as experimental results to verify the theoretical findings.

**Strengths:**

1. Generalizing the entropy regularizer to f-divergences is a reasonable idea.
2. The projection operator to avoid deterministic policies is useful.

**Weaknesses:**

1. The work seems to be generalizing and recovering existing work (entropy, Tsallis). It seems this work is not suggesting some new regularization which is better performing than existing ones.

**Questions:**

1. I am wondering how the methods perform when the interest is to obtain the unregularized optimal value/policy. I can imagine the operator to avoid small probability could fail (since it is unavoidable to get close to deterministic policies in that case). Does your method provide better iteration/sample complexity?

2. The authors mentioned that entropy and Tsallis, which are two important existing regularization, can be recovered by the f-divergences. Are there any new regularization which can perform better than existing ones (or the other way those two are the best possible)?

3. It is claimed that for the PL inequality, "our proof, based on the properties of Fenchel-Legendre conjugation, is much simpler". After checking it seems to me the key ideas of smoothness and PL inequality proofs are largely similar to existing proofs in entropy regularized proofs, especially the lower bounding of policy gradient using Eq. (52) and $H(w_\theta)$, and upper bounding the suboptimality gap by Eq. (51). Could you elaborate how Fenchel-Legendre conjugation makes your proofs much simpler (my understanding is that this helps proving upper bounding suboptimality gap in general f-divergences)?

---

> ### Author Response · Authors · 2025-11-21
>
> We thank the reviewer for their careful reading of our paper and for the positive assessment of its soundness, presentation, and contribution. We are particularly grateful that the reviewer highlights both that (i) "the generalization from entropy regularization to $f$-divergence regularization as a reasonable idea", and that (ii) the "projection operator to avoid deterministic policies is useful".
>
> Below we address the reviewer’s questions in detail.
>
> > **"I am wondering how the methods perform when the interest is to obtain the unregularized optimal value/policy. I can imagine the operator to avoid small probability could fail (since it is unavoidable to get close to deterministic policies in that case). Does your method provide better iteration/sample complexity?"**
>
> This question goes to the core motivation of our work. Indeed, much of our analysis is aimed precisely at understanding how different regularizations affect the ability of policy gradient methods to efficiently approximate the **unregularized** optimal policy since it is a natural benchmark to comparing different regularizations.
>
> Corollaries 5.7 and 5.8 precisely analyze this regime: they provide guarantees on the quality of the **solution to the unregularized problem** obtained by choosing the regularization power appropriately. Furthermore, Corollary 5.9 shows that in the high-accuracy regime, when the goal is to approximate the unregularized optimum to tolerance $\varepsilon$, the theoretically optimal choice is to take $\alpha$ closer to $0$. This leads to improved dependence on $\varepsilon$ compared to entropy regularization.
>
> Following your remark, we decided to extend the discussion on these corollaries in the revised version of the manuscript, to make the implications of our result on the unregularized problem as clear as possible. Thank you again for this remark!
>
> Regarding the concern about clipping: in the high-accuracy regime, we set $\lambda = O(\varepsilon)$, and the clipping threshold $\tau$ is chosen to satisfy $\tau = O([f']^{-1}(-1/\epsilon))$. As $\varepsilon \to 0$, the clipping threshold (under our assumption $A_f(\pi_{ref})$) goes to zero, meaning that the projection becomes increasingly unlikely to be active. Thus, when we seek a highly accurate approximation of the unregularized optimal policy, clipping effectively disappears and does not interfere with approaching near-deterministic optimal policies. We will also clarify this intuition more explicitly in the revised version.
>
>
> > **"The authors mentioned that entropy and Tsallis, which are two important existing regularization, can be recovered by the f-divergences. Are there any new regularization which can perform better than existing ones (or the other way those two are the best possible)?"**
>
> Our theory covers a broad class of divergences, including among others divergences that interpolate between KL and Tsallis.
> However, we decided to focus our study on the widely used entropy and Tsallis divergence, which are the most relevant in practice, and for which we provide a theoretical analysis.
> From a theoretical standpoint, we note that finding "best" parameterization is challenging, as suggested in Corollary 5.9, where the optimal choice of divergence, and even the optimal $\alpha$ within the Tsallis family, depends non-trivially on the desired accuracy $\varepsilon$. Thus, selecting an optimal divergence in all regimes appears to be a difficult problem which could be studied in future work.
>
> That being said, there are other examples that may be promising in practice, and that are covered in our Theorem 5.4  (although we have not evaluated them empirically), for instance:
> - Jensen-Shannon divergence, induced by $f(u) = u \log u - (u+1) \log ((u+1)/2)$;
> - Squared Hellinger distance, induced by $f(u) = (\sqrt{u}-1)^2$.
> These are appealing because of their symmetry, in contrast to forward or reverse KL divergences, and constitute promising research directions.
>
> Still, we would like to emphasize that almost all practically used divergence examples are exactly Tsallis entropy with different parameters of $\alpha$, including $\alpha \to 0$, which recovers a reverse KL-divergence and $\alpha \to 1$, which recovers a usual KL-divergence. The only well-known exception is a Jensen-Shannon divergence that appears naturally in Generative Adversarial Networks (Goodfellow et al. 2014).
>
>
> Goodfellow, Ian J., et al. "Generative adversarial nets." Advances in neural information processing systems 27 (2014).

---

> > ### Author Response · Authors · 2025-11-21
> >
> > > **"It is claimed that for the PL inequality, "our proof, based on the properties of Fenchel-Legendre conjugation, is much simpler". After checking it seems to me the key ideas of smoothness and PL inequality proofs are largely similar to existing proofs in entropy regularized proofs, especially the lower bounding of policy gradient using Eq. (52) and $H(w_\theta)$, and upper bounding the suboptimality gap by Eq. (51). Could you elaborate how Fenchel-Legendre conjugation makes your proofs much simpler (my understanding is that this helps proving upper bounding suboptimality gap in general f-divergences)?"**
> >
> >
> > We agree with the reviewer that the overall proof structure, particularly the step involving lower-bounding the norm of the gradient of the regularized value function, is conceptually similar to existing analyses for entropy-regularized policy gradient methods. The simplification from Fenchel–Legendre conjugation appears in the step of **upper-bounding the soft-suboptimality gap (Lemma C.4)**.
> >
> > To highlight the contrast, consider the analysis of Mei et al. (2020). Their proof takes a substantially longer detour by introducing, in Lemma 26, a soft sub-optimality lemma. This lemma is highly specific to the entropy-regularized value function and cannot be extended to general
> > $f$-divergences, because it relies in an essential way on the logarithm's special properties. As a result, we would call this argument ad hoc and specialized only for entropy regularization rather than a structural one. Concretely, the lemma rewrites the soft sub-optimality gap
> > $v_{\star}(s) - v_{\pi}(s)$ as $\frac{\lambda}{1-\gamma} \sum_{s'\in \mathcal{S}} d_{s}^{\pi}(s') D^{\mathrm{KL}}(\pi(\cdot|s')||\pi_{\star}^{KL}(\cdot||s))$,
> > after which several additional bounds are required to control the KL term.
> >
> > In contrast, our proof proceeds more directly. We bound the soft sub-optimality gap (Eq. 56) by a quantity
> > $B(s)$ that is upper-bounded by the first-order Taylor expansion of the function $\mathrm{softmax}^{f}$ between $\theta$ and $q_{\theta}^f$. This term is a Taylor expansion only because the $\mathrm{softargmax}^{f}$ policy is the gradient of the $\mathrm{softmax}^{f}$ (which is tightly connected to Fenchel–Legendre conjugation properties).
> >
> > We have expanded the discussion in the revised manuscript to make this comparison clearer, see a newly introduced discussion after Theorem 4.3.
> >
> > **Thank you again for your comments and questions**.

---

### Official Review · Reviewer_s4Jk · 2025-11-03

**Soundness:** 2
**Presentation:** 2
**Contribution:** 2
**Rating:** 4
**Confidence:** 3

**Summary:**

This paper extends the entropy regularized softmax PG in Mei et al. 2020b to the a general regularization based on the coupled parameterization. Convergence guarantees are also established.

**Strengths:**

Extension of entropy regularized softmax PG.

**Weaknesses:**

* Presentation can be improved. In particular, more discussion of the theoretical results will be helpful for the reader to get a better understanding since there are quite a lot of parameters involved.
* More numerical experiments can be conducted, even though the authors argue that the goal is to verify the effectiveness of Tsallis divergence. It is known that (entropy regularized) softmax PG is highly inefficient compared to (entropy regularized) NPG due to that the appearance of the policy in the exponential term may cause a mislead. I am wondering whether the new algorithm based on Tsallis divergence will be efficient than NPG or not. At least, tests for the exact setting can be conducted.
* References are missing. For example in "Elementary analysis of policy gradient methods" by Liu et al 2024, it is shown that softmax PG (without regularization) can achieve sublinear convergence for ANY constant step size though the problem dependent constant still exists.

**Questions:**

* In Mei et al. 2020b, there are exists sublinear convergence result for the non-regularized softmax PG. As far as I can see, the authors only extend the regularized counterpart. Is it right? Even though the sample complexity can be obtained for the non-reguarlized case by choosing the parameter carefully, the sublinear convergence result in the exact setting for the non-regularized case cannot be obtained from the linear result for the regularized case. Isn't it?
* A major theoretical contribution claimed by the authors is that there is no problem dependent hidden in the convergence rate. Is it fully due to the projection operator or also relies the the particular divergence? Will it also be helpful for removing the constant in the sublinear convergence of the non-regularized softmax PG?

---

> ### Author Response · Authors · 2025-11-21
>
> We thank the reviewer for their time and effort for a careful reading of the manuscript. Below, we address all the raised concerns.
>
> >**" In particular, more discussion of the theoretical results will be helpful for the reader to get a better understanding [...]."**
>
> Thanks for the suggestion. We will use the additional page to incorporate the following changes, which we believe will significantly improve the clarity of our results:
>
> - We state after Theorem 4.3 that our result generalises the results of Mei et al. 2020, and give details on the technical contribution required for this generalization.
> - We comment on the result of Corollary 5.5, making explicit the different regimes where we obtain $O(\log(1/\epsilon)$ and $O(1/\epsilon \log(1/\epsilon)$ rates.
> - We give context for Corollary 5.6, which provides a rate for *solving the un-regularized problem* using our method with explicit rates, and discuss the influence of the choice of $f$-divergence and of the desired precision level on this rate.
> - After Corollary 5.8, we give more precision on comparison with existing algorithms, specifically stating that **our result is the first to derive polynomial sample complexity using only first-order information and without preconditioning**.
> We hope these additions help clarify our theoretical contributions. We are happy to further elaborate should the reviewer still find the discussion insufficient.
>
> >**"More numerical experiments can be conducted, even though the authors argue that the goal is to verify the effectiveness of Tsallis divergence. It is known that (entropy regularized) softmax PG is highly inefficient compared to (entropy regularized) NPG[...]. I am wondering whether the new algorithm based on Tsallis divergence will be efficient than NPG or not. At least, tests for the exact setting can be conducted."**
>
> Thank you for pointing this out! We fully agree that comparing these approaches is highly insightful, and we have thus **included such a comparison in the revised version of the paper**. In the updated experimental section, the new parameterization outperforms the classical softmax parameterization, and even surpasses NPG in some settings.
>
> Nonetheless, as the reviewer correctly noted, Natural Policy Gradient (NPG) methods can be highly efficient. However, we emphasize that our main objective here is **to study first-order methods without preconditioning**, that is, methods that do not rely on any information from the Hessian, Fisher information matrix, or Jacobian of the parameterization. Formally, we show that this method can guarantee polynomial sample complexity, demonstrating that the exponential convergence times often observed in standard policy gradient (PG) methods can be mitigated through a more principled choice of parameterization and regularization, rather than with preconditioning or second-order methods.
>
> We have extended the discussions on this point throughout the manuscript and in the experimental section, mentioning explicitly NPG and the key differences with our work.
>
> >**"References are missing. For example in "Elementary analysis of policy gradient methods" by Liu et al 2024, it is shown that softmax PG (without regularization) can achieve sublinear convergence for ANY constant step size though the problem dependent constant still exists."**
>
> Thank you very much for the reference, we have incorporated it into the revised version of the related work section.
>
> >**"In Mei et al. 2020b, there are exists sublinear convergence result for the non-regularized softmax PG. The authors only extend the regularized counterpart. Is it right? Even though the sample complexity can be obtained for the non-reguarlized case by choosing the parameter carefully, the sublinear convergence result in the exact setting for the non-regularized case cannot be obtained from the linear result for the regularized case. Isn't it?"**
>
> Thank you very much for raising this question. Indeed, your observation is completely correct. The main contribution of our work is to introduce novel parameterizations that are specifically adapted to the choice of an $f$-divergence–based regularization (hence the term *"coupled"*). This naturally requires studying regularized problems, which justifies our focus on the regularized setting. Moreover, in practice, a small entropy term is almost always added to promote exploration, further supporting the study of the regularized problem.
> We also emphasize that our analysis (see Corollaries 5.6-5.9) explicitly characterizes the sample-complexity/regularization tradeoff, which in particular allows to **solve an unregularized problem by using a small regularization coefficient**.
>
> That being said, we fully acknowledge that providing convergence guarantees directly for unregularized problem is a very important and challenging question. Investigating the properties of our novel parameterization in this setting would be an interesting direction for future work.

---

> > ### Author Response · Authors · 2025-11-21
> >
> > > **"A major theoretical contribution claimed by the authors is that there is no problem dependent hidden in the convergence rate. Is it fully due to the projection operator or also relies the particular divergence? Will it also be helpful for removing the constant in the sublinear convergence of the non-regularized softmax PG?"**
> >
> > The main reason for avoiding problem-dependent constants in our convergence rates is indeed *the use of a projection operator*, which applies for all the parameterizations we consider, including softmax. This projection operator can only be used when the problem is regularized, because only in this case is the **optimal parameter** $\theta^\star$ **finite** and has a **known bound on its norm**.
> >
> > In contrast, for the *unregularized policy gradient*, the optimal parameter tends to *diverge to infinity* under the softmax parameterization, as the policy converges to a **deterministic limit**. As a result, and in stark contrast with the regularized setting, it is **impossible to enforce a positive lower bound** on the minimal policy weight.
> >
> > **Thank you very much for your comments and questions. We hope we answered your concerns, and remain available should you have additional questions.**

---

### Official Review · Reviewer_q88P · 2025-11-08

**Soundness:** 3
**Presentation:** 2
**Contribution:** 1
**Rating:** 0
**Confidence:** 4

**Summary:**

This submission derives fast rates (i.e., $\tilde{O}(\epsilon^{-1})$) for learning against the $f$-divergence regularized objective in discounted MDPs, which is achieved in a batch-online manner. In the algorithm design, the *log-linear policy with a reference model $\pi^\text{ref}$* under KL-regularization is extended to the so-called "coupled parametrization" in this submission. Finally, the submission claims a separation result between Tsallis-entropy regularization and entropy regularization by comparing *two upper bounds*.

**Strengths:**

- This is the first work deriving fast rates $\tilde{O}(\epsilon^{-1})$ for learning w.r.t. divergence-regularized objectives in the discounted settting.
- The proofs are correct.

**Weaknesses:**

> **To AC: the reviewer is willing and ready to further discuss about this submission with AC or even SAC if needed.**

- Every time $[f']^{-1}$ appears in this submission, it is **mathematically wrong**: because, for example, chi-square divergence is already nice enough, but the $a \mapsto \max\{0, a\}$ (also appears as ReLU in the lierature) in the Lemma G.2 of [10] already certificates that it is not possible to simply subsume the solution to this constrained optimization problem using $[f']^{-1}(\cdot)$ and the complementary slackness condition might need to be tackled in a case-by-case way.

- The smoothness property (i.e., the Hessian spectrum upper bound) is trivial and does not even deserve a theorem, i.e., Theorem 4.3, because it is well known that the Fenchel conjugate of a $\alpha$-strongly-convex function is $\alpha^{-1}$-smooth (under very mild qualitative regularity conditions); which is nearly exactly the case for the regularized objective with equation (9) as the parametrization
- The artificial assumptions from Line 201 to Line 210 are far from enlighting and even **known to be** highly unnecessary for important cases like reverse-KL regularization and chi-square divergence regularization:
  1. to be concrete, the reviewer **strongly disagrees** with Line 215-216 (the sentence around "which prevents") because the analysis of learning w.r.t. the reverse-KL-regularized objectives have become sharp (in terms of the dependency on $\epsilon$, i.e., $\tilde{O}(\epsilon^{-1})$) for contextual bandits and episodic MDPs in the hybrid setting [6, 1], for contextual bandits in the offline setting [9,7], and for both contextual bandits and episodic MDPs in the online setting [8].
  2. Even without Assumption P, and even in the pure offline setting without exploration, the analysis of learning against many divergence regularized objectives is doable, as manifested in the [7]
- The term "coupled parametrization" is confusing because it is just an extension of the "log-linear policy *with a reference policy*" to the general $f$-divergence setting, see, e.g., Definition 1.1 in [1]
- The PL condition, i.e., the "essentially strongly concave" property of the regularized objective **should not appear as a brandly new contribution** in this submission because it has been presented in a more minimalist setting in the offline contextual bandits setting in [7]


- If the authors do plan to claim the separaion between Tsallis regularization and vanlla entropy regularization for learning against the unregularized objective at the end of Section 5, they should not compare two upper bounds.
- The study of learning w.r.t. divergence-regularized value functions and objectives dates back to a long line a previous effors **no later than** [2, 3], at least in the episodic MDP setting. And **divergence-regularized performance difference lemma** appeared **no latter than** Section 5 in [8] and Lemma 3 in [5]
  - If the authors do consider their analysis is totally unrelated to the divergence-regularized performance difference lemma (or the so-called soft peformance difference lemma) in the literature, they should justify it **technically instead of secretly**.
  - Also, the **divergence-regularied Bellman operator** has been illustrated in detail in both [3] and [5], which is certainly not a "newly introduced" concept in this submission.
- The reviewer does not want to claim that the authors are **plagiarizing or rephrasing previous works in a more involved way**, but the authors should respect the previous efforts in the theory community in a decent way.
  - If the authors do consider the discounted setting in this submission is fundamentally different from the finite-horizon episodic MDP settings or the contextual bandit settings considered in [1-10] in the literature and does not plan to discuss the relation between this submission and any of [1-10], they should justify it **technically instead of secretly**.
  - **To AC: the reviewer is willing to discuss about this point if necessary.**


References

[1] Foster, Dylan J., Zakaria Mhammedi, and Dhruv Rohatgi. "Is a Good Foundation Necessary for Efficient Reinforcement Learning? The Computational Role of the Base Model in Exploration." arXiv preprint arXiv:2503.07453 (2025).


[2] Xiong, Wei, et al. "Iterative preference learning from human feedback: Bridging theory and practice for rlhf under kl-constraint." arXiv preprint arXiv:2312.11456 (2023).

[3] Xie, Tengyang, et al. "Exploratory preference optimization: Harnessing implicit q*-approximation for sample-efficient rlhf." arXiv preprint arXiv:2405.21046 (2024).

[4] Huang, Jiawei, et al. "Can rlhf be more efficient with imperfect reward models? a policy coverage perspective." arXiv preprint arXiv:2502.19255 (2025).

[5] Yuan, Yurun, et al. "Trajectory Bellman Residual Minimization: A Simple Value-Based Method for LLM Reasoning." arXiv preprint arXiv:2505.15311 (2025).


[6] Zhao, Heyang, et al. "Sharp analysis for kl-regularized contextual bandits and rlhf." arXiv preprint arXiv:2411.04625 (2024).

[7] Zhao, Qingyue, et al. "Towards a Sharp Analysis of Offline Policy Learning for $ f $-Divergence-Regularized Contextual Bandits." arXiv preprint arXiv:2502.06051 (2025).

[8] Zhao, Heyang, et al. "Logarithmic regret for online kl-regularized reinforcement learning." arXiv preprint arXiv:2502.07460 (2025).


[9] Aminian, Gholamali, et al. "Theoretical Analysis of KL-regularized RLHF with Multiple Reference Models." arXiv preprint arXiv:2502.01203 (2025).


[10] Huang, Audrey, et al. "Is best-of-n the best of them? coverage, scaling, and optimality in inference-time alignment." arXiv preprint arXiv:2503.21878 (2025).

**Questions:**

N/A

**Details Of Ethics Concerns:**

The authors are plagiarizing or rephrasing previous works in a more involved way, which has been detailed by the reviewer in the **Weaknesses** section.

---

> ### Author Response · Authors · 2025-11-17
>
> We thank the reviewer for the time and effort spent evaluating our submission. We appreciate the acknowledgment that “this is the first work deriving fast rates for learning with respect to divergence-regularized objectives in the discounted setting” and for confirming the correctness of our proofs.
>
> However, the review also raises several technical objections that seem to result from misunderstandings of our framework. More importantly, it contains a serious allegation regarding research integrity. We address this issue first, before responding to the technical points in detail.
>
> ##  On the ethics flag and allegation of plagiarism
>
> The review claims that our work “plagiarizes or rephrases previous efforts,” but does not identify any copied text, results, derivations, or arguments. This claim is made without specific evidence. We state unequivocally:
>
> > **Our work is original and does not plagiarize any existing work.**
>
> We emphasize that our paper, like all scientific research, **builds upon prior foundations while contributing novel theoretical and methodological insights**. In particular, our contributions regarding the **coupled $f$-divergence–induced parametrizations for policy gradient methods in discounted MDPs** are original and have not appeared in the cited literature.
>
> Below, we provide a detailed clarification of how our approach differs from prior work, addressing both conceptual and technical distinctions. We hope this will dispel any misunderstanding regarding the novelty and originality of our contribution.
>
> ##  Core misunderstanding
>
> We believe many of the reviewer's concerns arise from a misunderstanding of our work's scope. Our paper examines **policy gradient methods** and how their learning dynamics and convergence are influenced by the combined choice of regularization and parametrization. In contrast, the works cited by the reviewer [1-10] do not analyze policy gradient methods under any parametrization. Instead, they focus on value-based or preference-learning methods that estimate value functions (or preferences) from inexact reward or value signals and then derive policies from these estimates. **These are fundamentally different algorithmic families**, and this distinction is central to our contribution. We believe conflating these settings has led the reviewer to perceive our work as more overlapping than it actually is.
>
> In the reviewer’s summary, our contribution is described as *“the log-linear policy with a reference model under KL-regularization is extended to the so-called coupled parameterization.”*
> This characterization is **inaccurate** and does not reflect the main contribution of our paper.
>
> Our proposed algorithmic framework is **not** a simple extension of the log-linear (softmax) policy. Instead, it is built upon a **divergence-induced parameterization** that is *explicitly matched* to the chosen $f$-divergence regularizer. Concretely, we introduce a new parameterization derived from the **$f$-softargmax operator** (Eq.~(9), Sec.~4), which establishes a principled **coupling between the policy parameterization and the divergence** used in the regularized objective — hence the term *coupled*.
>
> The **standard softmax policy** emerges only as a **special case** corresponding to the KL divergence, i.e., when $f(u) = u \log u - (u - 1)$.
> Our framework therefore **generalizes the log-linear parameterization** to a broad and theoretically grounded family of $f$-divergence–induced parameterizations, offering new insights into the structure of policy gradient methods under regularization.
>
> To the best of our knowledge, our work is the first to employ such general $f$-divergence–induced parametrizations in the **analysis of policy gradient methods** in reinforcement learning, and to derive explicit convergence guarantees in this setting.

---

> > ### Author Response · Authors · 2025-11-17
> >
> > ## Response to technical comments
> >
> > > **"Every time $[f']^{-1}$ appears in this submission, it is mathematically wrong: because, for example, chi-square divergence is already nice enough, but the $a \mapsto \max\{0, a\}$ (also appears as ReLU in the lierature) in the Lemma G.2 of [10] already certificates that it is not possible to simply subsume the solution to this constrained optimization problem using $[f']^{-1}(\cdot)$ and the complementary slackness condition might need to be tackled in a case-by-case way"**
> >
> > This objection does not apply to our work. The chi-square divergence generator $f\colon t \rightarrow (t-1)^2$ does not satisfy our assumption $A_{f}(\pi_{ref})$ because $f'(t) = 2(t-1) \not\to -\infty$ as $t \to 0$ and is therefore explicitly excluded from our analysis, along with all *sparse* divergences.
> >
> > This exclusion is intentional: *sparse* divergences lead to **non-smooth parametrization**, as discussed after Assumption $A_f$, lines 211–216, and in a discussion below. Therefore, the chi-square example in the review is not a counterexample to any claim made in our paper, and we firmly believe our results are mathematically correct. If the reviewer finds it helpful, we can add a more explicit remark clarifying why sparse divergences do not satisfy our assumptions.
> >
> > > **"The smoothness property  is trivial and does not even deserve a theorem, i.e., Theorem 4.3, because it is well known that the Fenchel conjugate of a $\alpha$-strongly-convex function is $\alpha^{-1}$-smooth (under very mild qualitative regularity conditions); which is nearly exactly the case for the regularized objective with equation (9) as the parametrization."**
> >
> > We respectfully disagree with the reviewer's comment. **The regularized value function $v_{\pi_{\theta}}^{f}(\rho)$ is not the Fenchel conjugate of the divergence generator**. If it were, even the regularized value function with softmax parameterization would be convex, which is known to be false [13,14]. Instead, the regularized value function is defined as the fixed point of the regularized Bellman operator $\mathsf{T}_{\pi}^{f}$ [11] and is **non-convex** with respect to both $\pi$ and the softmax parameter $\theta$.
> >
> > While it is true that the optimal value function in the bandit setting is a Fenchel conjugate of the corresponding divergence, **this holds only in the policy space**. In contrast, our analysis establishes the smoothness property in **the parameter space of general value functions for MDPs**, where the smoothness of the parameterization is central.
> >
> > Regarding the smoothness of the parameterization, we emphasize that **strong convexity alone is not sufficient to guarantee smoothness of the value function as a function of parameters**, even in the bandit setting. For example, parameterizations induced by $\chi^2$-divergence are sparse (Roulet et al., 2025) and therefore not smooth, since small perturbations of the parameters do not necessarily change the entries set to zero. As a concrete counterexample, consider a non-regularized bandit with $A=3$ and rewards $r(0)=1, r(1)=0, r(2)=0$. In this case, the value function is $\pi_\theta(0)$, where the first term is defined via the ReLU function (as acknowledged by the reviewer in Lemma G.2 of [10]), **and thus cannot be smooth**. In the policy space, however, this function is linear and smooth. We will clarify this distinction further in the discussion following Theorem 4.2 to avoid confusion.
> >
> > Finally, if the reviewer has a shorter or alternative proof of Theorem 4.2 with a similar bound, we would be glad to incorporate it into the manuscript and acknowledge their contribution.
> >
> > > **" the reviewer strongly disagrees with Line 215-216 because the analysis of learning w.r.t. the reverse-KL-regularized objectives have become sharp (in terms of the dependency on $\epsilon$, i.e., $\tilde{O}(\epsilon^{-1})$) for contextual bandits and episodic MDPs in the hybrid setting [6, 1], for contextual bandits in the offline setting [9,7], and for both contextual bandits and episodic MDPs in the online setting [8]."**
> >
> > We believe there is a misunderstanding regarding the distinction between **KL divergence** and **reverse KL divergence**.
> > The KL divergence is generated by $f(t) = t \ln t - (t-1)$, whereas the reverse KL divergence corresponds to the divergence generator $f(t) = -\ln t$: these two divergence have very different properties.
> > For a concise comparison of these divergences, we refer the reviewer to standard references, e.g. [16].
> >
> > We appreciate the reviewer’s effort in providing additional references, and we will include them in the revised version to emphasize the extensive prior exploration of KL-based regularization. However, we reiterate that our contribution lies precisely in **broadening the scope of policy gradient methods in MDPs beyond the KL case**, by introducing and analyzing **a family of parameterizations specifically adapted to  general $f$-divergences penalization**.

---

> > > ### Author Response · Authors · 2025-11-17
> > >
> > > > **'Even without Assumption P, and even in the pure offline setting without exploration, the analysis of learning against many divergence regularized objectives is doable, as manifested in the [7]'**
> > >
> > > Thank you for bringing up these references, which we will incorporate in our discussion to acknowledge that $f$-divergence regularization has attracted attention in other domains.
> > > That said, we wish to clarify several key differences between [7] and our work, as they **address fundamentally different problem settings**.
> > >
> > > First, [7] studies  the **offline contextual bandit setting** and therefore **does not involve any policy gradient methods**. Moreover, it requires $f$ to be strongly convex, which **excludes** both the **KL divergence** and **$\alpha$-Tsallis divergences** with $\alpha \in (0,1)$.
> > >
> > > Regarding Assumption P, we note that it is primarily a technical assumption. Under our Assumption $A_f$ on the $f$-divergence, $\pi^{ref}(a|s) = 0$ automatically implies $\pi_\theta(a|s) = 0$ for any parameter $\theta$ (Lemma B.1, a simple corollary of Proposition 1 in Roulet et al., 2025, as noted on line 789) and $\pi_\theta$ does not depend on $\theta(s,a)$ at all. Allowing a reference policy to be zero only requires handling an action space of variable size under Assumption P, which seems to be feasible. Alternatively, we could simply assume $\pi^{ref}(a^\star|s) > 0$ for some optimal action $a^\star$, without altering the essence of our argument. Within this framework, our results would more align with the results of [7]. We are happy to include a discussion of this point in the revised manuscript.
> > >
> > > Regarding the exploration assumption, we emphasize that **it is standard in analyses of policy gradient methods** (e.g., Assumption 2 in [13] and the implicit assumption $\tilde{\rho} > 0$ following Lemma 2.6 in [15]). To our knowledge, no existing policy gradient analysis fully avoids this assumption. In standard offline RL, removing the need for sufficient exploration typically requires additional pessimistic arguments to control the importance ratio between the optimal occupancy measure and the data-generating distribution, an approach not directly available in policy gradient analyses.
> > >
> > > If the reviewer has suggestions for **weakening or reformulating** this assumption within our setting, we would be very interested in exploring them.
> > >
> > > Finally, we highlight that *under this assumption* our paper provides **the first convergence rate with explicit constants** and **a theoretical demonstration that appropriate parameterization choices can accelerate learning**. This constitutes a substantive advance in the theoretical understanding of policy gradient methods.
> > >
> > > - **"The term "coupled parametrization" is confusing because it is just an extension of the "log-linear policy with a reference policy" to the general $f$-divergence setting"**
> > >
> > > We respectfully disagree with the reviewer’s statement that *the term “coupled parametrization” merely extends the log-linear policy with a reference policy*. While our framework does accommodate a broader family of regularizations, this is only **one aspect** of our contribution. Our work is **more fundamental**, as we introduce **new parameterizations that are inherently matched to the specific divergence used for regularization**, and we provide **explicit convergence rates** under these parameterizations.
> > >
> > > The term *“coupled parametrization”* is intended to highlight that the **policy parameterization is derived from the same $f$-divergence** that defines the regularizer in the objective function. Throughout the paper, it is consistently presented together with its corresponding divergence-based objective.
> > >
> > > That said, we understand that the terminology may have caused confusion. If the reviewer finds it clearer, we are happy to adopt an alternative, more explicit name such as **$f$-softargmax parametrization**, in analogy with the standard *softmax* parameterization, to better convey the structural link between the divergence and the induced policy representation.

---

> > > > ### Author Response · Authors · 2025-11-17
> > > >
> > > > > **"If the authors do plan to claim the separation between Tsallis regularization and vanilla entropy regularization
> > > > for learning against the unregularized objective at the end of Section 5, they should not compare two upper
> > > > bounds."**
> > > >
> > > > We thank the reviewer for raising this important point. It has been previously established in [12] that softmax policy gradient methods can require exponential time to converge. While we are not aware of similar lower bounds for the optimization of regularized objectives, we also note that no existing regularization is known to achieve a polynomial-time upper bound when softmax parametrization is used without any preconditioning.
> > > >
> > > > Our approach **demonstrates that this exponential dependency can be overcome by using Tsallis divergence regularization together with its coupled parametrization** (or equivalently, a Tsallis-softargmax parametrization), yielding explicit polynomial-time convergence guarantees. We appreciate the reviewer highlighting the potential for confusion and will clarify this point in the revised manuscript.
> > > >
> > > > > **"'The study of learning w.r.t. divergence-regularized value functions and objectives dates back to a long line
> > > > a previous efforts no later than [2, 3], at least in the episodic MDP setting. And divergence-regularized
> > > > performance difference lemma appeared no latter than Section 5 in [8] and Lemma 3 in [5]"**
> > > >
> > > > We would like to clarify that we never claimed the proof of the performance-difference lemma as our main technical contribution. We acknowledge that this result is standard in the literature; however, we were unable to find a version that formally applies to arbitrary regularizers beyond KL divergence or entropy. If the reviewer is aware of specific references covering this case, we would be glad to include them in the manuscript.
> > > >
> > > > Additionally, all references provided by the reviewer **focus on value-based reinforcement learning with KL regularization**. None of these works derive explicit finite-time convergence guarantees for *policy gradient methods* with softmax parameterization, nor do they extend to other $f$-divergences.
> > > >
> > > > We emphasize that **the main contribution of our paper is to introduce coupled parameterizations for policy gradient methods with general $f$-divergence regularization and to derive explicit convergence rates**, which **substantially departs from all works cited by the reviewer**.
> > > >
> > > > > **"If the authors do consider their analysis is totally unrelated to the divergence-regularized performance difference lemma (or the so-called soft peformance difference lemma) in the literature, they should justify it technically instead of secretly."**
> > > >
> > > > We would like to clarify that we **do not** claim that our results are totally unrelated to the divergence-regularized performance difference lemma. Indeed, the performance-difference lemma is a **standard** and **very useful tool** for the analysis of many RL algorithms, and we fully acknowledge its importance in prior work.
> > > >
> > > > We also never claimed it as **our original technical contribution**, and did not try to **conceal prior work** on this matter. The reason why we re-derive this lemma stems from the fact that we did not find a version that can handle arbitrary $f$-divergence regularizers.
> > > >
> > > > If the reviewer is aware of a reference that establishes such a general version, we would be **happy to cite it** and clarify this connection explicitly in the revised manuscript.
> > > >
> > > > > **"Also, the divergence-regularied Bellman operator has been illustrated in detail in both [3] and [5], which is certainly not a "newly introduced" concept in this submission."**
> > > >
> > > > Once again, we would like to emphasize that we **do not claim the divergence-regularized Bellman operator is a newly introduced concept** in this submission, and we apologize if any part of the text suggested otherwise. We have **explicitly credited** this concept to [11] (see lines 120–122).
> > > >
> > > > If the reviewer is aware of **earlier formulations that extend to general $f$-divergence regularizers**, we would be **glad to include and acknowledge them** in the revised manuscript.

---

> ### Author Response · Authors · 2025-11-17
>
> > **"If the authors do consider the discounted setting in this submission is fundamentally different from the finite-horizon episodic MDP settings or the contextual bandit settings considered in [1-10] in the literature and does not plan to discuss the relation between this submission and any of [1-10], they should justify it technically instead of secretly"**
>
> Regarding the discounted setting, we emphasize that **the main distinction between our work and [1–10] lies in the problem setting**. Our paper studies **policy gradient methods for general MDPs**, a family of algorithms  **fundamentally different** from **value-based**, **preference-based**, or **bandit** approaches. We extend policy gradient methods through **novel $f$-divergence-induced parameterizations** and introduce a **new analytical framework** for their convergence analysis.
>
> Papers [1–6] and [8–10] investigate reinforcement learning with **KL regularization**, which corresponds to only a **special case** of our framework, as we consider **general $f$-divergences**. Moreover, even in the KL setting, **none** of these works provide *finite-sample convergence guarantees with explicit constants* for the KL-regularized policy gradient method under *softmax parameterization*.
>
> Paper [7], in contrast, focuses on the **offline contextual bandit** setting with **strongly convex regularizers**, which **excludes** the Tsallis divergences for $ \alpha \in (0,1) $.
>
> We will include all the papers mentioned by the reviewer in the revised manuscript and will **explicitly note that KL regularization has been extensively studied in prior RL research**. We thank the reviewer for these helpful references, which we believe will **enhance the completeness and balance of our discussion of related work**.
>
> ##  Concluding remarks
>
> We thank the reviewer for the additional references and for highlighting areas where our discussion of related work can be further expanded. We hope the clarifications provided above **address all remaining technical concerns**.
>
> Regarding the **allegation of plagiarism**, we respectfully emphasize that our work is **entirely original** and **appropriately cites prior literature**. We acknowledge that some related references were not included in the initial submission and appreciate the reviewer’s effort in bringing them to our attention.
>
> If the reviewer continues to believe that our work constitutes plagiarism, we **respectfully request specific references or evidence** indicating where our results or derivations have previously appeared, so that we may address them precisely and transparently.
> Absent such evidence, we **kindly ask that the plagiarism flag be reconsidered**, and we remain fully available to provide any further clarification or discussion the reviewer deems necessary.
>
> ### References
>
> [1] Foster, Dylan J., Zakaria Mhammedi, and Dhruv Rohatgi. "Is a Good Foundation Necessary for Efficient Reinforcement Learning? The Computational Role of the Base Model in Exploration." arXiv (2025).
>
> [2] Xiong, Wei, et al. "Iterative preference learning from human feedback: Bridging theory and practice for rlhf under kl-constraint." arXiv preprint arXiv:2312.11456 (2023).
>
> [3] Xie, Tengyang, et al. "Exploratory preference optimization: Harnessing implicit q*-approximation for sample-efficient rlhf." arXiv preprint arXiv:2405.21046 (2024).
>
> [4] Huang, Jiawei, et al. "Can rlhf be more efficient with imperfect reward models? a policy coverage perspective." arXiv preprint arXiv:2502.19255 (2025).
>
> [5] Yuan, Yurun, et al. "Trajectory Bellman Residual Minimization: A Simple Value-Based Method for LLM Reasoning." arXiv preprint arXiv:2505.15311 (2025).
>
> [6] Zhao, Heyang, et al. "Sharp analysis for kl-regularized contextual bandits and rlhf." arXiv  (2024).
>
> [7] Zhao, Qingyue, et al. "Towards a Sharp Analysis of Offline Policy Learning for f-Divergence-Regularized Contextual Bandits." arXiv (2025).
>
> [8] Zhao, Heyang, et al. "Logarithmic regret for online kl-regularized reinforcement learning." arXiv  (2025).
>
> [9] Aminian, Gholamali, et al. "Theoretical Analysis of KL-regularized RLHF with Multiple Reference Models." arXiv (2025).
>
> [10] Huang, Audrey, et al. "Is best-of-n the best of them? coverage, scaling, and optimality in inference-time alignment." arXiv (2025).
>
> [11] Geist et al., 2019, A theory of regularized Markov decision processes, ICML 2019
>
> [12] Li et al., 2023, Softmax policy gradient methods can take exponential time to converge, Mathematical Programming
>
> [13] Mei et al., 2020, On the Global Convergence Rates of Softmax Policy Gradient Methods, ICML 2020
>
> [14] Labbi et al., 2025, On Global Convergence Rates for Federated Policy Gradient under Heterogeneous Environment, arXiv preprint
>
> [15] Liu, Jiacai, Wenye Li, and Ke Wei. "Elementary analysis of policy gradient methods." arXiv preprint
>
> [16] https://en.wikipedia.org/wiki/F-divergence

---

### Meta-Review · Area_Chair_C4SN · 2026-01-04

**Summary:**

I recommend an acceptance of this paper. The result is interesting and brings new insight in policy gradient research. The review giving strong rejection might be heavily skewed by the unfounded plagiarism claim.

**Reviewer Concerns:**

Main concerns

Reviewer q88P:
- Reviewer brought concern of plagiarism but no evidence was provided or found
- Reviewer uses chi square and other sparse cases as counterexamples.
- - Authors state such sparse divergences are intentionally excluded because they violate assumptions and yield non smooth parameterizations.
- Reviewer points out that two upper bounds should not be compared
- - Authors clarify that softmax policy gradient’s known lower bound is exponential on some problems, and their Tsallis coupled method has a polynomial time upper bound under their assumptions, so their method can avoid that known worst case behavior.

Reviewer s4Jk:
- Theory focus is on regularized setting only, not direct unregularized convergence guarantees.
- - Authors agree and frame unregularized solving via a small regularization coefficient, but direct unregularized guarantees are left as future work.
- mentions missing comparison to Natural Policy Gradient and unclear efficiency
- - Authors add NPG comparisons and claim the new parameterization can beat softmax and sometimes NPG, while emphasizing focus on first order methods without preconditioning.

Reviewer zseX:
- Contribution may mainly recover existing regularizers (entropy, Tsallis), not a new better one.
- - This aspect is not resolved

Reviewer ri98:
- Experiments too small (5×5 GridWorld) and unclear advantage over KL plus softmax.
- - Authors replace GridWorld with DeepSea and report KL better on easy cases, Tsallis better when exploration is harder.

**Reviewer Scores:**

It is hard to say who would have increased scores but the authors addressed the main concerns of the reviewers.
Bad

---

### Decision · Program_Chairs · 2026-01-26

Accept (Poster)